# Scalable Spatio-Temporal SE(3) Diffusion for Long-Horizon Protein Dynamics

**Nima Shoghi**[1,2]*, **Yuxuan Liu**[1,3]*, **Yuning Shen**[1], **Rob Brekelmans**[1], **Pan Li**[2], **Quanquan Gu**[1,†]

[1]ByteDance Seed  [2]Georgia Institute of Technology  [3]University of California, Los Angeles

{yuning.shen, quanquan.gu, rob.brekelmans3}@bytedance.com
{nimash, panli}@gatech.edu,  yxliu@math.ucla.edu

## Abstract

Molecular dynamics (MD) simulations remain the gold standard for studying protein dynamics, but their computational cost limits access to biologically relevant timescales. Recent generative models have shown promise in accelerating simulations, yet they struggle with long-horizon generation due to architectural constraints, error accumulation, and inadequate modeling of spatio-temporal dynamics. We present **STAR-MD** (Spatio-Temporal Autoregressive Rollout for Molecular Dynamics), a scalable SE(3)-equivariant diffusion model that generates physically plausible protein trajectories over microsecond timescales. Our key innovation is a causal diffusion transformer with joint spatio-temporal attention that efficiently captures complex space-time dependencies while avoiding the memory bottlenecks of existing methods. On the standard ATLAS benchmark, STAR-MD achieves state-of-the-art performance across all metrics–substantially improving conformational coverage, structural validity, and dynamic fidelity compared to previous methods. STAR-MD successfully extrapolates to generate stable microsecond-scale trajectories where baseline methods fail catastrophically, maintaining high structural quality throughout the extended rollout. Our comprehensive evaluation reveals severe limitations in current models for long-horizon generation, while demonstrating that STAR-MD's joint spatio-temporal modeling enables robust dynamics simulation at biologically relevant timescales, paving the way for accelerated exploration of protein function. Project page: https://bytedance-seed.github.io/ConfRover/starmd.

## 1 Introduction

Protein functions emerge from conformational dynamics – the continuous structural changes that underlie important biological processes such as catalysis, binding, and allosteric regulations (McCammon, 1984; Berendsen & Hayward, 2000). Classical molecular dynamics (MD) simulation remains the gold standard and relies on physical models to integrate atomic motions over time using Newtonian mechanics. However, the need for small integration steps ($\sim$ femtoseconds) severely limits its practicality for exploring the microsecond–millisecond timescales often required to capture biologically relevant events. Recent advances in protein structure prediction (Jumper et al., 2021; Abramson et al., 2024; Lin et al., 2023) and generative modeling for conformational dynamics (Jing et al., 2024b; Cheng et al., 2025; Shen et al., 2025; Costa et al., 2024) offer promising data-driven approaches to accelerate simulations by learning the dynamics directly from data and generating trajectories at coarser temporal resolutions. However, existing methods are still constrained by short time horizons (typically up to nanoseconds) and struggle to scale to larger proteins. These limitations highlight an urgent need for generative models capable of producing physically plausible protein trajectories over extended timescales while remaining computationally efficient.

Scaling protein dynamics modeling to long timescales requires both *an efficient structural representation* and *an architecture capable of capturing complex spatio-temporal dependencies*. Current approaches often rely on pairwise residue representations or computationally expensive architectures (e.g., AlphaFold2-style triangular attention), leading to quadratic memory growth and cubic computational cost with respect to protein size. These challenges become even more pronounced when accounting for spatio-temporal coupling across multiple conformations during trajectory modeling.

---

*Work done during internship at ByteDance Seed. [†]Corresponding author.

As a result, most existing models treat spatial and temporal components separately, employing interleaved "spatial" and "temporal" modules (Jing et al., 2024b; Cheng et al., 2025; Shen et al., 2025), which limits their expressiveness in capturing coupled dynamics. These limitations confine current models to small proteins and short simulation horizons, ultimately hindering their ability to learn long-range temporal dependencies and generate high-quality conformations over extended rollouts.

We introduce **S**patio-**T**emporal **A**utoregressive **R**ollout for **M**olecular **D**ynamics (STAR-MD), an SE(3)-equivariant autoregressive diffusion model for generating physically plausible protein trajectories over microsecond timescales, even for large protein systems. At its core, STAR-MD employs *a causal diffusion transformer* with *joint spatio-temporal attention*, enabling improved autoregressive generation by dynamically computing attention over historical context during diffusion and capturing long-range dependencies through expressive yet memory-efficient spatio-temporal modeling.

Our main contributions are summarized as follows:

- We present STAR-MD, a novel autoregressive diffusion transformer that leverages efficient spatio-temporal attention to model the complex dependencies underlying protein dynamics.
- Through several key technical improvements, including historical context noise addition during training and inference, block-diffusion-style causal training, and architectural optimizations, STAR-MD achieves efficient, scalable training and stable trajectory generation, significantly improving over current state-of-the-art models.
- We perform an exhaustive evaluation across multiple different simulation timescales ranging from $100\,\text{ns}$ to $1\,\mu\text{s}$, providing a comprehensive assessment of conformation quality, coverage, and dynamic fidelity. Our analyses yield new insights into the limitations of current state-of-the-art models in long-horizon generation and offer valuable guidance for future model design.

## 2 RELATED WORK

**Protein Conformation Generation.** Several models (Jing et al., 2024a; Lewis et al., 2024; Wang et al., 2024) build on advances in protein structure prediction (Jumper et al., 2021; Abramson et al., 2024; Lin et al., 2023) and diffusion models (Ho et al., 2020) to directly generate time-independent conformations. These methods provide an efficient alternative to MD by enabling parallel sampling. However, they capture only the equilibrium distribution of conformations and cannot model the temporal evolution of protein dynamics.

**MD Trajectory Generation.** To model the temporal evolution, operator-based methods Klein et al. (2024); Costa et al. (2024) aim to learn transport operators that predict conformations at lagged times. These methods approximate evolution as a Markovian process, and thus fail to capture the non-Markovian properties often present in partially observed systems such as protein dynamics data. In contrast, methods generating trajectories, akin to video generations, consider dependencies across multiple time frames. AlphaFolding (Cheng et al., 2025) incorporates higher-order information through additional "motion frames" and generates multiple future frames simultaneously. Similarly, MDGen (Jing et al., 2024b) models the joint distribution of frames across 100-ns trajectories. However, both methods only capture the dependencies in a fixed context and prediction window. When generating longer trajectories through extension, they discard memory of earlier windows, breaking temporal consistency. To address this, ConfRover (Shen et al., 2025) adopts language model-style autoregression that can generate trajectories of arbitrary lengths while maintaining full memory via KV-caching. STAR-MD follows this approach to avoid the "memory break" in block-based models.

**Scalable Structural and Temporal Modeling.** Current structural models, including AlphaFolding and ConfRover, represent protein structures using single and pairwise features and process them with specialized architectures such as Pairformer and Invariant Point Attention (IPA) (Jumper et al., 2021). While these modules are highly expressive and preserved the required roto-translational symmetries, they are computationally and memory intensive. This limits model efficiency and scalability, especially when extended to modeling trajectories across multiple time frames. MDGen sidesteps this limitation by anchoring trajectories to key frames, encoding only single embeddings with standard transformers. While this allows modeling of long trajectories (e.g., 250 frames for ATLAS), the reliance on key frames limited flexibility and model performance is suboptimal. Lastly, all the above approaches model dynamics between frames using interleaved spatial and temporal modules. While this decoupled design reduces computational cost, it limits the model's expressive

power to capture complex spatio-temporal relationships. In contrast, STAR-MD proposes to employ joint spatio-temporal attentions on the single embeddings' direct space-time processing while keeping the memory footprint manageable.

## 3 METHOD

To generate long and physically plausible protein trajectories, we propose **S**patio-**T**emporal **A**utoregressive **R**ollout for **M**olecular **D**ynamics (STAR-MD), a spatio-temporal SE(3)-diffusion model that operates within an autoregressive framework. In the following sections, we first describe the overall autoregressive diffusion framework, then detail the architecture of our diffusion model, and finally explain the training and inference procedures.

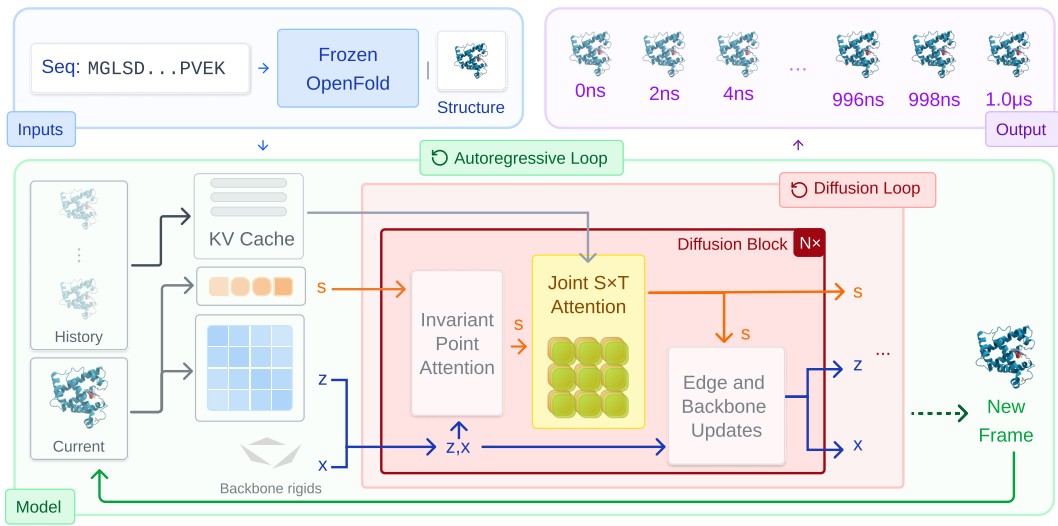

Figure 1: **Overview of STAR-MD generation.** Input contains protein sequence and a starting conformation. In autoregressive diffusion generation, structural information of previously generated conformations and current noisy conformations are encoded into single and pair representations. A joint spatio-temporal attention block is employed to capture context information to update the single representation of the current frame. The main model block iterated to diffuse a clean conformation for the current frame and added to history for generating next frames.

### 3.1 AUTOREGRESSIVE SE(3)-DIFFUSION MODELS FOR PROTEIN TRAJECTORIES

**Autoregressive Trajectory Generation.** We aim to generate protein conformation *trajectories* $\boldsymbol{x}_{1:L}$ in an autoregressive fashion $\prod_{\ell=1}^{L} p(\boldsymbol{x}_\ell \,|\, \boldsymbol{x}_{<\ell}, \Delta t_\ell)$, where $\Delta t_\ell$ represents the time interval between *frames*. This formulation amortizes trajectory generation into a frame-level process conditioned on the entire past history, enabling the model to generate future dynamics of arbitrary length while maintaining a flexible memory representation learned from data. Causal transformers, widely used in language models for efficient autoregressive sequence modeling (Touvron et al., 2023), provide a natural architectural choice for this task.

**SE(3) Diffusion Models.** To model the generation process of each frame $p(\boldsymbol{x}_\ell|\boldsymbol{x}_{<\ell}, \Delta t_\ell)$, we use a (conditional) diffusion model on the Riemannian manifold SE(3), where $\boldsymbol{x} = [\boldsymbol{T}, \boldsymbol{R}]$ which captures translation positions $\boldsymbol{T} = \{T^i\}_{i=1}^N \in \mathbb{R}^{3N}$ and rotations $\boldsymbol{R} = \{R^i\}_{i=1}^N \in \mathrm{SO}(3)^N$ for each amino-acid residue in a protein sequence of length $N$ (Yim et al., 2023; Wang et al., 2024; Shen et al., 2025). The forward diffusion process independently corrupts translations and rotations across the diffusion time $\tau$

$$\boldsymbol{T}_\tau = \sqrt{\alpha_\tau}\boldsymbol{T}_0 + \sqrt{1-\alpha_\tau}\epsilon, \quad \epsilon \sim \mathcal{N}(0, I_{3N}) \tag{1}$$

$$\boldsymbol{R}_\tau \sim \mathrm{IGSO}_3(\boldsymbol{R}_0, \sigma_\tau^2), \tag{2}$$

where $\mathrm{IGSO}_3$ is the isotropic Gaussian on SO(3). Given a noisy structure $\boldsymbol{x}^\tau$, diffusion time $\tau$, and condition $\boldsymbol{c}$, a denoising score network

$$s_\theta\left(\boldsymbol{x}^\tau, \tau, \boldsymbol{c}\right) = \left[s_\theta^{\boldsymbol{T}}, s_\theta^{\boldsymbol{R}}\right](\boldsymbol{x}^\tau, \tau, \boldsymbol{c}) \tag{3}$$

is trained via denoising score matching to predict the noise added to both components. During inference, the learned score function can be used to sample from the reverse diffusion process and generate protein structures.

To capture temporal dependencies needed for autoregressive generation, the condition $c$ for generating the current frame should incorporate all past history $x_{<\ell}$. Specifically, we design an efficient and expressive *causal diffusion transformer* network to compute $c\left(x_{<\ell}^0, x_\ell^\tau, \tau\right)$ from the preceding clean frames and the current noisy frame $x_\ell^\tau$. This design differs from prior works such as MDGen (Jing et al., 2024b) & Alphafolding (Cheng et al., 2025) which do not include autoregressive conditioning, and ConfRover (Shen et al., 2025) which compresses all preceding frames into a static condition $c\left(x_{<\ell}\right)$. In the following section, we detail the design of $c\left(x_{<\ell}^0, x_\ell^\tau, \tau\right)$ and how it efficiently integrates spatio-temporal information from preceding frames.

## 3.2 ARCHITECTURE

In this section, we describe the architecture choices underlying STAR-MD. The main features of this architecture lie in the diffusion decoder, which uses *spatio-temporal attention* for autoregressive conditioning on previous frames and *block diffusion* allowing for efficient training and KV caching.

**Input Module.** We mirror the `FrameEncoder` module in Shen et al. (2025), starting with a frozen OpenFold `FoldingModule` for sequence-level single-residue and pairwise features $s, z = \{s_i, z_{ij}\}_{i,j=1}^N$. We incorporate these time-independent features with amino-acid metadata and pairwise $C_\beta$ distance within each frame to obtain time-dependent features $s_\ell^{\text{init}}, z_\ell^{\text{init}}$, which are used as initial input to the diffusion blocks.

**Diffusion Blocks.** Our diffusion module contains submodules stacked into blocks:

1. Invariant Point Attention layers (Jumper et al., 2021) update single features $s_\ell$ using pair features $z_\ell$ and noisy frames $x_\ell^\tau$. This step is independent for each frame.
2. Joint Spatio-Temporal Attention layers update the single-residue features $s_\ell$ by attending to those from previous frames $s_{\leq\ell}$, which allows for exchange of temporal informaiton.
3. Pair features $z_\ell$ are updated from single features $s_\ell$ using the `EdgeTransition` layer from Jumper et al. (2021). Noisy frames $x_\ell^\tau$ are updated from single features $s_\ell$ using an MLP-based backbone update.

After stacking the above blocks, the final update of the coordinates $x_\ell^\tau$ is used as the score function prediction and fed to the denoising score matching loss.

**Joint Spatio-Temporal Attention.** Our joint spatio-temporal (S×T) attention mechanism integrates information across the temporal dimensions, departing from previous models that employ factorized, "space-then-time" attention which imposes a restrictive bias that spatial and temporal dependencies are separable. Instead, our S×T attention operates on tokens representing residue-frame pairs $(i, \ell)$, allowing it to directly model non-separable relationships, such as how motion at one residue is coupled to past motion at a distant site. 2D Rotary Position Embedding (2D-RoPE) (Heo et al., 2024) is used to embed residue and frame indices, which allows extrapolation past the training number of frames.

This architectural choice is key to our model's scalability. We analyze the per-layer computational complexity of our S×T attention ($\mathcal{O}(N^2 L^2)$) against general S+T (i.e., "space-then-time") architectural paradigms (see Appendix B for details). Most baseline architectures employ a Pairformer (Jumper et al., 2021) backbone, incurring a cubic spatial complexity $\mathcal{O}(N^3 L)$ due to expensive triangular attention operations. We consider two representative baseline configurations: (1) **Pairformer + Pair Temporal Attention** (e.g., ConfRover (Shen et al., 2025)): This configuration scales as $\mathcal{O}(N^3 L + N^2 L^2)$. Here, the pairwise temporal attention alone incurs $\mathcal{O}(N^2 L^2)$ cost, matching that of STAR-MD, but the additional cubic spatial overhead makes it significantly more expensive. Furthermore, this approach incurs a quadratic memory cost $\mathcal{O}(N^2 L)$ for caching pairwise key-value states during autoregressive generation, which becomes a severe bottleneck for long rollouts. In contrast, STAR-MD only requires $\mathcal{O}(NL)$ memory for caching single-residue features. (2) **Pairformer + Single Temporal Attention**: This configuration scales as $\mathcal{O}(N^3 L + NL^2)$. While the temporal scaling is theoretically superior to STAR-MD's $\mathcal{O}(N^2 L^2)$ in the limit $L \gg N$, the cubic spatial cost $\mathcal{O}(N^3 L)$ dominates in practical regimes. STAR-MD thus offers a favorable trade-off, enabling efficient modeling of long trajectories for large proteins without the cubic spatial overhead.

### 3.3 ADDITIONAL MODEL DETAILS

**Block-causal Attention for Efficient Training.** Traditional video diffusion models (Peebles & Xie, 2023; Ma et al., 2024) denoise all frames simultaneously, using a fixed window size. For autoregressive generation with variable context length, the model denoises one frame at a time, attending to clean history contexts. This is straightforward during inference, where we cache clean frames for efficient autoregressive generation. However, given the sequential nature of this process, extra work is needed for parallel training. To simultaneously denoise all tokens with clean history context, we follow Arriola et al. (2025); Teng et al. (2025); Deng et al. (2024) to concatenate clean and noisy frames as input sequence and employ a special block-wise attention pattern: all frames only attend to clean history frames, preserving the causal structure during training. At the cost of doubling the sequence length, the model can optimize the training loss for all frames in a single forward pass, aligning parallel teacher-forcing training with sequential autoregressive inference.

**Contextual Noise Perturbation for Robust Rollouts.** To mitigate compounding errors during long autoregressive rollouts, we apply a contextual noise perturbation technique inspired by Diffusion Forcing (Chen et al., 2024; Song et al., 2025). During training, we perturb historical context frames $\boldsymbol{x}_{<\ell}$ by applying the forward diffusion process (Eqs. 1-2) with a small, randomly sampled noise level $\tau \sim \mathcal{U}[0, 0.1]$ to obtain noisy contexts $\boldsymbol{x}_{<\ell}^{\tau}$. The model then learns to predict frame $\ell$ conditioned on this perturbed history. Critically, at inference time, we apply the same noise perturbation: after generating frame $\ell$, we add noise at level $\tau$ before using it as context for subsequent frames. This training-inference consistency ensures that the model experiences similar input distributions during deployment as during training, making it robust to its own prediction errors and preventing compounding drift in long rollouts.

**Continuous-time Conditioning.** To handle the vast range of timescales in protein motion, we draw the physical stride $\Delta t \sim \text{LogUniform}[10^{-2}, 10^1]$ ns independently for every training example. By conditioning the network on $\Delta t$ through adaptive layernorm (AdaLN, Peebles & Xie (2023)), the model learns to modulate its internal activations as a function of both diffusion progress and physical timescale. Crucially, this training strategy decouples the physical duration of a trajectory from the number of frames in the context window. Even when restricted to small context windows during training (as most methods are), randomly sampling large strides exposes the model to relative time deltas spanning orders of magnitude. This allows the network to learn long-range temporal dependencies without the computational cost of training on long sequences or the need for complex context-length extrapolation techniques.

### 3.4 THEORETICAL JUSTIFICATION FOR STAR-MD ARCHITECTURE

In this subsection, we use the Mori-Zwanzig formalism (Mori, 1965; Zwanzig, 1961) (See Appendix A for more details) to provide theoretical justification for two key aspects of our architecture: the necessity of temporal history in coarse-grained modeling, and the specific requirement for *joint* spatio-temporal attention arising from the removal of explicit pairwise features.

Atomistic molecular dynamics simulations evolve according to Hamiltonian mechanics and are Markovian in the full phase space of atomic positions and momenta. However, practical generative models must operate on coarse-grained representations, such as per-residue coordinates for protein structures. The Mori-Zwanzig formalism shows that projecting Markovian dynamics onto coarse-grained variables yields a Generalized Langevin Equation with three terms (Eq. (5)): a Markovian component, a memory kernel encoding history dependence, and a stochastic force representing eliminated degrees of freedom. Consequently, accurate modeling of coarse-grained protein dynamics requires non-Markovian models that incorporate temporal history to capture memory effects arising from eliminated degrees of freedom. Our work makes this connection explicit, providing the theoretical justification underlying recent temporal architectures in trajectory modeling (Jing et al., 2024b; Cheng et al., 2025; Shen et al., 2025).

Existing models typically rely on explicit pairwise features to capture spatial structure, but this incurs prohibitive $\mathcal{O}(L^2 N^2 + LN^3)$ or $\mathcal{O}(LN^3)$ costs (see Section 3.2). STAR-MD circumvents this by projecting out pairwise features, a design choice we analyze through our *Memory Inflation* proposition (Proposition 1, Appendix A). We show that removing explicit spatial correlations "inflates" the memory kernel for the remaining residues, necessitating a significantly richer temporal history to compensate. Crucially, this inflated kernel exhibits non-separable spatio-temporal

coupling (Remark 2), meaning spatial and temporal modes cannot be factorized. This theoretical insight directly motivates our architecture: instead of the interleaved spatial and temporal blocks used in prior work, STAR-MD employs *joint* spatio-temporal attention to approximate this complex, non-separable memory kernel, balancing physical fidelity with computational scalability.

## 4 EXPERIMENTS

We conduct a comprehensive set of experiments to evaluate STAR-MD's ability to generate long-horizon protein dynamics. First, we benchmark STAR-MD against state-of-the-art models on the standard 100 ns trajectory generation task (Section 4.2). Next, we assess its extrapolation capabilities by generating trajectories over longer, microsecond-scale horizons not seen during training (Section 4.3). Finally, we perform a series of analyses and ablations to investigate model stability, scalability, and the contributions of our key design choices (Section 4.4).

### 4.1 EXPERIMENTAL SETUP

**Dataset.** ATLAS dataset (Vander Meersche et al., 2024) contains 100 ns MD trajectories for 1390 structurally diverse proteins. We use the standard train/val/test splits from prior works (Jing et al., 2024b; Shen et al., 2025) to assess model performance in a transferable setting. To evaluate long-horizon generation–a key focus of our work—we extend the benchmark by running new MD simulations to produce 250 ns and 1 µs trajectories for selected proteins, using the original ATLAS simulation protocols for consistency. Further details are provided in Appendix C.

**Implementations.** We follow the training procedure described in Section 3.3 and generate trajectories using the configurations listed in Table 4. We compare STAR-MD with three state-of-the-art trajectory generative models trained on ATLAS: AlphaFolding (Cheng et al., 2025), MDGen (Jing et al., 2024b), and ConfRover (Shen et al., 2025). All models parametrize proteins using $SE(3)$ backbone rigids (translations and rotations) with torsional angles for side-chain atoms. To standardize evaluation, trajectories from variable-stride models (STAR-MD, ConfRover) are generated directly at the required intervals. Trajectories from fixed-stride models (AlphaFolding, MDGen) are first generated at their native resolution and then subsampled. Finally, we include an oracle reference based on MD simulations run independently to represent the target performance. All trajectories are aligned to the starting frame of the reference ATLAS trajectory via C$\alpha$ superposition prior to analysis (Appendix D). Specific parameters for each benchmark are detailed in the relevant sections below, with full implementation details in Appendix F.

- **Structural Quality:** We assess the physical plausibility of generated conformations using a hierarchy of geometric and stereochemical metrics. First, we perform coarse-grained checks for C$\alpha$-C$\alpha$ clashes (non-bonded atoms too close) and chain breaks (consecutive C$\alpha$ atoms too far apart). Second, we use the MolProbity suite (Chen et al., 2010) for fine-grained, all-atom analysis of backbone Ramachandran and side-chain rotamer outliers. We report three distinct validity metrics based on different criteria: **C$\alpha$-level Validity**, passing only C$\alpha$ checks; **All-Atom Validity**, passing only MolProbity checks; and **Combined Validity**, passing all checks simultaneously. See Appendix D for details on the thresholds.

- **Conformational Coverage:** To evaluate how well generated trajectories explore the conformational space of the reference MD simulation, we follow the protocol of Shen et al. (2025). We project all conformations into a low-dimensional space defined by the principal components of the reference trajectory and compute the Jensen-Shannon Divergence (JSD) and conformation recall between the distributions. To ensure that coverage reflects physically plausible exploration, we report these metrics computed exclusively on structurally valid conformations ("Cov Valid").

- **Dynamic Fidelity:** We assess temporal coherence using four metrics. First, we use **tICA lag-time correlation** to quantify the preservation of slow collective variables (Molgedey & Schuster, 1994; Pérez-Hernández et al., 2013; Shen et al., 2025); crucially, this is computed only on valid transitions to avoid artifacts from broken structures. Additionally, we evaluate **RMSD** to measure the magnitude of structural change over varying intervals, **autocorrelation** to assess temporal memory, and **VAMP-2 score** to evaluate how well slow dynamical modes of the system are captured. For last three metrics, we report the deviation from MD reference, with details in Appendix D.

### 4.2 STANDARD BENCHMARK: 100 NS TRAJECTORY GENERATION

We first evaluate STAR-MD on the standard 100 ns ATLAS benchmark. Following the protocol of Shen et al. (2025), we generate trajectories of 80 frames at a 1.2 ns interval. Results are summarized

Table 1: Quantitative results on 100 ns trajectory generation (ATLAS test set). STAR-MD achieves state-of-the-art performance across all metrics, with particularly significant improvements in conformational coverage (JSD, F1) and dynamic fidelity (tICA, average difference of RMSD, autocorrelation and VAMP-2 score to MD reference).

| | Cov Valid | | Dynamic Fidelity | | | | Validity | | |
|---|---|---|---|---|---|---|---|---|---|
| Model | JSD↓ | Rec↑ | tICA↑ | RMSD↓ | AutoCor↓ | VAMP-2↓ | CA%↑ | AA%↑ | CA+AA%↑ |
| MD (Oracle) | 0.31 | 0.67 | 0.17 | 0.00 | 0.00 | 0.02 | 98.37 | 98.07 | 96.43 |
| MDGen | $0.56_{\pm0.01}$ | $0.28_{\pm0.01}$ | $0.12_{\pm0.00}$ | $0.38 \pm 0.01$ | $\underline{0.05} \pm 0.00$ | $0.38 \pm 0.01$ | $71.83_{\pm1.90}$ | $95.03_{\pm0.59}$ | $68.31_{\pm2.20}$ |
| AlphaFolding* | $0.59_{\pm0.01}$ | $0.20_{\pm0.01}$ | N/A | $3.31 \pm 0.06$ | $0.12 \pm 0.01$ | $1.56 \pm 0.01$ | $10.58_{\pm0.09}$ | $0.82_{\pm0.10}$ | $0.47_{\pm0.04}$ |
| ConfRover | $\underline{0.52}_{\pm0.01}$ | $\underline{0.36}_{\pm0.01}$ | $\underline{0.15}_{\pm0.01}$ | $\underline{0.20} \pm 0.00$ | $0.08 \pm 0.00$ | $\underline{0.47} \pm 0.02$ | $56.94_{\pm0.52}$ | $92.47_{\pm0.25}$ | $52.06_{\pm0.36}$ |
| STAR-MD | $\mathbf{0.43}_{\pm0.01}$ | $\mathbf{0.54}_{\pm0.01}$ | $\mathbf{0.17}_{\pm0.00}$ | $\mathbf{0.07} \pm 0.02$ | $\mathbf{0.02} \pm 0.00$ | $\mathbf{0.10} \pm 0.02$ | $\mathbf{86.81}_{\pm0.64}$ | $\mathbf{98.18}_{\pm0.05}$ | $\mathbf{85.29}_{\pm0.62}$ |

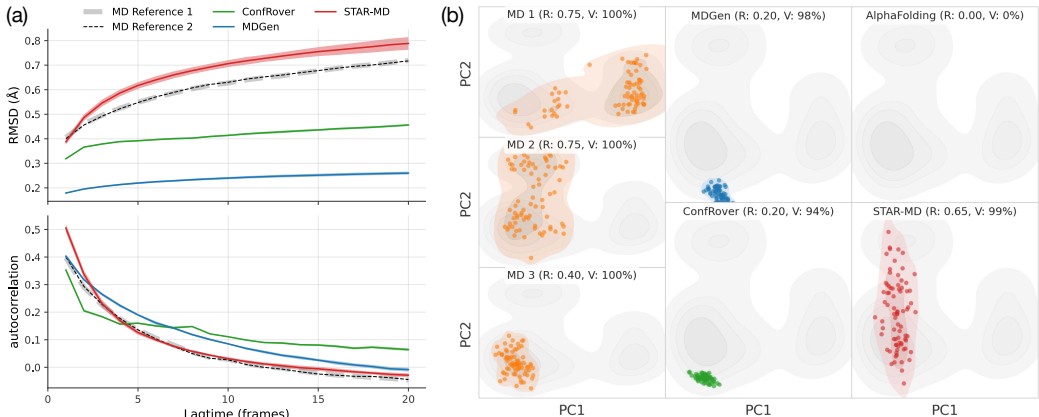

Figure 2: **Kinetic fidelity and conformational coverage on the ATLAS 100 ns benchmark. (a)** Comparing C$\alpha$ coordinate RMSD (top) and autocorrelation (bottom) at varying lagtime for different models. STAR-MD better captures the overall trend and characteristic magnitudes, similar to the MD reference runs (dashed lines). Shaded bands represent ±1 standard deviation. The small size of the shaded bands demonstrate the robustness of this metric. **(b)** Conformational coverage comparison for `6XB3-H` for all models and 3 MD simulations. Generated trajectories are projected onto the first two principal components (PCs) of the reference MD simulation (gray contours). Only structurally valid frames are considered for this plot. Recall and Validity are reported for each run. Baseline methods (MDGen, ConfRover) exhibit limited diversity, becoming confined to a small region of the conformational landscape. AlphaFolding's generated trajectories consist of all structurally implausible frames, with a validity of 0%. In contrast, STAR-MD demonstrates significantly broader exploration (with a recall value of 0.65), visiting two of the major modes observed in the MD reference, matching the diversity seen in independent MD runs.

in Table 1. It is important to note that due to its high computational cost, AlphaFolding failed to run on the four largest proteins in the test set, highlighting its scalability limitations even on this standard benchmark. On the full test set, STAR-MD, achieves a superior balance of conformational coverage, dynamic fidelity, and structural quality.

The results demonstrate clear performance deficiencies in all baseline models. Baselines like MDGen and ConfRover exhibit low conformational coverage (0.30 and 0.42 Recall, respectively) and produce a substantial number of structurally invalid frames, with validity rates of only 64.9% and 49.7%, respectively. AlphaFolding achieves a higher raw recall (0.51), but this metric is misleading as a negligible 0.11% of its generated frames are structurally valid; when controlled for valid structures, this recall drops to 0.01, rendering AlphaFolding as a very unreliable model for this task. In contrast, STAR-MD achieves the highest conformational coverage (0.58 recall) and structural validity (86.36%) of any generative model, significantly narrowing the performance gap to the ground-truth MD simulation which serves as the oracle for this task.

In Figure 2(a) and 8, we compare the level of conformation changes at different lag times, as an indicator how well the model captures the ground truth motion changes. STAR-MD shows improved coherence with MD reference trajectories, while AlphaFolding significantly overestimates the motion and MDGen and ConfRover tend to underestimate the dynamics level.

---

*AlphaFolding results are evaluated on 78/82 proteins due to out-of-memory error for 4 large proteins.

Table 2: Results on molecular dynamics trajectory generation at 240 ns and 1 μs timescales. STAR-MD demonstrates competitive performance across different temporal scales, with particularly strong quality metrics at both timescales.

| Model | Time | Cov Valid | | Dynamic Fidelity | | Validity | | |
| | | JSD↓ | Rec↑ | RMSD↓ | AutoCor↓ | CA%↑ | AA%↑ | CA+AA%↑ |
|---|---|---|---|---|---|---|---|---|
| MD (Oracle) | 240 ns | 0.26 | 0.75 | 0.01 | 0.00 | 99.53 | 96.83 | 96.36 |
| | 1 μs | 0.23 | 0.91 | 0.00 | 0.00 | 96.25 | 86.50 | 82.75 |
| MDGen | 240 ns | $0.52_{\pm0.01}$ | $0.38_{\pm0.01}$ | $0.48_{\pm0.01}$ | $0.25_{\pm0.01}$ | $63.25_{\pm2.10}$ | $87.83_{\pm1.13}$ | $56.60_{\pm2.10}$ |
| | 1 μs | $0.56_{\pm0.01}$ | $0.36_{\pm0.03}$ | $0.37_{\pm0.02}$ | $0.39_{\pm0.01}$ | $36.11_{\pm7.34}$ | $56.99_{\pm4.52}$ | $24.81_{\pm4.30}$ |
| Alphafolding | 240 ns | $0.57_{\pm0.03}$ | $0.20_{\pm0.03}$ | $1.76_{\pm0.05}$ | $\underline{0.14_{\pm0.01}}$ | $8.96_{\pm0.25}$ | $0.94_{\pm0.12}$ | $0.63_{\pm0.16}$ |
| | 1 μs | $0.65_{\pm0.00}$ | $0.20_{\pm0.00}$ | $0.78_{\pm0.03}$ | $\mathbf{0.04_{\pm0.01}}$ | $9.64_{\pm0.02}$ | $0.19_{\pm0.00}$ | $0.06_{\pm0.00}$ |
| ConfRover-W | 240 ns | $\underline{0.51_{\pm0.01}}$ | $\underline{0.42_{\pm0.02}}$ | $\underline{0.35_{\pm0.01}}$ | $0.39_{\pm0.01}$ | $44.71_{\pm1.55}$ | $73.13_{\pm0.84}$ | $36.51_{\pm1.22}$ |
| | 1 μs | $\underline{0.55_{\pm0.02}}$ | $\underline{0.45_{\pm0.02}}$ | $\underline{0.33_{\pm0.02}}$ | $0.38_{\pm0.03}$ | $54.74_{\pm1.79}$ | $62.32_{\pm3.43}$ | $36.91_{\pm1.39}$ |
| STAR-MD | 240 ns | $\mathbf{0.44_{\pm0.01}}$ | $\mathbf{0.59_{\pm0.01}}$ | $\mathbf{0.20_{\pm0.02}}$ | $\mathbf{0.03_{\pm0.01}}$ | $\mathbf{85.16_{\pm1.91}}$ | $\mathbf{97.57_{\pm0.13}}$ | $\mathbf{83.15_{\pm1.99}}$ |
| | 1 μs | $\mathbf{0.46_{\pm0.01}}$ | $\mathbf{0.61_{\pm0.02}}$ | $\mathbf{0.13_{\pm0.02}}$ | $\underline{0.10_{\pm0.02}}$ | $\mathbf{88.47_{\pm1.09}}$ | $\mathbf{89.81_{\pm0.65}}$ | $\mathbf{79.93_{\pm1.04}}$ |

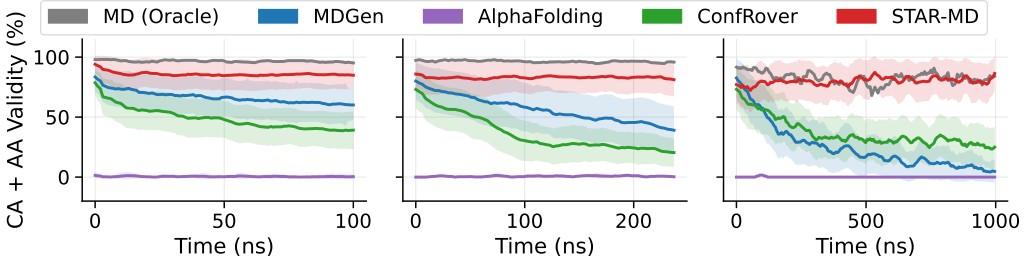

Figure 3: **Long-horizon stability and error accumulation across different time scales.** We plot the structural validity percentage over time for trajectories generated by STAR-MD and baseline models, evaluated at 100 ns (left), 240 ns (middle), and 1 μs (right) horizons. Shaded bands represent ±1 standard deviation across 5 different repeats. While most models exhibit clear error accumulation over simulation time, STAR-MD maintains high structural validity, regardless of the simulation time scale.

Finally, Figure 2(b) provides a qualitative assessment of conformational coverage using the principal component projection for an example ATLAS 100 ns protein. STAR-MD exhibits better significantly better exploration and diversity than comparison methods and is able to populate two distinct modes.

## 4.3 Extrapolating for Long-Horizon Trajectories

The ability to generate stable, physically realistic trajectories over extended time horizons is critical for modeling functionally relevant protein dynamics. Most conformational transitions of biological interest occur beyond the 100 ns timescale, where existing generative models begin to fail. We assess STAR-MD's long-horizon capability through two increasingly challenging settings: 240 ns simulation with 32 proteins and 1 μs simulation with 8 proteins, with all proteins unseen during training. To rigorously evaluate temporal extrapolation, all models were trained exclusively on the 100 ns ATLAS trajectories and received no fine-tuning for longer horizons. This setup directly tests the ability to generalize to dynamics far beyond the training data distribution. Since the ATLAS benchmark only provides trajectories up to 100 ns, we generated our own molecular dynamics simulations for longer timescales to create proper reference data for evaluation. These extended simulations follow the same simulation protocols as ATLAS but continue the dynamics to longer timescales.

Due to its scaling issues, ConfRover could not be evaluated on either 240 ns or 1 μs generation tasks. This is because ConfRover performs temporal attention over pair features, requiring previous frames' pair features to be stored in memory as KV cache (see Appendix B for more detailed analysis). As such, even with CPU offloading of the KV cache, ConfRover's memory requirements exceeded our hardware limits (1869 GB CPU RAM, 8× H100 GPUs with 80GB VRAM each). As a result and for fair comparison, we utilize a variant of ConfRover with windowed attention with attention sink tokens (as described in Xiao et al. (2023b)) to reduce memory usage. We report results for this windowed variant, labeled ConfRover-W, in Table 2.

Table 2 summarizes the results for these two extended timescales. Remarkably, STAR-MD maintains stable and competitive performance at both the 240 ns timescale and the challenging 1 μs

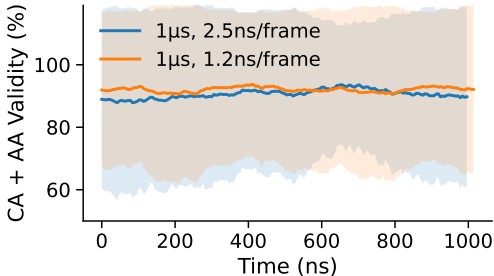 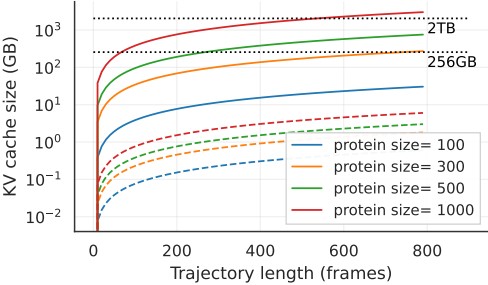

Figure 4: **Stability of STAR-MD across different temporal strides for 1 µs generation.** We plot the structural validity over time for two 1 µs trajectories generated with different strides: 2.5 ns/frame (400 steps) and a more challenging 1.2 ns/frame (∼833 steps). Solid lines show mean validity, while shaded bands represent ±1 standard deviation across test proteins. Our models remain stable and maintain high structural quality even when generating much longer sequences of frames than seen in training.

Figure 5: **Memory footprint of KV caches** with varying protein sizes and trajectory lengths. ConfRover is shown in solid lines and STAR-MD is shown in dashed lines. Both models contain 8 attention layers with hidden dimension of 128 (ConfRover) or 256 (STAR-MD). STAR-MD show much smaller memory cost compared to ConfRover. Horizontal lines marks the 256 GB and 2 TB memory caps.

timescale, demonstrating the effectiveness of our joint spatio-temporal attention and contextual noise techniques for long-horizon stability. Further, STAR-MD exhibits controlled error accumulation, with low clash and break rates that remain consistent throughout the extended trajectory (Figure 3 (right), Section 4.4). Among the methods achieving reasonable structural quality, STAR-MD attains the best balance of coverage and fidelity, nearing the oracle MD performance. In both the 240 ns and 1 µs case, AlphaFolding produces physically implausible, high-error structures despite good coverage metrics, while ConfRover-W and MDGen degrade significantly over the microsecond horizon. We further report kinetic metrics for the long trajectory cases in Fig. 8, where STAR-MD shows the best alignment with the RMSD and autocorrelation of the MD reference from among baseline methods.

### 4.4 ADDITIONAL ANALYSIS

**Error Accumulation.** A key challenge in long-horizon generation is error accumulation, where minor inaccuracies compound over time, leading to trajectory degradation. Figure 3 plots the average validity percentage over simulation time for the 100 ns, 240 ns, and 1 µs simulations. As the simulation horizon extends, baseline models exhibit rapidly deteriorating performance, with structural validity declining sharply. In contrast, STAR-MD remains stable, maintaining high structural validity close to the MD oracle across all timescales. This demonstrates our model's robustness against error accumulation, a quality largely attributable to the historical-context noise mechanism, which enables stable long-horizon autoregressive generation.

**Varying temporal resolution for long-horizon generation.** The 1 µs generation task above uses a stride of 2.5 ns, resulting in 400 frames. Thanks to our continuous-time conditioning, we can generate at different temporal resolutions without retraining. We test this by generating a 1 µs trajectory with a 1.2 ns stride, nearly doubling the number of frames to ∼850. Figure 4 shows that STAR-MD remains stable even with this much longer sequence of frames, maintaining high structural quality. This further underscores our model's robustness to different sampling intervals and trajectory lengths, showing strong capability for stable, long-horizon autoregressive generation.

**KV Cache Analysis.** A critical limitation of ConfRover is the need to maintain the KV-cache of single and pair embedding for temporal attentions. In contrast, STAR-MD only requires maintaining the KV cache for singles. By only maintaining attentions among single representations, STAR-MD has a magnitudes lower memory footprints (see Fig. 5 for a comparison). This advantage allows our model to maintain a full KV caches in GPU memory without resorting to CPU offloading (e.g, additional overhead) or sliding window style KV caches (compromises temporal history).

### 4.5 ABLATION STUDIES ON KEY COMPONENTS

To validate our key design choices, we conduct systematic ablations of the main components of STAR-MD. Table 3 summarizes the quantitative results on the 100 ns benchmark. Appendix H.2 contains full ablation tables for the 100 ns, 240 ns, and 1 µs benchmarks.

**S×T attention vs. separable attention.** We compare our joint S×T attention with a separable alternative that processes spatial and temporal dimensions sequentially. The separable model suffers a significant drop in conformational coverage and a modest drop in dynamic fidelity. This confirms that joint S×T attention is crucial for capturing the complex, non-separable spatio-temporal dependencies inherent in protein dynamics, which directly impacts the model's ability to explore the correct conformational space.

**S×T layer placement.** We evaluated placing the spatio-temporal attention module outside the diffusion decoder as a static conditioner which compresses the full trajectory history into a single conditioning vector, mirroring Shen et al. (2025). This variant underperforms our full model in both coverage and fidelity, confirming that integrating spatio-temporal attention directly within the diffusion module is essential for effective context utilization.

**Historical-context noise.** Removing the contextual noise perturbation leads to a substantial drop in structural quality. This highlights the importance of this technique for maintaining generation stability, a finding that is further supported by our long-horizon experiments in Section 4.3. Figure 12 compares the CA+AA validity over time for models with and without contextual noise on the 240 ns and 1 μs benchmarks, showing that the noise helps maintain structural quality over long horizons.

These ablation studies demonstrate that each component of STAR-MD addresses a specific challenge in protein dynamics modeling. The S×T attention provides the expressivity needed for complex spatio-temporal dependencies. The historical-context noise ensures stable long-horizon generation by preventing error accumulation.

Table 3: **Ablation study on 100 ns simulation.** The full STAR-MD model is compared against variants with key components removed (no contextual noise, separate spatial and temporal attention, and placing S×T attention outside the diffusion block). These modifications affect model coverage and conformation quality. Metrics degraded in each ablation setting are highlighted in red. See complete ablation results on longer trajectories in Appendix H.2.

| Model | Cov Valid | | Dynamic Fidelity | | | | Validity | | |
|---|---|---|---|---|---|---|---|---|---|
| | JSD↓ | Rec↑ | tICA↑ | RMSD↓ | AutoCor↓ | VAMP-2↓ | CA%↑ | AA%↑ | CA+AA%↑ |
| STAR-MD | 0.42 | 0.57 | 0.17 | 0.09 | 0.02 | 0.12 | 86.62 | 98.28 | 85.18 |
| w/o Noise | 0.43 | 0.54 | 0.18 | 0.04 | 0.11 | 1.22 | 77.82 | 97.76 | 76.12 |
| w/ Sep Attn | 0.46 | 0.46 | 0.17 | 0.09 | 0.05 | 0.49 | 87.95 | 98.34 | 86.70 |
| w/ Preproc Attn | 0.47 | 0.48 | 0.17 | 0.09 | 0.05 | 0.54 | 84.55 | 97.81 | 82.56 |

## 5 CONCLUSION

We introduced STAR-MD, a novel causal diffusion transformer model for generating long-horizon protein dynamics trajectories. STAR-MD addresses the challenges of long-horizon generation through several key innovations: joint spatio-temporal attention that efficiently models complex space-time couplings, continuous-time conditioning for generation at arbitrary timescales, and a noisy-context training scheme that mitigates error accumulation. Our comprehensive evaluations demonstrate that STAR-MD achieves new state-of-the-art performance on the standard 100 ns AT-LAS benchmark, outperforming previous methods in conformational coverage, structural quality, and kinetic fidelity. More importantly, our model successfully extrapolates to generate stable, high-fidelity trajectories up to the microsecond regime, where prior models often fail due to computational costs or compounding errors. By balancing expressiveness and scalability, STAR-MD establishes an efficient and empirically strong foundation for modeling protein dynamics at biologically relevant scales, paving the way for accelerated exploration of complex biological processes.

**Limitations and future work.** While STAR-MD represents a significant step forward, there are avenues for future improvement. The temporal consistency of the generated trajectories, while strong, does not yet perfectly match that of oracle MD simulations. The model's performance could be further enhanced by training on larger and more diverse MD simulation datasets, such as MD-CATH (Mirarchi et al., 2024). Additionally, a promising direction for future work is to extend the model's capabilities to simulate the dynamics of protein complexes or their interactions with small molecules, which are crucial for understanding biological processes and drug design.

## AUTHOR CONTRIBUTIONS AND ACKNOWLEDGMENTS

**Author Contributions.** Y.S. and Q.G. conceived the study, designed the experimental protocols, and supervised methodology development. N.S., Y.L., and Y.S. developed the spatiotemporal causal diffusion architecture and analyzed model performance. P.L. and Y.S. conceptualized the contextual noise perturbation, which N.S. implemented. N.S., P.L. and R.B. conducted the theoretical verification of the Mori-Zwanzig formalism and validated the mathematical frameworks. All authors participated in the manuscript preparation and critical review for publication.

**Acknowledgments.** We would like to thank Chan Lu, Cheng-Yen (Wesley) Hsieh, Yi Zhou, Zaixiang Zheng, Yilai Li and other members of ByteDance for their valuable feedback and suggestions that helped improve this work.

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

CONTENTS

# Theoretical Analysis

## A    THEORETICAL FOUNDATION FOR LEARNING COARSE-GRAINED PROTEIN DYNAMICS

### A.1    OVERVIEW AND MOTIVATION

Modeling protein dynamics in solution requires a complete description of the system (e.g., coordinates and momenta of all protein and solvent atoms), making it computationally prohibitive for current generative models. Any practical implementation must adopt coarse-grained representations, such as per-residue coordinates solely on protein structures. The Mori-Zwanzig formalism (Mori, 1965; Zwanzig, 1961) shows that projecting full-atom dynamics into a reduced subspace (as done by our coarse-graining) introduces memory effects, rendering the dynamics non-Markovian and requiring an explicit **memory kernel** to preserve the original dynamics in this coarse-grained space. For protein structural learning, the reduction from singles-and-pairs representations to singles-only representations introduces a complex spatio-temporal-coupling memory kernel. Therefore, to remain scalable without the overhead from expensive pair representations, a model *must* employ an architecture capable of learning complex, non-separable spatio-temporal dependencies. This analysis motivates our departure from prior "space-then-time" factorized models to a joint spatio-temporal attention mechanism.

Following sections formalize the effects of coarse-graining on protein dynamic learning: We introduce Mori-Zwanzig formalism in Appendix A.2.1, its specialized form in protein structural representations in Appendix A.2.2, and the implications for neural architecture design in Appendix A.3.

### A.2    FORMAL DERIVATION OF MEMORY INFLATION

#### A.2.1    BACKGROUND: MORI-ZWANZIG FORMALISM

The Mori-Zwanzig formalism (Mori, 1965; Zwanzig, 1961) provides a mathematical framework for deriving reduced dynamical models from high-dimensional systems. We begin with a high-dimensional system described by variables $\Gamma(t)$ evolving according to:

$$\frac{d\Gamma}{dt} = \mathcal{L}\Gamma(t) \tag{4}$$

where $\mathcal{L}$ is the Liouville operator. When we project onto a lower-dimensional subspace of variables $A(t)$ using a linear projection operator $\mathcal{P}^*$, the Mori-Zwanzig theorem provides the exact generalized Langevin equation (GLE) for their evolution:

$$\frac{dA(t)}{dt} = \mathcal{P}\mathcal{L}A(t) + \int_0^t K(t - \tau)A(\tau)d\tau + F(t) \tag{5}$$

where:

- $A(t)$ represents our chosen variables (the projection of $\Gamma(t)$).

- $\mathcal{P}\mathcal{L}A(t)$ captures Markovian (instantaneous) dynamics.

- $\int_0^t K(t - \tau)A(\tau)d\tau$ is the memory term encoding history dependence and $K(t) = \mathcal{P}\mathcal{L}e^{(1-\mathcal{P})\mathcal{L}t}(1 - \mathcal{P})\mathcal{L}$ is the memory kernel, which encodes the information over the degrees of freedom that are not explicitly modeled.

- $F(t)$ is the random force, a rapidly-fluctuating noise term arising from degrees of freedom in the $(1 - \mathcal{P})$ subspace.

---

*In the context of the Mori-Zwanzig formalism, $\mathcal{P}$ is a linear operator that projects onto a chosen subspace of the full phase space.

Accounting for the memory term induced by coarse-graining is crucial to correctly capture the dynamics of the original system.

### A.2.2 PROTEIN DYNAMICS PROJECTIONS

In deep learning models for proteins, per-residue features (singles) and their pairwise representations (pairs) are commonly adopted to model their structures ((Jumper et al., 2021; Abramson et al., 2024; Jing et al., 2024a; Shen et al., 2025; Chen et al., 2024) Here, we consider two specific projections:

**Singles-and-Pairs Representation**  We project onto both per-residue features $s(t) \in \mathbb{R}^{N \times d_s}$ and pairwise features $z(t) \in \mathbb{R}^{N \times N \times d_z}$, yielding state $A^{(1)}(t) = (s(t), z(t))$. The resulting dynamics can be written in block form:

$$
\begin{pmatrix} \dot{s}(t) \\ \dot{z}(t) \end{pmatrix} = \begin{pmatrix} \Omega_{ss} & \Omega_{sz} \\ \Omega_{zs} & \Omega_{zz} \end{pmatrix} \begin{pmatrix} s(t) \\ z(t) \end{pmatrix} + \int_0^t \begin{pmatrix} K_{ss}^{(1)}(t-\tau) & K_{sz}^{(1)}(t-\tau) \\ K_{zs}^{(1)}(t-\tau) & K_{zz}^{(1)}(t-\tau) \end{pmatrix} \begin{pmatrix} s(\tau) \\ z(\tau) \end{pmatrix} d\tau + \begin{pmatrix} F_s(t) \\ F_z(t) \end{pmatrix} \quad (6)
$$

where the memory kernel $K^{(1)}$ has block structure corresponding to singles-singles ($K_{ss}^{(1)}$), singles-pairs ($K_{sz}^{(1)}$), pairs-singles ($K_{zs}^{(1)}$), and pairs-pairs ($K_{zz}^{(1)}$) interactions.

**Singles-Only Representation**  We project directly onto per-residue features $s(t)$ alone, yielding the more compact state $A^{(2)}(t) = s(t)$. The dynamics become:

$$
\dot{s}(t) = \Omega^{(2)} s(t) + \int_0^t K^{(2)}(t-\tau) s(\tau) d\tau + F^{(2)}(t) \quad (7)
$$

### A.2.3 MEMORY INFLATION ON SINGLES-ONLY REPRESENTATIONS

**Proposition 1.** *[Memory Inflation] Under linearization, the memory kernel $\tilde{K}^{(2)}$ for the singles-only representation relates to the singles-and-pairs memory kernel $\tilde{K}^{(1)}$ by:*

$$
\tilde{K}^{(2)}(p) = \tilde{K}_{ss}^{(1)}(p) + \underbrace{(\Omega_{sz} + \tilde{K}_{sz}^{(1)}(p))\left[pI - \Omega_{zz} - \tilde{K}_{zz}^{(1)}(p)\right]^{-1}(\Omega_{zs} + \tilde{K}_{zs}^{(1)}(p))}_{\text{Inflation Term}}, \quad (8)
$$

*where $\tilde{K}(p)$ is the Laplace transform of the memory kernel, $p$ is the Laplace variable, $I$ is the identity matrix, $\Omega$ represents instantaneous (Markovian) dynamics, and the subscripts $s$ and $z$ refer to singles and pairs features, respectively.*

*Proof.* We begin by taking the Laplace transform of the singles-and-pairs dynamics:

$$
\begin{pmatrix} p\tilde{s}(p) - s(0) \\ p\tilde{z}(p) - z(0) \end{pmatrix} = \begin{pmatrix} \Omega_{ss} & \Omega_{sz} \\ \Omega_{zs} & \Omega_{zz} \end{pmatrix} \begin{pmatrix} \tilde{s}(p) \\ \tilde{z}(p) \end{pmatrix} + \begin{pmatrix} \tilde{K}_{ss}^{(1)}(p)\tilde{s}(p) & \tilde{K}_{sz}^{(1)}(p)\tilde{z}(p) \\ \tilde{K}_{zs}^{(1)}(p)\tilde{s}(p) & \tilde{K}_{zz}^{(1)}(p)\tilde{z}(p) \end{pmatrix} + \begin{pmatrix} \tilde{F}_s(p) \\ \tilde{F}_z(p) \end{pmatrix} \quad (9)
$$

From the second row, we can express $\tilde{z}(p)$ in terms of $\tilde{s}(p)$:

$$
\tilde{z}(p) = \left[pI - \Omega_{zz} - \tilde{K}_{zz}^{(1)}(p)\right]^{-1}\left[z(0) + (\Omega_{zs} + \tilde{K}_{zs}^{(1)}(p))\tilde{s}(p) + \tilde{F}_z(p)\right] \quad (10)
$$

Substituting this into the first row equation:

$$
p\tilde{s}(p) - s(0) = \Omega_{ss}\tilde{s}(p) + \tilde{K}_{ss}^{(1)}(p)\tilde{s}(p) + \tilde{F}_s(p) + \quad (11)
$$

$$
(\Omega_{sz} + \tilde{K}_{sz}^{(1)}(p))\left[pI - \Omega_{zz} - \tilde{K}_{zz}^{(1)}(p)\right]^{-1}\left[z(0) + (\Omega_{zs} + \tilde{K}_{zs}^{(1)}(p))\tilde{s}(p) + \tilde{F}_z(p)\right] \quad (12)
$$

Collecting terms with $\tilde{s}(p)$ and comparing with the Laplace transform of the singles-only representation:

$$p\tilde{s}(p) - s(0) = \Omega^{(2)}\tilde{s}(p) + \tilde{K}^{(2)}(p)\tilde{s}(p) + \tilde{F}^{(2)}(p) \tag{13}$$

We identify:

$$\Omega^{(2)} = \Omega_{ss} \tag{14}$$

$$\tilde{K}^{(2)}(p) = \tilde{K}^{(1)}_{ss}(p) + (\Omega_{sz} + \tilde{K}^{(1)}_{sz}(p))\left[pI - \Omega_{zz} - \tilde{K}^{(1)}_{zz}(p)\right]^{-1}(\Omega_{zs} + \tilde{K}^{(1)}_{zs}(p)) \tag{15}$$

$$\tilde{F}^{(2)}(p) = \tilde{F}_s(p) + (\Omega_{sz} + \tilde{K}^{(1)}_{sz}(p))\left[pI - \Omega_{zz} - \tilde{K}^{(1)}_{zz}(p)\right]^{-1}\left[z(0) + \tilde{F}_z(p)\right] \tag{16}$$

This completes the proof, showing that the memory kernel in the singles-only representation incorporates additional terms that account for the eliminated pair dynamics. □

### A.3 IMPLICATIONS FOR ARCHITECTURE DESIGN

From the Memory Inflation result in Proposition 1, we can draw two important remarks that guide our architectural design:

#### A.3.1 MEMORY ENRICHMENT

**Remark 1** (Memory Enrichment). *When spatial detail is removed from a dynamical system, the memory kernel must become more expressive to preserve the system's physical accuracy.*

This follows directly from the inflation term in Proposition 1. By eliminating the explicit representation of pairwise features $z(t)$, we force the memory kernel $K^{(2)}$ to incorporate additional dynamics that were previously captured through the direct modeling of pairs. This represents a fundamental trade-off between:

1. **Spatial complexity**: Using $O(N^2)$ variables to explicitly represent all pairwise relationships

2. **Temporal complexity**: Using a richer memory structure that implicitly encodes these relationships through time

The theorem quantifies exactly how much additional memory structure is required: it must include all dynamical information that would have flowed through the pairs variables in the more complex representation.

#### A.3.2 SPATIO-TEMPORAL COUPLING

**Remark 2** (Spatio-Temporal Coupling). *The memory kernel in the singles-only representation cannot be factorized as independent spatial and temporal components, requiring models that capture non-separable coupling between spatial indices and temporal frequencies.*

The inflation term:

$$(\Omega_{sz} + \tilde{K}^{(1)}_{sz}(p))\left[pI - \Omega_{zz} - \tilde{K}^{(1)}_{zz}(p)\right]^{-1}(\Omega_{zs} + \tilde{K}^{(1)}_{zs}(p)) \tag{17}$$

has a critical property: it cannot be factorized as a product of purely spatial and purely temporal operators. To see this, consider its structure:

1. $(\Omega_{sz} + \tilde{K}^{(1)}_{sz}(p))$ couples spatial indices $(i, j)$ with temporal frequency $p$

2. The matrix inverse $\left[pI - \Omega_{zz} - \tilde{K}^{(1)}_{zz}(p)\right]^{-1}$ mixes these couplings in a non-separable way

3. The final term $(\Omega_{zs} + \tilde{K}^{(1)}_{zs}(p))$ further couples the result with additional spatial indices

In general, in the time domain, this corresponds to a memory kernel with structure:

$$K_{ij}^{(2)}(t - \tau) \neq u_{ij} v(t - \tau) \tag{18}$$

where $i, j$ are residue indices. This non-separability means that the memory kernel cannot be decomposed into independent spatial and temporal components—spatial relationships evolve with time, and temporal patterns differ across spatial relationships.

### A.3.3 CONNECTION TO MACHINE LEARNING ARCHITECTURES

These theoretical results directly inform architectural design for protein dynamics models:

1. **Factorized attention** (spatial then temporal, or vice versa) cannot capture the non-separable coupling revealed by Remark 2.
2. **Joint spatio-temporal attention** with tokens indexing both residue and time provides exactly the structure needed to learn the inflated memory kernel demanded by Remark 1.

This analysis demonstrates that architectures that model protein dynamics without explicit pairwise features require sophisticated temporal modeling capabilities. The mathematical structure of the memory kernel dictates the minimum expressivity requirements for any machine learning model that aims to capture the underlying physics accurately.

## B COMPLEXITY ANALYSIS OF SPATIO-TEMPORAL ATTENTION

To understand the computational complexity and scaling behavior of different protein dynamics models, we compare the the spatial and temporal operations in ConfRover and our STAR-MD approach.

### B.1 PROBLEM FORMULATION AND NOTATION

For a protein with $N$ residues and a trajectory with $L$ frames, we define:

- $s_\ell \in \mathbb{R}^{N \times d_s}$: Per-residue (singles) features at frame $\ell$
- $z_\ell \in \mathbb{R}^{N \times N \times d_z}$: Pairwise features at frame $\ell$

The full representation across all frames would be:

$$S = [s_1, s_2, \ldots, s_L] \in \mathbb{R}^{L \times N \times d_s} \tag{19}$$
$$Z = [z_1, z_2, \ldots, z_L] \in \mathbb{R}^{L \times N \times N \times d_z} \tag{20}$$

### B.2 COMPUTATIONAL COMPLEXITY OF DIFFERENT ATTENTION MECHANISMS

### B.2.1 FULL SPATIO-TEMPORAL ATTENTION (THEORETICAL)

The full spatio-temporal attention computes across all residues and all time frames for both singles and pairs, with computational complexity:

$$\mathcal{O}((L \times (N + N^2))^2) = \mathcal{O}(L^2 N^4) \tag{21}$$

This is computationally prohibitive for any realistic protein system.

### B.2.2 CONFROVER APPROACH: TWO SOURCES OF COMPUTATIONAL COMPLEXITY

ConfRover faces computational challenges from two distinct sources:

**1. Channel-Factorized Temporal Attention.** The primary limitation in ConfRover is its temporal attention mechanism. It applies temporal attention independently to each single feature and each pair feature:

- For singles: $N$ separate $L \times L$ temporal attention operations, complexity $\mathcal{O}(N \times L^2)$

- For pairs: $N^2$ separate $L \times L$ temporal attention operations, complexity $\mathcal{O}(N^2 \times L^2)$

Total temporal attention complexity: $\mathcal{O}(N \times L^2 + N^2 \times L^2) = \mathcal{O}((N + N^2) \times L^2)$

While this factorization makes computation manageable, it assumes that temporal dynamics can be modeled independently per channel (whether single or pair), which fundamentally limits the model's ability to capture complex spatio-temporal couplings that govern protein dynamics.

**2. Pairformer Operations for Spatial Modeling.** In addition to the temporal attention, ConfRover employs a Pairformer layer for spatial interactions at each time step, which has complexity:

- $\mathcal{O}(N^3)$ per frame due to the interaction between all residues and all pairs
- $\mathcal{O}(N^3 \times L)$ for the entire trajectory

The combined time complexity is therefore $\mathcal{O}((N + N^2) \times L^2 + N^3 \times L)$. For large proteins, the $N^3$ term dominates, making scaling to large systems prohibitive both in terms of computation time and memory usage.

### B.2.3 STAR-MD APPROACH: SINGLE-RESTRICTED SPATIO-TEMPORAL ATTENTION

Our approach restricts spatio-temporal attention to only single-residue features:

- We apply one $(N \times L) \times (N \times L)$ attention operation on the flattened singles tensor
- Computational complexity: $\mathcal{O}((N \times L)^2) = \mathcal{O}(N^2 L^2)$

Critically, we eliminate the Pairformer component and its $\mathcal{O}(N^3)$ complexity. Instead, we rely on the spatio-temporal attention mechanism to implicitly learn the necessary pairwise relationships through the Memory Inflation mechanism described in Section A.1.

### B.3 STAR-MD VS. FACTORIZED SPATIAL-TEMPORAL ARCHITECTURES

STAR-MD employs a joint spatio-temporal attention mechanism with complexity $\mathcal{O}(N^2 L^2)$. In contrast, some architectures utilize spatial Pairformer layers ($\mathcal{O}(N^3 L)$) and temporal attention layers on singles ($\mathcal{O}(NL^2)$), resulting in a total complexity of $\mathcal{O}(N^3 L + NL^2)$.

To identify the regime where each model is more efficient, we equate their complexities:
$$N^2 L^2 = N^3 L + NL^2 \tag{22}$$
Assuming $N, L > 0$, we can simplify this to find the crossover point:
$$L \approx N \quad (\text{specifically } L = \frac{N^2}{N - 1}) \tag{23}$$

This reveals two distinct regimes:

- **Regime 1** ($L \gtrsim N$)**:** When the context length exceeds the protein size (e.g., small peptides with very long history), the $\mathcal{O}(NL^2)$ temporal scaling of baselines is advantageous.
- **Regime 2** ($N \gg L$)**:** For realistic protein modeling, where system size $N$ is large (hundreds to thousands of residues) and context history $L$ is fixed (e.g., 50-100 frames), STAR-MD is significantly more efficient. In this regime, the baselines' cubic spatial scaling ($\mathcal{O}(N^3 L)$) becomes a prohibitive bottleneck.

As illustrated in Figure 6, for a standard context length of $L = 80$, STAR-MD's computational cost scales much more favorably with protein size than baseline methods, enabling the simulation of large protein complexes that are computationally intractable for $\mathcal{O}(N^3)$ methods.

### B.4 KV CACHE EFFICIENCY DURING INFERENCE

An important aspect to consider in autoregressive protein trajectory generation is the memory cost during inference, particularly related to the key-value (KV) cache used for efficient generation. This becomes especially important for long-horizon generation where hundreds or thousands of frames must be maintained in context.

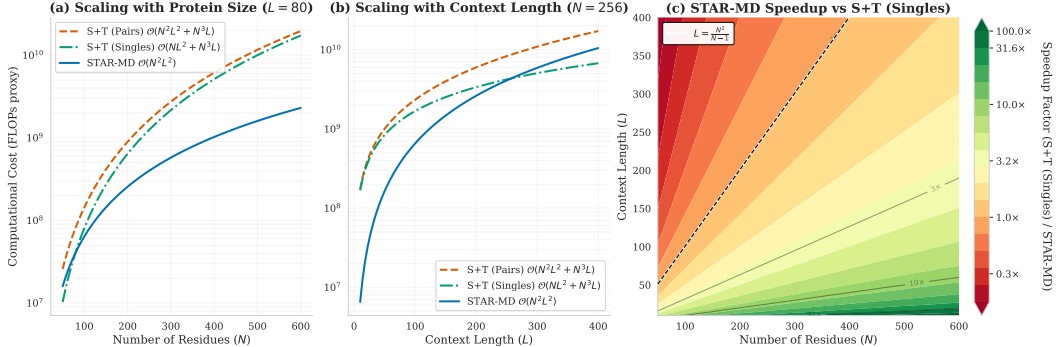

Figure 6: **Computational complexity scaling.** Comparison of computational cost (FLOPs proxy) for STAR-MD ($\mathcal{O}(N^2L^2)$), Pairformer + Temporal Attention on Pairs ($\mathcal{O}(N^2L^2 + N^3L)$), and Pairformer + Temporal Attention on Singles ($\mathcal{O}(NL^2 + N^3L)$). **Left:** Scaling with protein size $N$ for a fixed context length $L = 80$. STAR-MD scales significantly better for large proteins due to the absence of cubic spatial terms. **Middle:** Scaling with context length $L$ for a fixed protein size $N = 256$. While baselines have lower temporal complexity, their high spatial cost dominates in the realistic regime where $N > L$. **Right:** Heatmap showing the speedup factor of STAR-MD over Pairformer + Temporal Attention on Singles as a function of protein size $N$ and context length $L$. STAR-MD achieves significant speedups in the realistic regime where $N > L$.

**ConfRover KV Cache Requirements.** During autoregressive generation, ConfRover must store in its KV cache:

- Singles features: $\mathcal{O}(N \times L \times d)$ memory
- Pairs features: $\mathcal{O}(N^2 \times L \times d)$ memory

Total KV cache memory: $\mathcal{O}((N + N^2) \times L \times d)$

For a medium-sized protein with $N = 200$ residues, a context window of $L = 32$ frames, and embedding dimension $d = 256$, this results in approximately:

$$\text{Memory for each layer} = (200 + 200^2) \times 32 \times 256 \times 4 \text{ bytes} \tag{24}$$
$$= (200 + 40000) \times 32 \times 256 \times 4 \text{ bytes} \tag{25}$$
$$\approx 1.3 \text{ GB} \tag{26}$$

**STAR-MD KV Cache Requirements.** In contrast, STAR-MD only needs to store keys and values for the singles features:

$$\text{Memory for each layer} = N \times L \times d \times 4 \text{ bytes} \tag{27}$$
$$= 200 \times 32 \times 256 \times 4 \text{ bytes} \tag{28}$$
$$\approx 6.6 \text{ MB} \tag{29}$$

This represents a **196× reduction** in KV cache memory requirements compared to ConfRover, which becomes increasingly significant as the protein size increases or when using longer context windows.

**Scaling to Larger Proteins and Longer Trajectories.** Figure 5 shows the estimated KV cache memory requirements for different protein sizes and trajectory lengths.

### B.5 PRACTICAL SCALING BEHAVIOR

While the asymptotic complexity of STAR-MD with respect to protein size $N$ is still quadratic ($\mathcal{O}(N^2L^2)$), our approach offers significant practical advantages:

1. **Elimination of cubic scaling terms**: The most expensive $\mathcal{O}(N^3)$ operations from the Pairformer are eliminated.

| Simulation time | Intervals (snapshots) | Number of frames |
|---|---|---|
| 100 ns | 120 | 80 |
| 240 ns | 120 | 200 |
| 1 μs | 250 | 400 |

Table 4: Inference task configurations for ATLAS dataset.

2. **Efficient attention implementations**: Our attention pattern is amenable to highly optimized implementations like FlashAttention Dao et al. (2022), which provides significant practical speedups.

3. **KV cache efficiency**: As shown in Section B.4, our approach dramatically reduces memory requirements during inference by avoiding the need to store pair features in the KV cache.

# Data & Implementation Details

## C  ATLAS DATASET

The ATLAS (Vander Meersche et al., 2024) dataset contains triplicated 100 ns simulation for 1390 proteins with diverse structures and dynamics, representative for different ECOD X-class domains. Following prior work (Jing et al., 2024a;b; Wang et al., 2024; Shen et al., 2025), we adopt a time-based split for ATLAS. Specifically, the train/validation/test sets are divided based on the release date of each protein, using cutoff dates of May 1, 2018 and May 1, 2019. Proteins released before May 1, 2018 are used for training, and proteins released after May 1, 2019 are used for testing. The diverse nature of ATLAS makes it a standard benchmark for evaluating protein conformation and trajectory generation under transferrable settings.

For each 100 ns trajectories, snapshots of atom coordinates were saved at 10 ps intervals. We use the coordinates for all heavy atoms in protein for model training and evaluation. Similar to Shen et al. (2025), we exclude the training proteins longer than 384 amino acids, leading to 1080 training proteins. We include the inference task configurations for different simulation time in Table 4.

## D  DETAILS ON EVALUATION METRICS

**Trajectory Alignment.**  Prior to any quantitative analysis (including conformational coverage and kinetic fidelity), all model-generated trajectories are aligned to the corresponding reference MD simulation. This is performed via C$\alpha$ superposition to a common reference frame, ensuring that all comparisons are independent of global rotational and translational differences.

**Conformational Coverage.**  Conformational state recovery is evaluated by comparing the distribution of model-generated and reference conformations in a PCA space. Each conformation is parameterized by the 3D coordinates of C$\alpha$ atoms. The PCA space is constructed using conformations from reference MD simulations in ATLAS. To compare distributions, each principal component is discretized into 10 evenly spaced bins. After projecting the conformations into this space, we count their occurrences in each bin and compute the distribution similarity using Jensen-Shannon Distance (JSD). We also binarize the occupancy counts to compute precision, recall, and F1-score – evaluating whether sampled conformations fall within known states, following prior works (Lu et al., 2024; Wang et al., 2024; Zheng et al., 2024; Shen et al., 2025).

**Kinetic Fidelity.**  Previous metrics operate on individual proteins and ensembles of proteins, independent of their temporal ordering. To characterize the temporal evolution of protein conformations, we evaluate four complementary metrics: coordinate root mean squared distance (RMSD), autocorrelation, tICA correlation (Shen et al., 2025), and VAMP-2 score (Wu & Noé, 2020).

Let $\boldsymbol{x}_{1:L} = (\boldsymbol{x}_1, \ldots, \boldsymbol{x}_L)$ denote a trajectory of $L$ frames, where each frame $\boldsymbol{x}_\ell \in \mathbb{R}^{N_{\text{atom}} \times 3}$ is the conformation at frame index $\ell$. Given a model-generated trajectory $\hat{\boldsymbol{x}}_{1:L}$ and MD reference $\boldsymbol{x}_{1:L}$, we project all trajectories onto the top 32 principal components computed using MD reference C$\alpha$ coordinates. For a particular lagtime $\tau$, the coordinate RMSD is computed via

$$\mathbb{E}_\ell[\text{RMSD}(\boldsymbol{x}_{\ell+\tau}, \boldsymbol{x}_\ell)], \tag{30}$$

which is an approximate average velocity highlighting the dynamical timescales of conformational changes. Autocorrelation is computed via

$$\mathbb{E}_\ell\left[\frac{(\boldsymbol{x}_\ell - \mu)(\boldsymbol{x}_{\ell+\tau} - \mu)}{\sigma^2}\right], \tag{31}$$

which is a measure of covariance within each trajectory. The aim of each metric is to approximate how the corresponding measure behaves in the MD reference data (dashed lines), capturing not only their overall trend but also characteristic magnitudes.

For tICA correlation, we follow Shen et al. (2025) and compare the principal dynamic modes of the generated and reference trajectories. Input trajectories are aligned and represented by $C_\alpha$ coordinates flattened to $d = 3N$ features. We first compute a validity mask $M \in \{0,1\}^L$ for the generated trajectory, retaining only frames that pass the "CA + AA Validity" checks defined in Appendix D. Only time-lagged pairs $(\boldsymbol{x}_\ell, \boldsymbol{x}_{\ell+\tau})$ where both frames are valid ($M_\ell = 1, M_{\ell+\tau} = 1$) are used for analysis. We fit separate tICA models to the reference and generated trajectories by solving $C_\tau \boldsymbol{v}_i = \lambda_i C_0 \boldsymbol{v}_i$, where $C_0$ and $C_\tau$ are the instantaneous and time-lagged covariance matrices, respectively. We use lagtimes $\tau \in [1, 5, 10, 20]$, apply kinetic map scaling, and set the regularization cutoff to $\epsilon = 10^{-6}$. For the top $k$ independent components (PC1 and PC2), we extract the left singular vector $\boldsymbol{v}^{(k)} \in \mathbb{R}^{3N}$ and compute a per-residue contribution score $S_i^{(k)} = \max\left(|v_{i,x}|, |v_{i,y}|, |v_{i,z}|\right)$ for each residue $i$. The final metric is the absolute Pearson correlation coefficient between the reference and generated scores: $|r_k| = |\text{Pearson}(S_{\text{ref}}^{(k)}, S_{\text{gen}}^{(k)})|$, averaged across PC1 and PC2. Our evaluation procedure mirrors Shen et al. (2025), with the sole exception that we only consider valid pairs via $M$. For any given lagtime, we report the tICA correlation only if at least 30 time-lagged pairs are valid; otherwise, we denote the result as "N/A".

Finally, the VAMP-2 score measures how well a set of features captures the slow dynamical modes of the system (Wu & Noé, 2020). The Koopman matrix is approximated via

$$K = C_{00}^{-1/2} C_{0\tau} C_{\tau\tau}^{-1/2}, \quad \text{where } C_{\ell_1 \ell_2} = \mathbb{E}_\ell[(\boldsymbol{x}_{\ell_1} - \mu)(\boldsymbol{x}_{\ell_1+\ell_2} - \mu)], \tag{32}$$

and the VAMP-2 score is the squared Frobenius norm of the Koopman matrix. We compute VAMP-2 score as a function of lagtime, projecting all trajectories onto the top 32 principal components computed using MD reference C$\alpha$ coordinates, consistent with the RMSD and autocorrelation metrics.

**Structural Quality and Validity**  To assess the physical plausibility of generated protein conformations, we use a suite of metrics that measure different aspects of structural integrity. A generated frame is considered **structurally valid** if it simultaneously satisfies all of the following conditions, which are based on thresholds derived from oracle MD simulations:

- **No C$_\alpha$ Chain Breaks:** The distance between consecutive C$_\alpha$ atoms must be below a certain threshold.

- **No C$_\alpha$ Clashes:** The distance between any two non-adjacent C$_\alpha$ atoms must be above a certain threshold.

- **Plausible Backbone Geometry:** The percentage of residues that are outliers in the Ramachandran plot must be below a threshold.

- **Plausible Side-Chain Geometry:** The percentage of residues with outlier rotamers must be below a threshold.

The specific thresholds for these metrics were determined by analyzing the distribution of these quality metrics in the ground-truth 100 ns MD simulation trajectories (our "oracle"). For each metric, we computed its value across all frames of the oracle trajectories and set the threshold at the 99th percentile (i.e., accepting 99% of the oracle frames). An exception was made for the C$_\alpha$ chain break

rate. The oracle MD simulations contained no chain breaks, resulting in a 99th percentile threshold of 0. This is too strict for current generative models. To allow for a more meaningful comparison, we set a small, non-zero tolerance for chain breaks.

The exact thresholds used for determining a valid frame are listed in Table 5. We define three sets of criteria for validity: **C$\alpha$ + All-Atom**, which requires passing all four checks; **All-Atom Only**, which checks only Ramachandran and rotamer criteria; and **C$\alpha$-Only**, which checks only for C$\alpha$ clashes and breaks.

Table 5: **Structural quality thresholds for validity.** Thresholds were derived from the 99th percentile of the oracle 100 ns MD simulations, except for the C$_\alpha$ break rate.

| Metric | Threshold |
|---|---|
| $C_\alpha$ + All-Atom Evaluation | |
| Ramachandran Outliers (%) | $\leq 4.12$ |
| Rotamer Outliers (%) | $\leq 7.05$ |
| C$_\alpha$ Clash Rate (%) | $\leq 1.29$ |
| C$_\alpha$ Break Rate (%) | $\leq 0.2$ |
| All-Atom Only Evaluation | |
| Ramachandran Outliers (%) | $\leq 4.12$ |
| Rotamer Outliers (%) | $\leq 7.05$ |
| $C_\alpha$-Only Evaluation | |
| C$_\alpha$ Clash Rate (%) | $\leq 1.29$ |
| C$_\alpha$ Break Rate (%) | $\leq 0.2$ |

# Architecture Details

## E   ARCHITECTURE DETAILS: PAIR FEATURES

STAR-MD modifies Shen et al. (2025)'s standard AlphaFold-derived architecture by removing the Pairformer module to reduce computational cost, while retaining explicit pairwise feature updates.

**FrameEncoder.** STAR-MD takes as input the pre-trained pairwise features from a frozen Open-Fold (Jumper et al., 2021) model. Then, it employs a `FrameEncoder` module Shen et al. (2025) which incorporates pairwise geometric information into the pair features.

**EdgeTransition.** Pairwise features $z \in \mathbb{R}^{N \times N \times d_z}$ are initialized by the `FrameEncoder` and updated via `EdgeTransition` layers. Unlike the global attention in Pairformer, `EdgeTransition` uses a local MLP update after each spatio-temporal attention block. For residues $i$ and $j$:

$$z_{ij} \leftarrow z_{ij} + \mathrm{MLP}(s_i, s_j, z_{ij}) \tag{33}$$

This maintains spatial context without $\mathcal{O}(N^3)$ complexity.

**Singles-Only Spatio-Temporal Attention.** Joint spatio-temporal attention is applied exclusively to single-residue features $s$. This avoids the $\mathcal{O}(N^2 L)$ memory cost of temporal attention over pairs. Pair features modulate the attention mechanism solely through bias terms, similar to Invariant Point Attention (IPA).

## F  BASELINE IMPLEMENTATION

**MD oracles.** In 100 ns simulation, we use one of the triplicated trajectories in the ATLAS dataset as the oracle to estimate the expected performance from an independent simulation run. To simulate longer trajectories, we follow the setup described in Vander Meersche et al. (2024) using provided equilibrated structures and Gromacs `.tpr` files to preproduce production runs at the desired lengths. All simulations are conducted using Gromacs (version 2023.2) on NVIDIA V100 GPUs with 40 GB memories. Independent reference and oracle trajectories are generated by repeating simulations with different random seeds.

**AlphaFolding (Cheng et al., 2025).** We followed the official data preparation and inference procedures provided by the authors [†], using the recommended model weights (`frame16_step_95000.pth`). Trajectories were generated using the recommended extrapolation script (`run_eval_extrapolation.sh`) with parameters `sample_step=1`, `n_motion=2`, and `n_frames=16`. This model employs a block-wise autoregressive scheme: it generates a fixed-size block of 16 frames conditioned on the final frames of the preceding block. This process is repeated to achieve the desired total simulation time by adjusting the `extrapolation_time` parameter.

AlphaFolding has a fixed time step of 10 ps, which differs from our model's variable stride capability. Generated trajectories were sub-sampled to match the frame rates required for evaluation.

**MDGen (Jing et al., 2024b).** We followed the official data preparation and inference procedures provided by the authors [‡], using the provided model weights (`atlas.ckpt`). The model was trained on ATLAS trajectories preprocessed with 400 ps intervals (stride 40 with base interval 10 ps). For inference, we used the forward simulation script (`sim_inference.py`) with parameters `num_frames=250` and `suffix=_R1`.

This model generates trajectories autoregressively in blocks of 250 frames with an internal timestep of 400 ps per frame. A single rollout produces 100 ns of simulation time (250 frames × 400 ps = 100 ns). For 100 ns trajectories, we used `num_rollouts=1`. For 240 ns and 1 μs trajectories, we extended generation by setting `num_rollouts` to the required number of sequential blocks. Generated trajectories were then sampled post-generation to match the frame rates required for evaluation.

**ConfRover (Shen et al., 2025).** We use the code and model weights provided by the authors. ConfRover is a frame-level autoregressive model similar to STAR-MD, and we adopt the same setup as our model to generate trajectories for each experiments. For 100 ns results, we enable CPU-offloaded caching to store full key-value history in system memory, while for 240 ns and 1 μs results, we employ a sliding-window cache (size=14) with attention sinks (first two frames) (Xiao et al., 2023a), labeled ConfRover-W. This reduces its temporal complexity from $\mathcal{O}(L^2)$ to $\mathcal{O}(L \times W)$ and prevents out-of-memory errors. Despite slight quality drop, ConfRover-W maintains comparable performance to the full-attention model, see Figure 7 and Table 6 for the validation on the 100 ns benchmark.

All generative model baselines are evaluated on NVIDIA H100 GPUs with 80 GB GPU memory and 2 TB system memory.

---

[†]https://github.com/fudan-generative-vision/dynamicPDB/tree/main/applications/4d_diffusion
[‡]https://github.com/bjing2016/mdgen

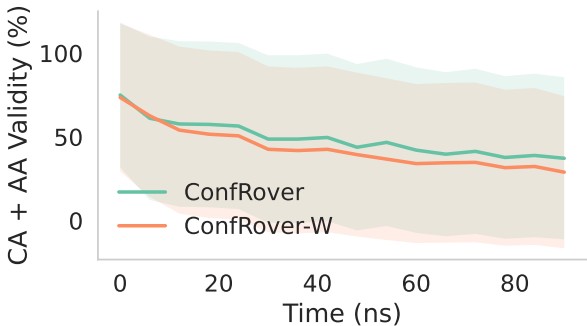

Figure 7: **Validation of ConfRover-W at 100 ns.** Comparison of Structural Validity (C$\alpha$ + All-Atom) between the standard full-attention ConfRover and the windowed ConfRover-W. The windowed variant maintains comparable performance, validating its use as a proxy for long-horizon experiments.

Table 6: **Comparison of ConfRover and ConfRover-W at 100 ns.** Quantitative metrics show that the windowed approximation (ConfRover-W) achieves performance comparable to the full-attention model (ConfRover), justifying its use for longer horizons.

| Model | Cov Valid | | Dyn. | Validity | | |
|---|---|---|---|---|---|---|
| | JSD↓ | Rec↑ | tICA↑ | CA%↑ | AA%↑ | CA+AA%↑ |
| ConfRover | 0.52 | 0.36 | 0.15 | 56.43 | 92.37 | 52.02 |
| ConfRover-W | 0.51 | 0.37 | 0.15 | 50.08 | 89.05 | 45.70 |
| STAR-MD | **0.42** | **0.57** | **0.17** | **86.83** | **98.23** | **85.35** |

# G    MODEL HYPERPARAMETERS

We include the details of our training and inference configuration in Tables 7 to 10. The model was trained on the ATLAS dataset following the train/val/test split in the previous works(Shen et al., 2025; Jing et al., 2024a;b; Wang et al., 2024; Cheng et al., 2025). STAR-MD contains 4 diffusion blocks, each containing 1 IPA layer and 2 S×T layers with hidden dimension 256 and 8 attention heads (Table 9). We train on trajectory snippets with context length $L = 8$ frames and global batch size of 8 (Table 7).

We use the Adam optimizer and learning rate $5 \times 10^{-5}$ with the similar loss setup as used in  Shen et al. (2025) (Table 10). Diffusion parameters are provided in Table 8.

For distributed training, we employed DeepSpeed Stage 2 optimization with gradient checkpointing to efficiently manage memory usage during training of large protein systems. All models are trained 8 NVIDIA H100 GPUs with 80 GB memory.

Table 7: **Dataset and Training Configuration**

| Parameter | Value |
|---|---|
| *Dataset Configuration* | |
| Training data | ATLAS train split (length $\leq$ 384) |
| Global batch size | 1 |
| Frames per trajectories | 8 |
| Frame intervals | $1 \sim 2^{10}$ of 10 ps snapshots |
| *Feature Representation* | |
| Single-residue feature dim | 384 |
| Pairwise feature dim | 128 |
| Number of recycles | 3 |
| *Data Augmentation* | |
| Clean noise max magnitude | 0.1 |
| Clean noise sampling prob | 0.75 |
| Noise sampling | Frame-level |

Table 8: **Diffusion Model Configuration**

| Parameter | Value |
|---|---|
| *SE(3) Diffusion* | |
| Coordinate scaling | 0.1 |
| Translation $b_{min}$/$b_{max}$ | 0.1 / 20.0 |
| Translation schedule | Linear |
| Rotation $\sigma_{min}$/$\sigma_{max}$ | 0.1 / 1.5 |
| Rotation schedule | Logarithmic |
| *Sampling* | |
| Method | Euler SDE |
| Diffusion steps | 200 |
| $t_{max}$ / $t_{min}$ | 1.0 / 0.01 |

Table 9: **Model Architecture**

| Component | Configuration |
|---|---|
| *Encoder (PseudoBetaPairEncoder)* | |
| Residue index embedding size | 128 |
| Output size | 128 |
| *Invariant Point Attention* | |
| IPA blocks | 4 |
| Single channel size | 256 |
| Pair channel size | 128 |
| Hidden channel size | 256 |
| Skip channel size | 64 |
| Attention heads | 4 |
| Query/key points | 8 |
| Value points | 12 |
| *Spatiotemporal Attention* | |
| Model dimension | 256 |
| Number of layers | 2 |
| Number of heads | 4 |
| RoPE 2D | Enabled |

Table 10: **Loss Functions and Optimization**

| Parameter | Value |
|---|---|
| *Loss Weights* | |
| Rotation score loss | 0.5 |
| Translation score loss | 1.0 |
| Torsion loss | 0.5 |
| Backbone FAPE loss | 0.5 |
| Sidechain FAPE loss | 0.5 |
| Backbone coordinates loss | 0.25 (diffusion $t \leq 0.25$) |
| Backbone distance map loss | 0.25 (diffusion $t \leq 0.25$) |
| *Optimization* | |
| Optimizer | Adam |
| Learning rate | 0.0002 |
| LR schedule | Linear warmup + cosine decay |
| Warmup epochs | 5 |
| Total epochs | 250 |
| Warmup start LR factor | 0.01 |
| Minimum LR factor | 0.1 |
| Gradient clipping | 1.0 (norm) |
| Precision | BF16 mixed precision |

# Extended Results

## H EXTENDED ATLAS ANALYSIS

### H.1 ADDITIONAL RESULTS ON DYNAMIC FIDELITY

We visualize the corresponding kinetic metrics for 100 ns, 240 ns, and 1 μs in Figure 8. For each metric, we include 2 separate MD simulations for reference (dashed lines). In all but one case (autocorrelation for 1 μs), STAR-MD better approximates the trend and values of MD references, showing higher kinetic fidelity than the rest of models.

We also include the corresponding metrics for our ablation study (Section 4.5) in Figure 9, where STAR-MD outperforms other variants in all but two cases (100 ns $C\alpha$ coordinate RMSD and 1 μs $C\alpha$ autocorrelation at large lagtime).

We further provide a detailed breakdown of the tICA correlation metric in Figure 10 and Table 11. Figure 10 shows the per-component and per-lagtime correlations for all valid runs, highlighting the variability across different simulation instances. Table 11 summarizes these results, demonstrating that STAR-MD achieves the highest mean correlation among generative models, closely matching the reference MD baseline.

Table 11: **Aggregated tICA Correlation Statistics.** Mean and standard deviation of tICA correlations averaged over PC1, PC2, and all lagtimes. We report results on both valid samples (Valid) and all samples (Unfiltered).

| Model | tICA (Valid) | tICA (Unfiltered) |
|---|---|---|
| MD (Oracle) | 0.170 | 0.17 |
| MDGen | $0.130 \pm 0.039$ | $0.12_{\pm 0.00}$ |
| AlphaFolding | N/A | 0.14 |
| ConfRover | $0.162 \pm 0.035$ | $\mathbf{0.18}_{\pm 0.01}$ |
| STAR-MD | $0.176 \pm 0.033$ | $\mathbf{0.18}_{\pm 0.01}$ |

We also report VAMP-2 score as a function of lagtime in Figure 11. STAR-MD better approximates the VAMP-2 score of MD references for all lagtimes compared to other models and ablation variants, demonstrating better capture of slow dynamical modes.

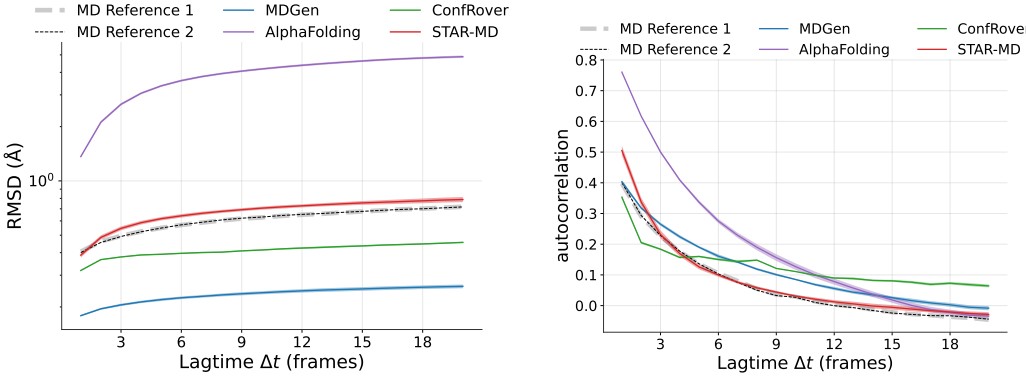

(a) Cα coordinate RMSD vs lagtime (100 ns simulation)  (b) Cα autocorrelation vs lagtime (100 ns simulation)

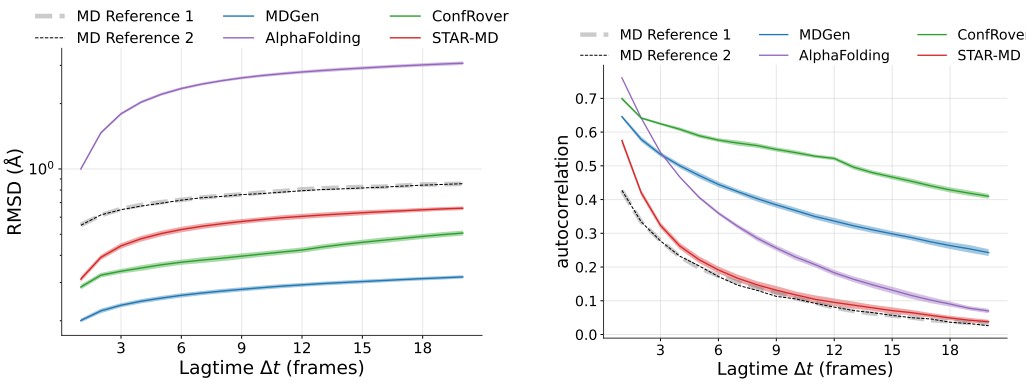

(c) Cα coordinate RMSD vs lagtime (240 ns simulation)  (d) Cα autocorrelation vs lagtime (240 ns simulation)

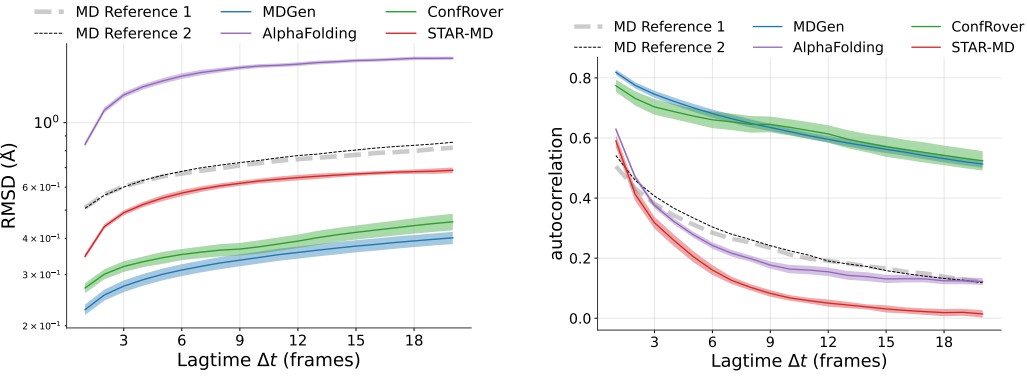

(e) Cα coordinate RMSD vs lagtime (1 μs simulation).  (f) Cα autocorrelation vs lagtime (1 μs simulation).

Figure 8: Kinetic metrics at different trajectory lengths: 100 ns, 240 ns, and 1 μs. Dashed lines are MD references (in most cases MD reference metrics are very close, demonstrating the metrics are robust across independent MD reference simulations). STAR-MD better approximates the MD references, demonstrating higher kinetic fidelity than the other models.

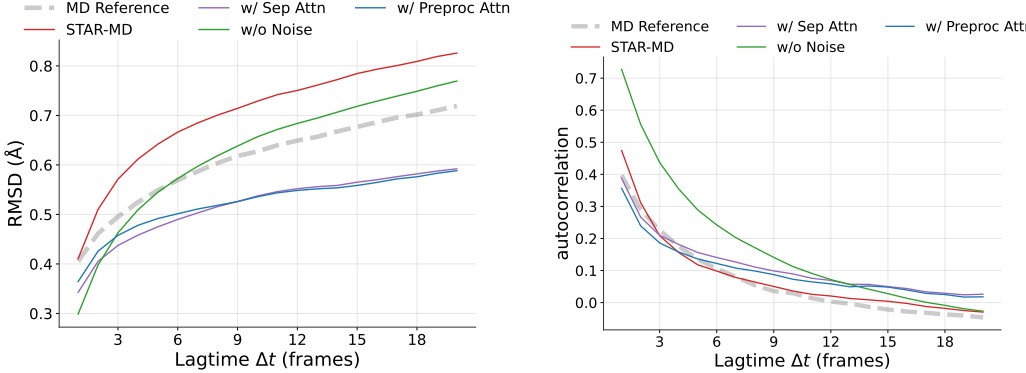

(a) Cα coordinate RMSD vs lagtime (100 ns simulation)  (b) Cα autocorrelation vs lagtime (100 ns simulation)

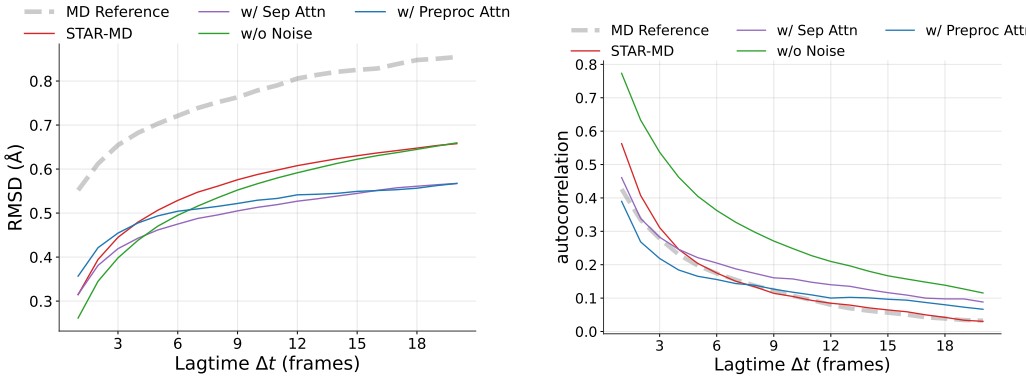

(c) Cα coordinate RMSD vs lagtime (240 ns simulation)  (d) Cα autocorrelation vs lagtime (240 ns simulation)

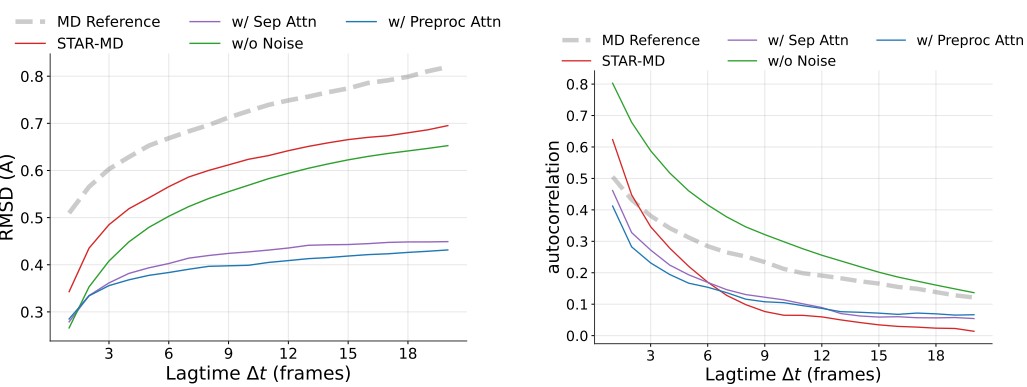

(e) Cα coordinate RMSD vs lagtime (1 μs simulation)  (f) Cα autocorrelation vs lagtime (1 μs simulation)

Figure 9: Kinetic metrics for ablation study. Dashed line is MD reference. STAR-MD has similar/better performances than other variants in all but two cases (100 ns Cα coordinate RMSD and 1 μs Cα autocorrelation at large lagtime).

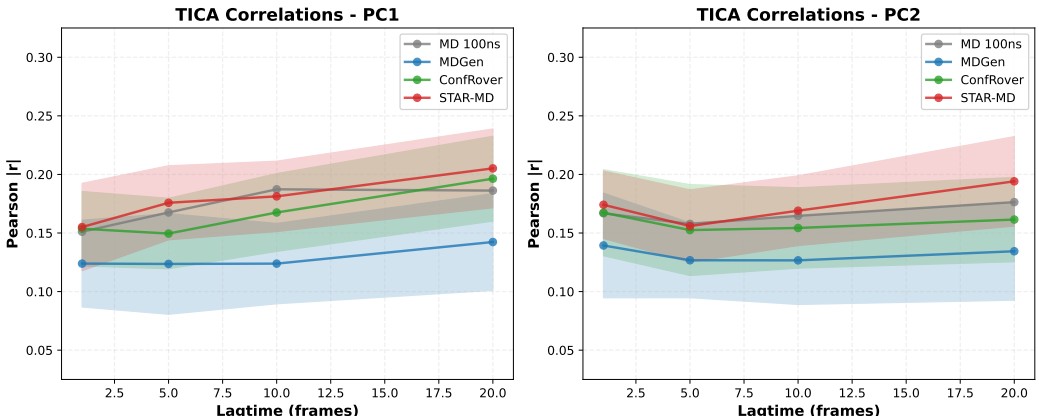

Figure 10: **Detailed tICA Correlation Analysis.** We report the per-component (PC1, PC2) and per-lagtime tICA correlation values for all runs that produced at least 30 valid pairs for computation. Shaded regions represent the standard deviation across valid runs.

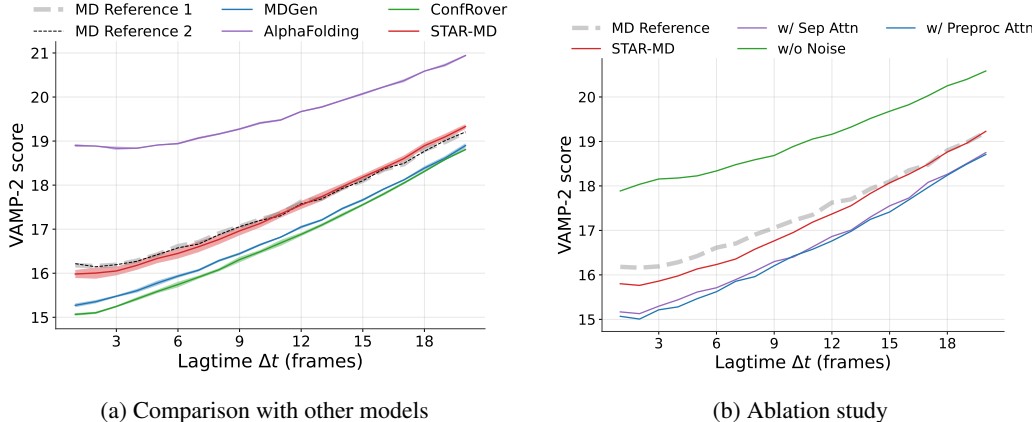

(a) Comparison with other models

(b) Ablation study

Figure 11: VAMP-2 score as a function of lagtime (100 ns simulation). Dashed line is MD reference. STAR-MD better approximates VAMP-2 score of MD reference than other models and variants, demonstrating better capture of slow dynamical modes.

## H.2 Full Ablation Results

We include the full results Tables for ablation study in this section (Table 12,13). We observe a consistent trend that contextual noise improves structure quality, and spatio-temporal attention improves conformation coverage.

Figure 12 compares the CA+AA validity over time for models with and without contextual noise on the 240 ns and 1 µs benchmarks, showing that the noise helps maintain structural quality over long horizons.

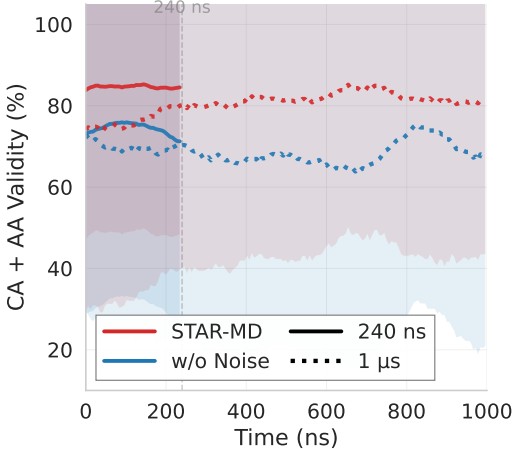

Figure 12: **Effect of Contextual Noise on Long-Horizon Stability.** Comparison of Structural Validity (C$\alpha$ + All-Atom) over time for STAR-MD with and without contextual noise on 240 ns and 1 µs benchmarks.

Table 12: **Complete ablation study results (240 ns).** We compare our full model, STAR-MD, against variants with key components removed. All metrics are reported for the 240 ns generation task. Metrics degraded in each ablation setting are highlighted in red.

| Model | Cov Valid | | Dynamic Fidelity | | Validity | | |
|---|---|---|---|---|---|---|---|
| | JSD↓ | Rec↑ | RMSD↓ | AutoCor↓ | CA%↑ | AA%↑ | CA+AA%↑ |
| STAR-MD | 0.45 | 0.59 | 0.20 | 0.02 | 86.44 | 97.61 | 84.62 |
| w/o Noise | 0.42 | 0.61 | 0.22 | 0.16 | 77.11 | 96.19 | 74.25 |
| w/ Sep Attn | 0.48 | 0.51 | 0.26 | 0.04 | 87.09 | 97.58 | 84.83 |
| w/ Preproc Attn | 0.47 | 0.52 | 0.25 | 0.03 | 87.25 | 96.72 | 84.28 |

Table 13: **Complete ablation study results (1 µs).** We compare our full model, STAR-MD, against variants with key components removed. All metrics are reported for the 1 µs generation task. Metrics degraded in each ablation setting are highlighted in red.

| Model | Cov Valid | | Dynamic Fidelity | | Validity | | |
|---|---|---|---|---|---|---|---|
| | JSD↓ | Rec↑ | RMSD↓ | AutoCor↓ | CA%↑ | AA%↑ | CA+AA%↑ |
| STAR-MD | 0.45 | 0.61 | 0.11 | 0.11 | 91.00 | 91.44 | 83.59 |
| w/o Noise | 0.48 | 0.63 | 0.17 | 0.10 | 75.84 | 91.22 | 68.66 |
| w/ Sep Attn | 0.51 | 0.49 | 0.30 | 0.10 | 93.12 | 92.03 | 85.31 |
| w/ Preproc Attn | 0.52 | 0.49 | 0.32 | 0.11 | 94.12 | 92.19 | 86.44 |

## H.3   100 ns Filtered Results

### H.3.1   Performance by Sequence Length

We break down the performance of STAR-MD and baselines across different protein sequence length buckets to analyze scalability and robustness. Tables 14 to 18 summarize these findings.

Table 14: **Sequence Length Bucket: 0-100 (17 chains)**

| Model | Cov Valid | | Dyn. | Validity | | |
|---|---|---|---|---|---|---|
| | JSD↓ | Rec↑ | tICA↑ | CA%↑ | AA%↑ | CA+AA%↑ |
| MD (Oracle) | 0.28 | 0.71 | 0.10 | 100.00 | 93.24 | 93.24 |
| MDGen | $0.48_{\pm0.01}$ | $0.38_{\pm0.01}$ | $\mathbf{0.11}_{\pm\mathbf{0.01}}$ | $85.66_{\pm3.15}$ | $87.34_{\pm1.76}$ | $75.91_{\pm2.08}$ |
| AlphaFolding | 0.56 | 0.28 | N/A | 30.07 | 3.60 | 2.06 |
| ConfRover | $0.48_{\pm0.02}$ | $0.39_{\pm0.02}$ | $\mathbf{0.11}_{\pm\mathbf{0.01}}$ | $79.59_{\pm2.00}$ | $81.24_{\pm0.18}$ | $65.26_{\pm1.30}$ |
| STAR-MD | $\mathbf{0.37}_{\pm\mathbf{0.01}}$ | $\mathbf{0.56}_{\pm\mathbf{0.02}}$ | $\mathbf{0.11}_{\pm\mathbf{0.01}}$ | $\mathbf{91.37}_{\pm\mathbf{1.87}}$ | $\mathbf{93.75}_{\pm\mathbf{0.47}}$ | $\mathbf{86.10}_{\pm\mathbf{1.88}}$ |

Table 15: **Sequence Length Bucket: 100-150 (16 chains)**

| Model | Cov Valid | | Dyn. | Validity | | |
|---|---|---|---|---|---|---|
| | JSD↓ | Rec↑ | tICA↑ | CA%↑ | AA%↑ | CA+AA%↑ |
| MD (Oracle) | 0.30 | 0.69 | 0.12 | 98.98 | 98.44 | 97.42 |
| MDGen | $0.51_{\pm0.02}$ | $0.35_{\pm0.02}$ | $0.10_{\pm0.01}$ | $74.50_{\pm7.40}$ | $94.17_{\pm1.39}$ | $70.62_{\pm7.62}$ |
| AlphaFolding | 0.58 | 0.10 | N/A | 14.38 | 0.23 | 0.16 |
| ConfRover | $0.53_{\pm0.01}$ | $0.35_{\pm0.01}$ | $\mathbf{0.12}_{\pm\mathbf{0.01}}$ | $55.89_{\pm0.95}$ | $86.38_{\pm0.78}$ | $49.08_{\pm1.18}$ |
| STAR-MD | $\mathbf{0.49}_{\pm\mathbf{0.01}}$ | $\mathbf{0.53}_{\pm\mathbf{0.01}}$ | $\mathbf{0.12}_{\pm\mathbf{0.01}}$ | $\mathbf{79.05}_{\pm\mathbf{3.41}}$ | $\mathbf{98.25}_{\pm\mathbf{0.35}}$ | $\mathbf{77.64}_{\pm\mathbf{3.49}}$ |

Table 16: **Sequence Length Bucket: 150-225 (11 chains)**

| Model | Cov Valid | | Dyn. | Validity | | |
|---|---|---|---|---|---|---|
| | JSD↓ | Rec↑ | tICA↑ | CA%↑ | AA%↑ | CA+AA%↑ |
| MD (Oracle) | 0.27 | 0.66 | 0.16 | 90.57 | 99.66 | 90.23 |
| MDGen | $0.52_{\pm0.02}$ | $0.32_{\pm0.03}$ | $0.12_{\pm0.01}$ | $66.86_{\pm5.84}$ | $97.18_{\pm0.91}$ | $65.23_{\pm5.65}$ |
| AlphaFolding | 0.72 | 0.10 | N/A | 4.55 | 0.34 | 0.11 |
| ConfRover | $0.47_{\pm0.02}$ | $0.39_{\pm0.01}$ | $0.15_{\pm0.01}$ | $73.39_{\pm2.56}$ | $98.77_{\pm0.34}$ | $72.43_{\pm2.43}$ |
| STAR-MD | $\mathbf{0.37}_{\pm\mathbf{0.02}}$ | $\mathbf{0.53}_{\pm\mathbf{0.03}}$ | $\mathbf{0.16}_{\pm\mathbf{0.01}}$ | $\mathbf{89.55}_{\pm\mathbf{1.70}}$ | $\mathbf{99.82}_{\pm\mathbf{0.22}}$ | $\mathbf{89.41}_{\pm\mathbf{1.81}}$ |

Table 17: **Sequence Length Bucket: 225-400 (15 chains)**

| Model | Cov Valid | | Dyn. | Validity | | |
|---|---|---|---|---|---|---|
| | JSD↓ | Rec↑ | tICA↑ | CA%↑ | AA%↑ | CA+AA%↑ |
| MD (Oracle) | 0.33 | 0.65 | 0.22 | 100.00 | 99.90 | 99.90 |
| MDGen | $0.62_{\pm0.01}$ | $0.21_{\pm0.01}$ | $0.14_{\pm0.00}$ | $61.69_{\pm4.22}$ | $97.71_{\pm1.06}$ | $60.40_{\pm4.75}$ |
| AlphaFolding | N/A | N/A | N/A | 1.15 | 0.00 | 0.00 |
| ConfRover | $\underline{0.53_{\pm0.00}}$ | $\underline{0.36_{\pm0.01}}$ | $\mathbf{0.22_{\pm0.02}}$ | $54.00_{\pm2.82}$ | $\underline{99.44_{\pm0.08}}$ | $53.77_{\pm2.95}$ |
| STAR-MD | $\mathbf{0.44_{\pm0.02}}$ | $\mathbf{0.53_{\pm0.02}}$ | $\mathbf{0.22_{\pm0.02}}$ | $\mathbf{82.38_{\pm1.62}}$ | $\mathbf{99.98_{\pm0.04}}$ | $\mathbf{82.37_{\pm1.63}}$ |

Table 18: **Sequence Length Bucket: 400+ (23 chains)**

| Model | Cov Valid | | Dyn. | Validity | | |
|---|---|---|---|---|---|---|
| | JSD↓ | Rec↑ | tICA↑ | CA%↑ | AA%↑ | CA+AA%↑ |
| MD (Oracle) | 0.33 | 0.66 | 0.24 | 99.86 | 100.00 | 99.86 |
| MDGen | $0.65_{\pm0.01}$ | $0.16_{\pm0.01}$ | $0.14_{\pm0.01}$ | $66.75_{\pm3.69}$ | $99.81_{\pm0.17}$ | $66.67_{\pm3.67}$ |
| AlphaFolding | N/A | N/A | N/A | 0.97 | 0.00 | 0.00 |
| ConfRover | $\underline{0.57_{\pm0.01}}$ | $\underline{0.32_{\pm0.01}}$ | $\mathbf{0.26_{\pm0.01}}$ | $28.54_{\pm1.14}$ | $\underline{99.61_{\pm0.31}}$ | $28.54_{\pm1.14}$ |
| STAR-MD | $\mathbf{0.45_{\pm0.01}}$ | $\mathbf{0.54_{\pm0.03}}$ | $\underline{0.24_{\pm0.01}}$ | $\mathbf{90.92_{\pm1.77}}$ | $\mathbf{100.00_{\pm0.00}}$ | $\mathbf{90.92_{\pm1.77}}$ |

### H.3.2 GENERALIZATION TO DISSIMILAR PROTEINS

To strictly evaluate generalization to unseen protein folds, we report results excluding two test proteins (7buy_A and 7e2s_A) that have high sequence similarity ($> 80\%$) to the training set. Table 19 summarizes these findings.

Table 19: **Non-similar Proteins, Excluding 7buy_A, 7e2s_A (80 chains)**

| Model | Cov Valid | | Dyn. | Validity | | |
|---|---|---|---|---|---|---|
| | JSD↓ | Rec↑ | tICA↑ | CA%↑ | AA%↑ | CA+AA%↑ |
| MD (Oracle) | 0.31 | 0.67 | 0.17 | 98.37 | 98.07 | 96.43 |
| MDGen | $0.56_{\pm0.01}$ | $0.28_{\pm0.01}$ | $0.12_{\pm0.00}$ | $\underline{71.83_{\pm1.90}}$ | $\underline{95.03_{\pm0.59}}$ | $\underline{68.31_{\pm2.20}}$ |
| AlphaFolding[*] | 0.58 | 0.22 | N/A | 10.98 | 0.92 | 0.52 |
| ConfRover | $\underline{0.52_{\pm0.01}}$ | $\underline{0.36_{\pm0.01}}$ | $\underline{0.15_{\pm0.01}}$ | $56.94_{\pm0.52}$ | $92.47_{\pm0.25}$ | $52.06_{\pm0.36}$ |
| STAR-MD | $\mathbf{0.43_{\pm0.01}}$ | $\mathbf{0.54_{\pm0.01}}$ | $\mathbf{0.17_{\pm0.00}}$ | $\mathbf{86.81_{\pm0.64}}$ | $\mathbf{98.18_{\pm0.05}}$ | $\mathbf{85.29_{\pm0.62}}$ |

## H.4 Additional Visualizations of Conformational Coverage

Figures 13 to 22 show conformational coverage of 10 proteins from the test set. These visualizations follow the same format as Figure 2(b) in the main text (i.e., we filter the generated conformations to show only those that are structurally valid (passing all validity checks)). This provides a more rigorous view of the useful conformational space explored by each model.

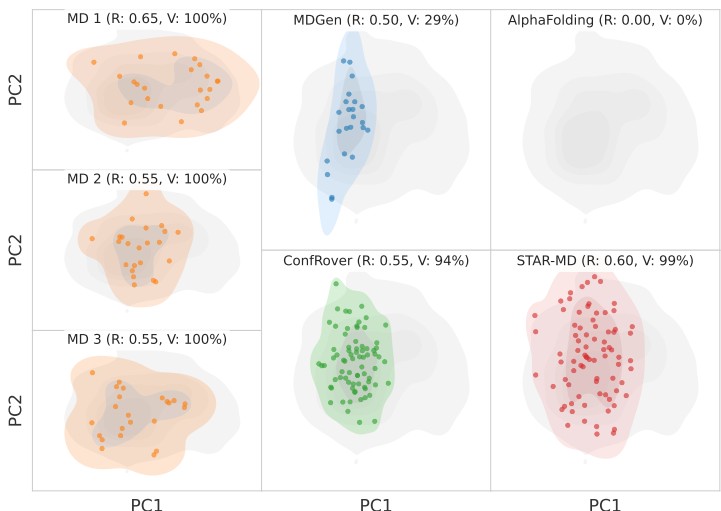

Figure 13: Conformational coverage for 6kty_A (Valid conformations only).

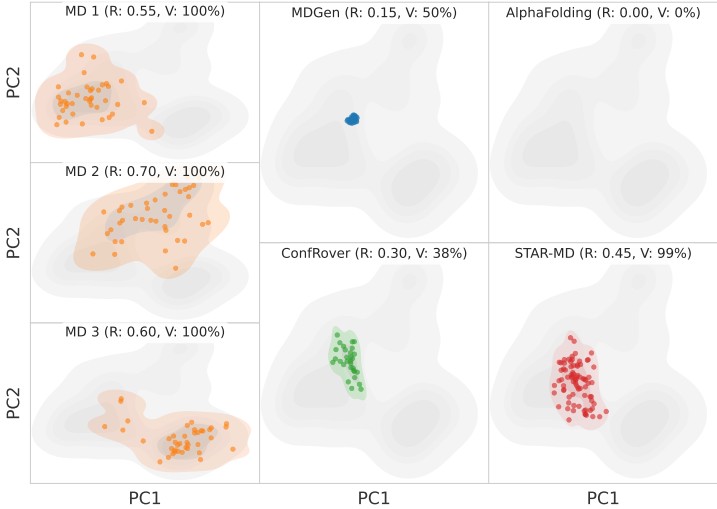

Figure 14: Conformational coverage for 6jwh_A (Valid conformations only).

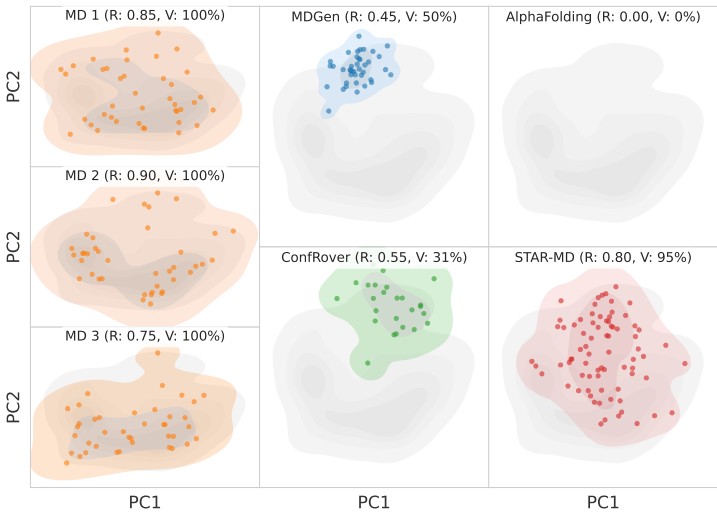

Figure 15: Conformational coverage for 6jpt_A (Valid conformations only).

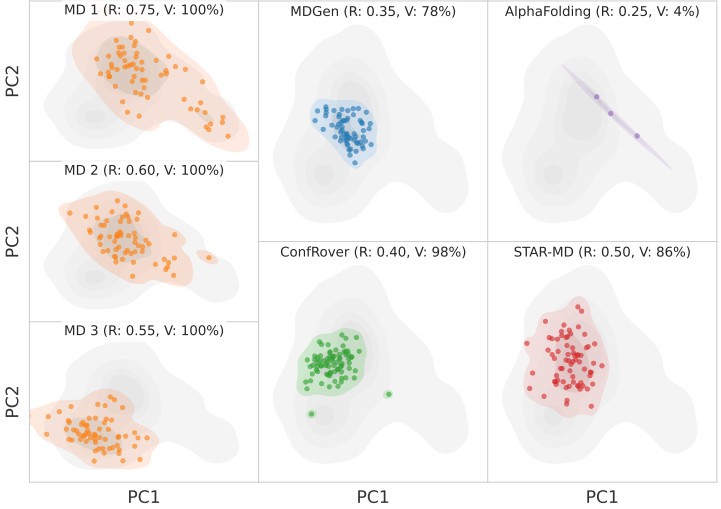

Figure 16: Conformational coverage for 6okd_C (Valid conformations only).

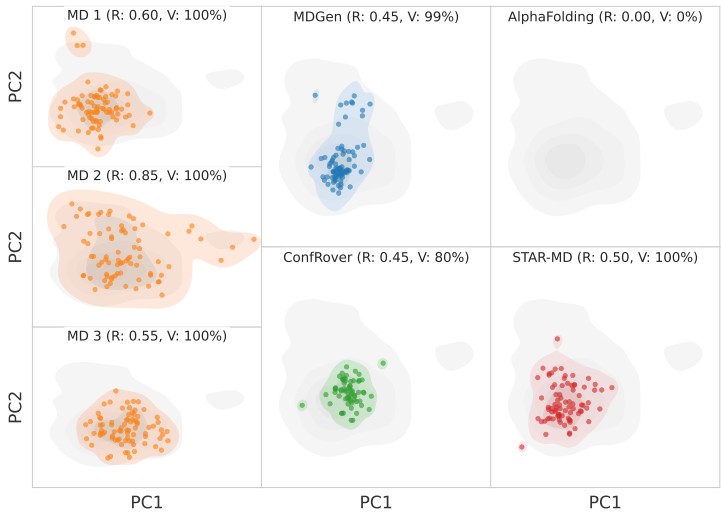

Figure 17: Conformational coverage for 6odd_B (Valid conformations only).

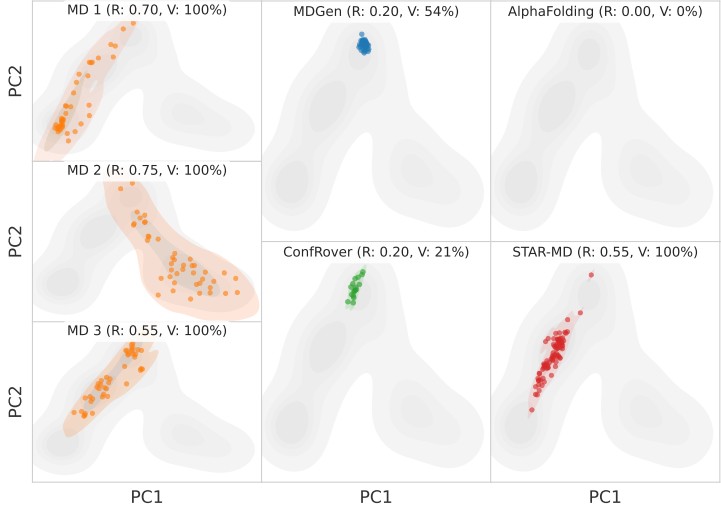

Figure 18: Conformational coverage for 7dmn_A (Valid conformations only).

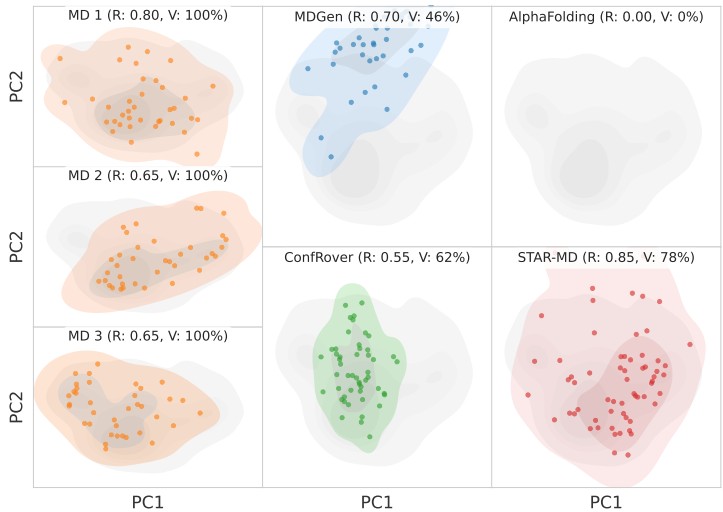

Figure 19: Conformational coverage for 7lp1_A (Valid conformations only).

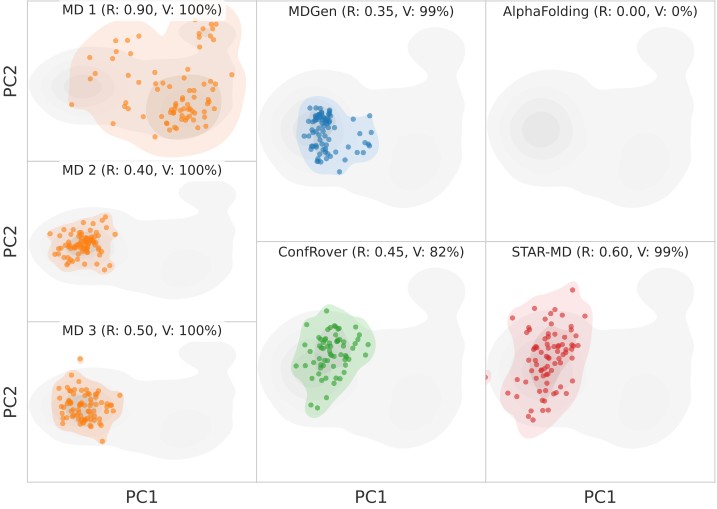

Figure 20: Conformational coverage for 6l34_A (Valid conformations only).

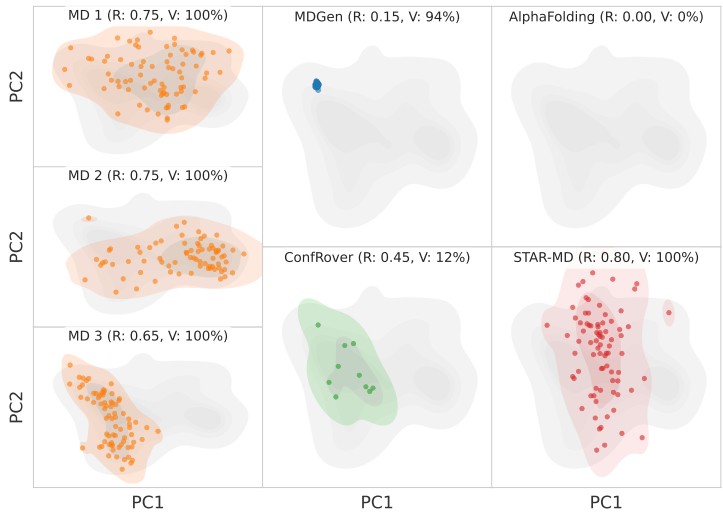

Figure 21: Conformational coverage for 5znj_A (Valid conformations only).

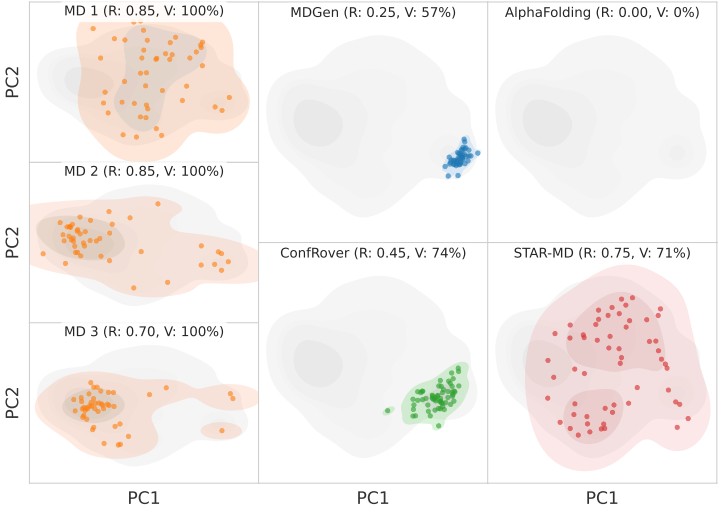

Figure 22: Conformational coverage for 6tly_A (Valid conformations only).

# I   STABILITY AND SCALABILITY ANALYSIS

## I.1   EXTENDED STABILITY ANALYSIS

To further characterize the stability of STAR-MD, we provide a detailed breakdown of the variability in structural validity (C$\alpha$ + All-Atom) for the 100 ns, 240 ns, and 1 µs settings. Figure 23 decomposes the total variability into inter-seed and inter-protein components.

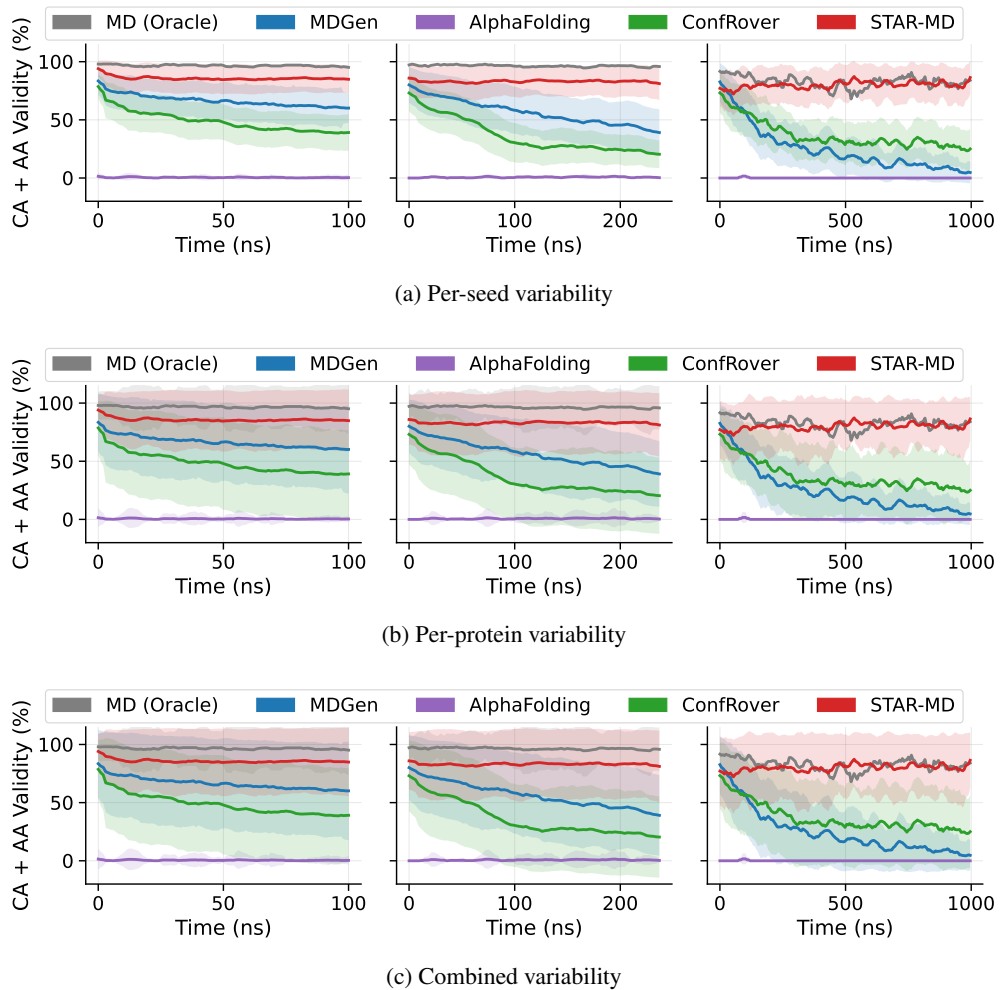

(a) Per-seed variability

(b) Per-protein variability

(c) Combined variability

Figure 23: **Breakdown of stability variability.** We analyze the standard deviation of the Structural Validity (C$\alpha$ + All-Atom) metric over time. (a) Variability across 5 random seeds for a single protein (as reported in the main text). (b) Variability across different test proteins for a single seed. (c) Combined variability incorporating both seed and protein variance.

I.2 EXTREME-HORIZON GENERATION (10 µs)

To probe the limits of STAR-MD's stability, we generated trajectories using the maximum training stride (∼10 ns) for 1000 steps, resulting in a total simulation time of approximately 10 µs. Table 20 reports the structural validity metrics averaged over the full 10 µs trajectory for the ATLAS test set. Figure 24 visualizes the stability of structural validity over the course of the generation.

Table 20: **Structural validity for 10 µs trajectories.** Metrics are averaged over 1000 steps (∼10 ns stride) for the ATLAS test set.

| Metric | Validity (%) |
|---|---|
| CA Validity | 85.21 |
| AA Validity | 90.93 |
| CA + AA Validity | 77.28 |

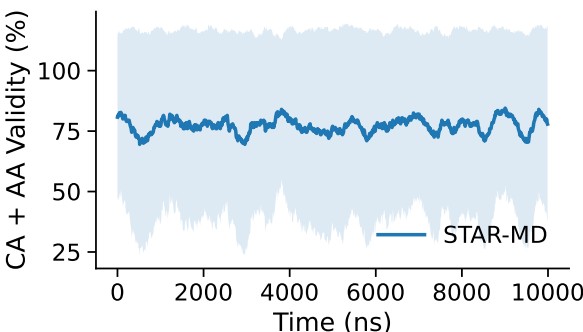

Figure 24: **Structural validity over 10 µs.** The average CA + AA Validity remains stable throughout the 1000-step generation process.

## I.3 INFERENCE COST

We evaluated the wall-clock time for generating 100 ns trajectories under our 100 ns benchmark setting for each methods, across proteins of varying lengths (50~750 amino acids). To estimate the runtime of molecular dynamic (MD) simulation, we used OpenMM with the Amber force field and implicit solvent, running a 0.1 ns simulation and extrapolating the cost to 100 ns. MDGen, trained with a time step of 400 ps and window of 250 frames, can directly generate 100 ns trajectories in single joint diffusion process and therefore provides the greatest acceleration. AlphaFolding, trained on a fixed time step of 10 ps and a window of 16 frames, requires substantially longer inference time to complete a 100 ns simulation. Both ConfRover and STAR-MD support varying-stride simulation and can generate under our evaluation configuration (stride=1.2 ns, 80 frames). While STAR-MD is slightly slower than ConfRover for small proteins, it is more efficient when scaling to larger proteins and is faster on 6SMS-A (724 AA). All methods are evaluated on a single NVIDIA H100 GPU.

Table 21: **Inference cost for different methods.** The wall-clock time (in seconds) to generate 100 ns trajectories across proteins with varying lengths (shown in parentheses). AlphaFolding encounters out-of-memory error for larger proteins and is reported as 'N/A'. All methods are evaluated on a single NVIDIA H100 GPU.

| Method | 6OKD-C (51) | 6Q9C-A (155) | 6XB3-H (241) | 7AQX-A (364) | 7P41-D (448) | 7MF4-A (554) | 6L8S-A (650) | 6SMS-A (724) |
|---|---|---|---|---|---|---|---|---|
| MD | 6322.8 | 9610.8 | 14420.3 | 24272.4 | 34850.8 | 49294.4 | 66296.7 | 78451.4 |
| AlphaFolding | 3135.7 | 5846.5 | 10024.8 | 19036.3 | 27340.3 | 40472.5 | N/A | N/A |
| MDGen | 4.1 | 8.1 | 13.0 | 20.5 | 28.2 | 35.3 | 41.2 | 54.4 |
| ConfRover | 404.5 | 423.6 | 463.5 | 573.7 | 710.4 | 934.4 | 1170.2 | 1617.7 |
| STAR-MD | 766.3 | 807.4 | 818.4 | 795.8 | 902.5 | 1055.5 | 1232.4 | 1381.2 |

## J APPLICATIONS TO OTHER SYSTEMS

### J.1 ENSEMBLE GENERATION

We examine the performance of STAR-MD to approximate target ensemble distributions from MD simulation, and compare it against the state-of-the-art ensemble emulator BioEmu (Lewis et al., 2024). Due to several differences in training dataset and paradigms, we evaluate them on two benchmarks: (1) the standard ATLAS ensemble benchmark used in prior works (Jing et al., 2024a;b), and (2) the CATH1 ensemble benchmark introduced in BioEmu.

We follow the standard protocol from Jing et al. (2024a), sampling 250 conformations for each of the 82 test proteins and comparing the generated ensembles with MD reference ensembles. For STAR-MD, we repeat the 100 ns simulation procedure five times and randomly sample 250 conformations from the collected set. As shown in Table 22, STAR-MD better captures the expected diversity, more accurately reflects both global and local flexibility, and matches the reference ensemble at distributional and ensemble-observable levels. In contrast, BioEmu tends to produce overly diverse samples and shows lower similarity to the reference ensemble across most metrics.

We then evaluate ensemble generation performance on the CATH1 benchmark from BioEmu. This benchmark contains the longer trajectories at microsecond level for 17 short proteins ($< 75$ amino acids). We follow BioEmu's setup by sampling 10,000 conformations per system and projecting them onto the reaction coordinates to estimate free-energy and compute metrics using author's evaluation code. For STAR-MD, we sample from five different starting frames of the initial MD runs in the datasets, generating 80 frames at a stride of 1.28 ns. We repeat this process 25 times to collect 10,000 samples per protein. As shown in Figure 25, BioEmu achieves higher accuracy with lower error and higher coverage. When compared to the ground-truth distributions, STAR-MD concentrates around the low energy basins, while BioEmu explores a broader range of conformations, occasionally extending beyond the region supported by the ground truth.

To verify this performance difference is not caused by the forward simulation setup used for STAR-MD, we also test unconditional generation (STAR-MD-iid) by inferring single frames from masked token, similar to (Shen et al., 2025). The similar results between STAR-MD and STAR-MD-iid suggest that the performance gap between STAR-MD and BioEmu on CATH1 likely arises from factors

other than the trajectory-generation archiecture, such as differences in training datasets, training objectives, or fine-tuning procedures.

| | STAR-MD | BioEmu |
|---|---|---|
| RMSF (=1.63) | **1.38** | 2.87 |
| Pairwise RMSD (=2.76) | **2.34** | 4.51 |
| Pairwise RMSD r (↑) | **0.52** | 0.33 |
| Global RMSF r (↑) | **0.56** | 0.52 |
| Per target RMSF r (↑) | **0.86** | 0.82 |
| RMWD (↓) | **2.85** | 4.33 |
| MD PCA W2 (↓) | **1.50** | 1.64 |
| Joint PCA W2 (↓) | **2.33** | 3.43 |
| PC sim > 0.5 (↑) | **35.4 %** | 26.8 % |
| Weak contacts J (↑) | **0.52** | 0.49 |
| Transient contacts J (↑) | **0.37** | **0.37** |
| Exposed residue J (↑) | **0.57** | 0.54 |
| Exposed MI matrix rho (↑) | 0.24 | **0.26** |

Table 22: **ATLAS ensemble results.** Reported metrics include diversity, flexibility accuracy, distributional similarity, and ensemble observables. Parentheses indicate whether higher/lower values are the better, or the expected value estimated from the ground truth trajectory. The better value is shown in bold.

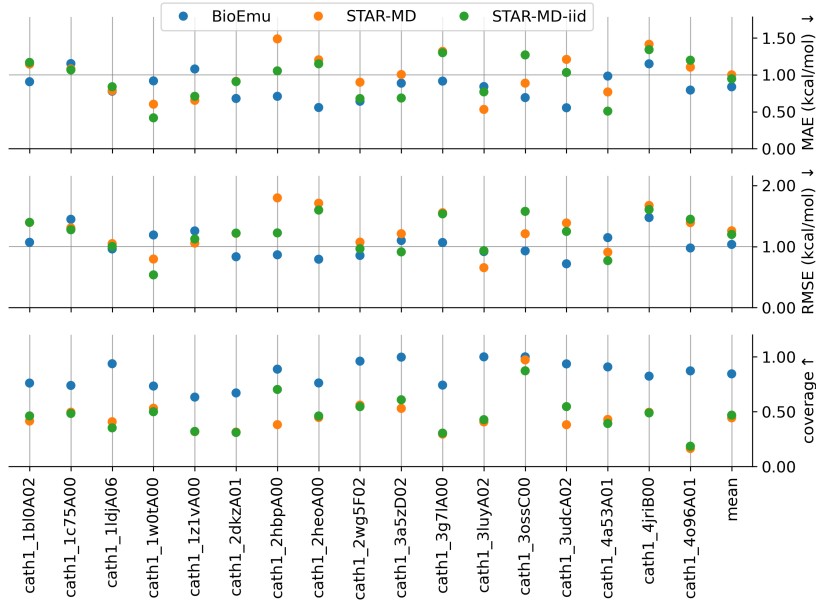

Figure 25: **Free energy accuracy and coverage from the BioEmu CATH1 ensemble benchmark.**

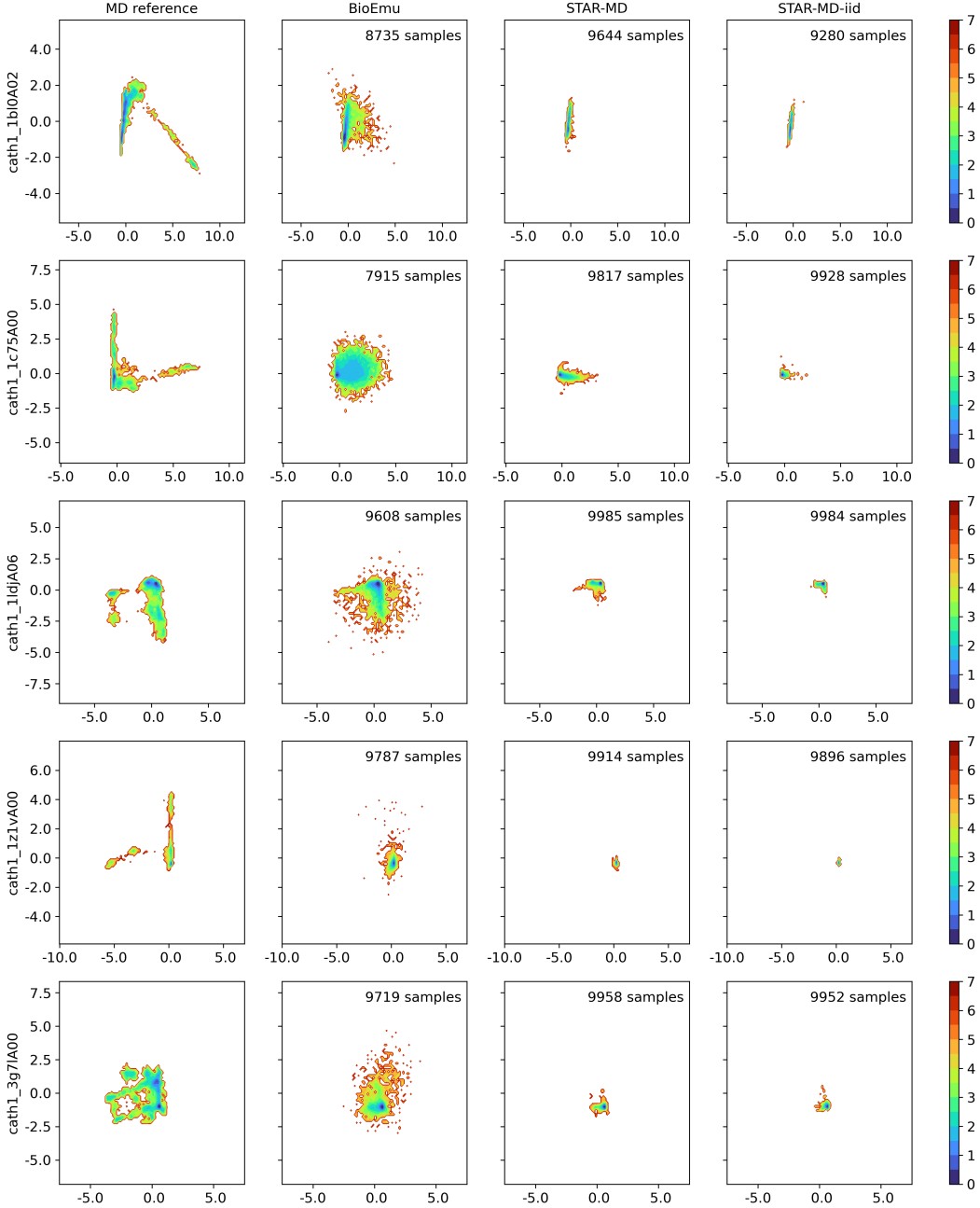

Figure 26: **Example free energy surfaces from the BioEmu CATH1 ensemble benchmark.** Five systems are randomly selected for visualization. The left columns shows the reference MD distributions provided in BioEmu. For each model and system, the number of valid samples (out of 10,000) that passes BioEmu's quality filter is indicated in the upper-right corner of each subplot.

## J.2 CASE STUDIES ON FUNCTION-RELATED CONFORMATIONAL DYNAMICS

### J.2.1 CRYPTIC-POCKET BINDING

Sampling conformational changes during ligand binding is challenging, especially for proteins with cryptic pockets, where cooperative and complex pocket dynamics are required to enable binding. To assess whether STAR-MD can sample bound-state (*holo*) conformations from unbound-state (*apo*) structures in such settings, we selected three cases from BioEmu's cryptic-pocket dataset: Adenylosuccinate Synthetase (AdS), an enzyme has several allosteric or competitive binding substrates; TEM-1 $\beta$-Lactamase (TEM-1), forming complexes with allosteric inhibitors; and Adenylate Kinase (AdK), a highly flexible protein that undergoes large structural arrangement to each closed *holo* conformation with covered 'lid'.

For each of the case, we initialize from the provided *apo* structure and generate 200 frames at a stride of 2.56 ns, approximating a 500 ns simulation. We compare sampled conformations with the reference **holo** structure and compute RMSD over the local binding residues involved in the conformational changes[§]. The best *holo* samples with minimum local RMSD are shown in Figure 27. Across all three cases, STAR-MD successfully samples *holo*-like conformations with RMSD < 1.5 Å (Lewis et al., 2024). For AdS and AdK, the sampled conformations closely algin with the target *holo* structures. For TEM-1, we observe remaining differences in the left small helix where the sampled structure does not fully form the helix. Nevertheless, these results demonstrate promising capability of STAR-MD in predicting complex, cooperative, and long-range conformational dynamics.

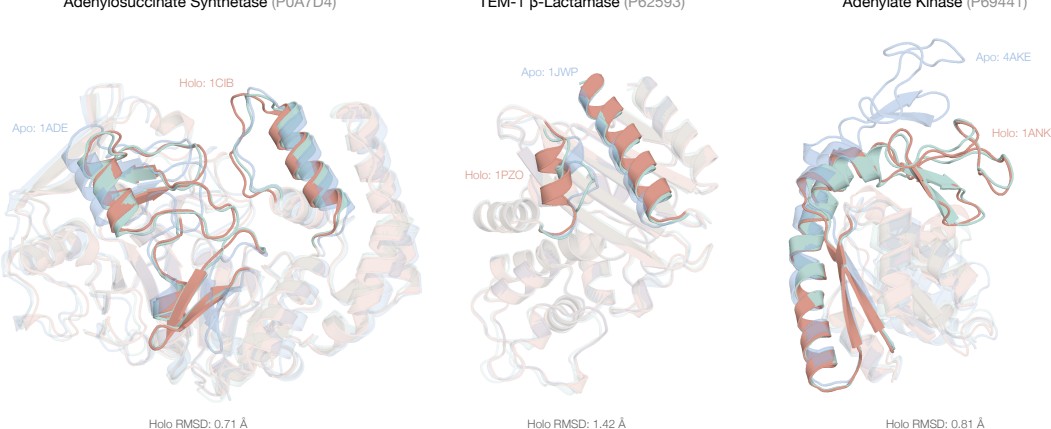

Figure 27: **STAR-MD samples *Holo* structures started from *Apo*.** Structures are superimposed and the local binding regions are highlighted. The Holo RMSD shown under each case is calcualted based on the local binding residues.

### J.2.2 KINASE ACTIVATION

Abl kinase is a signaling protein related to leukemia and other cancers. It undergoes conformational transitions between active and inactive states, which regulate its activity. These transitions involve several functional domains (P-loop, $\alpha$C-helix, and the Asp-Phe-Gly DFG motifs), imposing complex millisecond-scale dynamics (Xie et al., 2020). We investigate whether STAR-MD can sample the active-inactive transition when initialized from one of the two states. Specifically, we generate trajectories of 500 frames with a stride of 5.12 ns, corresponding to roughly 2.5 μs of simulation.

To assess the states of generated conformations, we extract pairwise C$\alpha$ distances and backbone dihedral angles ($\psi, \phi$) for residues around function domains involved in the transition (THR262-GLU277, VAL287-LEU317, LYS397-GLY417), from 20 active-state structures (PDB:6XR6) and 20 inactive-state structures (PDB:6XR7). We then apply principle component analysis to obtain a

---

[§]Residue information: https://github.com/microsoft/bioemu-benchmarks/tree/main/bioemu_benchmarks/assets/multiconf_benchmark_0.1/crypticpocket/local_residinfo

2D low-dimensional projection that separates the active and inactive conformations. We also define a core-RMSD metric measured as the $C\alpha$-RMSD between sample and the reference structure over the aforementioned residues of interest.

As shown in Figure 28, when starting from the inactive state, STAR-MD is able to sample towards the active state with the best core-RMSD to the active state as 2.00 Å. When sampling from the active state, the majority of sampled conformations remain near the active-state region, while a subset of samples explores the inactive region. This behavior is in line with the underlying energetic profile, where approximately 88% of conformations correspond to the active state and only $\sim$8% correspond to the inactive state (Xie et al., 2020). The best RMSD to the inactive reference (2.60 Å) is still smaller than the average pairwise RMSD between active and inactive conformations in PDB (3.30 $\pm$ 0.34 Å).

Overall, this case demonstrates the potential of STAR-MD to sample complex conformational dynamics such as active-inactive transition in kinases.

### Sample from inactive state (6XR7) to active state (6XR6)

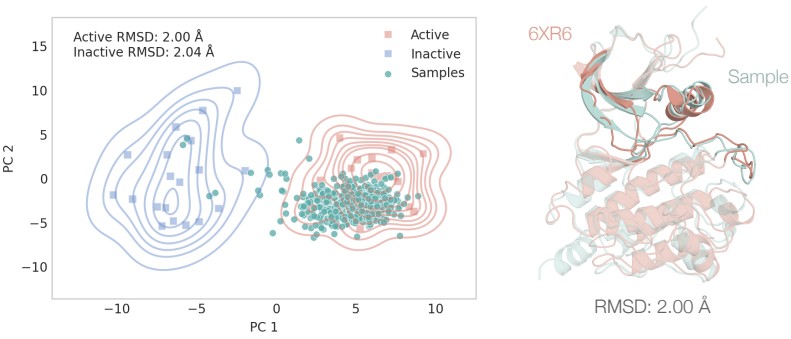

### Sample from active state (6XR6) to inactive state (6XR7)

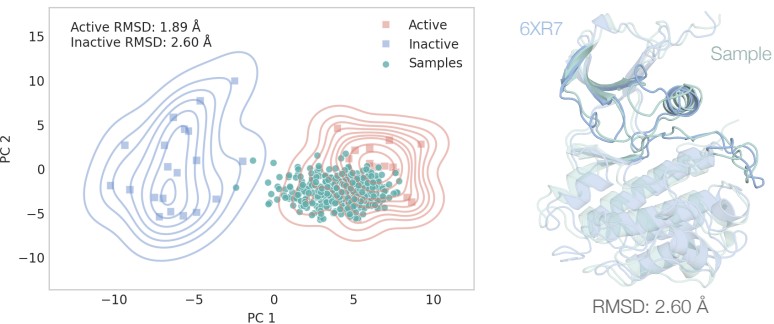

Figure 28: **Results on ABl Kinase sampled from the inactive state (top) and active state (bottom).** Left: PCA projection of reference conformations from the active and inactive PDB ensembles, with STAR-MD samples shown in green. Right: Superposed structures of the best sample and the corresponding target state, with core dynamic regions highlighted.

# Miscellaneous

## K    LLM Usage Disclosure

We utilized Large Language Models (LLMs) as a writing assistant during the preparation of this manuscript. For a few sections, our process involved authoring detailed outlines and initial drafts to establish the core scientific arguments and structure. Subsequently, we used LLMs to refine the verbiage, improve grammar, and enhance the overall clarity of the text in those specific sections. All scientific claims, experimental results, and theoretical assertions originated from the human authors, who take full responsibility for the final content of the paper, in accordance with ICLR 2026 policy.

