# OpenReview forum: "Scalable Spatio-Temporal SE(3) Diffusion for Long-Horizon Protein Dynamics"
_ICLR.cc/2026/Conference — ICLR 2026 Poster_

### Official Review · Reviewer_yvwX · 2025-10-28

**Soundness:** 3
**Presentation:** 3
**Contribution:** 3
**Rating:** 6
**Confidence:** 3

**Summary:**

This paper proposes STAR-MD, an autoregressive model for MD trajectories of protein systems. The key innovation is a causal diffusion transformer to predict the conditioning at a given frame using spatio-temporal attention in an efficient manner.  The model samples per-residue SE(3)-frames at each timestep via a diffusion process on SO(3), conditional on this conditioning vector. Experiments are performed on the ATLAS dataset, showing improved coverage of the conformational space as compared to previous methods, with significantly reduced memory requirements.

**Strengths:**

The model removes the Pairformer block present in existing protein models to save on memory. Instead, the model utilizes attention over the single-residue features across all frames and all residues. By clever training tricks such as feeding the clean and noisy frames together with a special attention mask, the training is parallelized over all frames, improving efficiency.
The experiments on long-horizon trajectory generation are good to see, and the model performs well in this challenging setting.

**Weaknesses:**

The empirical results in this paper are indeed impressive, but I would have liked to see a discussion and evaluation in identifying functional motions in proteins (eg. transitioning between active/inactive states, for example as represented by DFG in/out motifs in kinases. See https://www.rcsb.org/structure/6XR6 vs https://www.rcsb.org/structure/6XR7 for an example with Abl kinase.) Indeed, that is the main point of performing long MD simulations. ATLAS itself has very few functional motions, and tICA itself is not an ideal metric to extract these motions. That being said, I think these comments are more about the problem (trajectory generation) rather than this specific approach.

For the model itself, I would have liked to see generalization experiments to unseen lengths or sequences with low similarity to the training set, since that is a very common setting in practice.

**Questions:**

* Can you explain the structural noise aspect in the section “Contextual Noise Perturbation for Robust Rollouts”? Is it that you sample $τ ∼U[0.0, 0.1]$ and the corresponding noisy frame $x_\tau$ instead of keeping the clean history?
* Is the main reason for improved efficiency due to significantly lesser memory? The authors point out $O(N^2 L^2)$ for STAR-MD vs $O(N^3 L)$ for previous methods, but in the large $L$ limit, I would expect STAR-MD to become more inefficient?
* Please mention how the ATLAS test set was created (even if referring to prior work) since it helps put generalization results into context. Do you find that the model is able to generalize to new proteins with low sequence similarity?
* Please mention the inference costs of all models to enable a comparison to the ground truth MD.

---

> ### Author Response · Authors · 2025-11-22
> **Part 1**
>
> We thank the reviewer for their thoughtful review. We are glad they highlighted the efficient design of our model, noting its "significantly reduced memory requirements". We further appreciate their recognition that our model "performs well in this challenging setting" of long-horizon generation. We appreciate the suggestions regarding functional motions and generalization, which we address below.
>
> > W1. The empirical results in this paper are indeed impressive, but I would have liked to see a discussion and evaluation in identifying functional motions in proteins (eg. transitioning between active/inactive states, for example as represented by DFG in/out motifs in kinases. See [https://www.rcsb.org/structure/6XR6](https://www.rcsb.org/structure/6XR6) vs [https://www.rcsb.org/structure/6XR7](https://www.rcsb.org/structure/6XR7) for an example with Abl kinase.) Indeed, that is the main point of performing long MD simulations. ATLAS itself has very few functional motions, and tICA itself is not an ideal metric to extract these motions. That being said, I think these comments are more about the problem (trajectory generation) rather than this specific approach.
>
> We thank the reviewer for the suggestion. While our main contribution is the technical advancement over existing trajectory generation models, we agree that evaluating whether the model can capture functional motions is an important objective for these models.
>
> Following the reviewer's recommendation, in Appendix K.2 we assess STAR-MD on the active-inactive state transition of Abl kinase, which involves complex cooperative movements among several functional motifs. We simulate trajectories starting from both the active and inactive states to examine whether the model can capture the transition. Specifically, we generate 500 frames with a stride of 5.12 ns, corresponding to approximately 2.5 $\mu$s of simulation.
>
> We find that STAR-MD is able to sample from the inactive state to the active state, achieving a closest RMSD of 2.00 Å from the generated sample to the inactive reference structure (App K2 Fig. 27). When initialized from the active state, most samples remain within the active-state region, while a subset explores the inactive region. This observation is consistent with the expected energetic landscape of 88% active state and 8% inactive state reported in Xie et al [1]. The closest RMSD to the inactive reference is 2.60 Å, which is smaller than the average pairwise RMSD between active and inactive states (3.30 $\pm$ 0.34 Å).
>
> While we believe that more extensive experiments are required to fully characterize the model’s performance on functional protein dynamics, these preliminary results indicate that STAR-MD has promising capability in resolving complex and realistic conformational transitions.
>
> [1] Xie T, Saleh T, Rossi P, Kalodimos CG. Conformational states dynamically populated by a kinase determine its function. Science. 2020 Oct 9;370(6513):eabc2754. doi: 10.1126/science.abc2754. Epub 2020 Oct 1. PMID: 33004676; PMCID: PMC7920495.
>
> > W2. For the model itself, I would have liked to see generalization experiments to unseen lengths or sequences with low similarity to the training set, since that is a very common setting in practice.
>
> We agree that demonstrating generalization to unseen sequences and lengths is critical for practical utility. Our existing experimental design explicitly addresses these aspects:
>
> **Sequence Generalization:** We follow standard ATLAS splits based on temporal cutoffs, ensuring the test set represents unseen proteins (see Appendix C.1 for more details). We quantitatively verified that the test set has low sequence identity to the training set (average 19.6% ± 3.2%), with only two proteins exceeding 40% identity. To rigorously confirm generalization, we performed an additional analysis excluding these high-similarity cases (Appendix L), finding that model performance remains robust.
>
> **Length Generalization:** Our training data was restricted to proteins with $\le$ 384 residues, whereas the test set includes significantly larger systems (up to 724 residues). We have added a breakdown of performance by protein length in Appendix L, which demonstrates that STAR-MD generalizes well to these larger, unseen system sizes.
>
> **Horizon Generalization:** Furthermore, our long-horizon evaluations (240 ns and 1 $\mu$s) demonstrate generalization to longer timescales far beyond the original 100 ns time horizons seen in training.
>
> We have updated the manuscript to include these detailed breakdown analyses.

---

> ### Author Response · Authors · 2025-11-22
> **Part 2**
>
> > Q1. Can you explain the structural noise aspect in the section "Contextual Noise Perturbation for Robust Rollouts"? Is it that you sample $\tau \sim U[0.0, 0.1]$ and the corresponding noisy frame $x_\tau$ instead of keeping the clean history?
>
> Yes, the reviewer's understanding is correct, with the caveat that this procedure is applied both during training and inference to ensure consistency. More specifically:
>
> During training, instead of conditioning on clean history frames, we perturb the historical context frames $\mathbf{x}_{\<\ell}$ by applying the forward diffusion process with a small noise level $\tau \sim \mathcal{U}[0, 0.1]$ to obtain noisy contexts. The model is then trained to predict the current frame conditioned on this noisy history.
>
> Crucially, we apply this same perturbation strategy during inference: after generating a clean frame, we add noise at level $\tau$ before using it as context for subsequent generation steps. This ensures consistency between training and inference and makes the model robust to its own prediction errors, effectively preventing error accumulation during long rollouts. We have updated Section 3.3 to explicitly clarify this procedure.
>
> > Q2. Is the main reason for improved efficiency due to significantly lesser memory? The authors point out $O(N^2 L^2)$ for STAR-MD vs $O(N^3 L)$ for previous methods, but in the large L limit, I would expect STAR-MD to become more inefficient?
>
> We thank the reviewer for this insightful comment. We agree that our original complexity analysis only considered the Pairformer layer and did not account for the temporal attention mechanisms of the baselines. We have updated Section 3.2 and Appendix B.3 to provide a more rigorous per-layer complexity comparison.
>
> To answer the reviewer's question: the primary efficiency gain of STAR-MD arises from eliminating the cubic spatial complexity $\mathcal{O}(N^3 L)$ present in prior Pairformer-based models. Below, we consider two cases: (1) Pairformer + temporal attention on pairs and (2) Pairformer with temporal attention on singles.
>
> **Pairformer + temporal attention on pairs**: For (1), representative of ConfRover (Shen et al., NeurIPS 2025) which is the most competitive baseline, the per-layer complexity is $\mathcal{O}(N^3 L + N^2 L^2)$. Here, we have also accounted for the $O(N^2 L^2)$ complexity of pairwise temporal attention, which alone matches the full complexity of STAR-MD. The additional cubic spatial overhead of Pairformer makes this approach significantly more expensive with any practical protein size (e.g., $N>100$). Furthermore, the KV cache memory requirement for temporal attention on pairs is $\mathcal{O}(N^2 L)$, which becomes a bottleneck for long rollouts. This alone limits the feasible trajectory lengths that can be modeled, as we discuss in our sections 3.2 and 4.4.
>
> **Pairformer + temporal attention on singles**: For (2), which has a per-layer complexity of $\mathcal{O}(L N^3 + N L^2)$, the reviewer is correct that for $L \gg N$, our $\mathcal{O}(N^2L^2)$ scaling would eventually become less efficient than methods using single-residue temporal attention ($\mathcal{O}(NL^2)$), with the crossover point being at $L = \frac{N}{N-1} \approx N$. Practically, this has not been a limitation in our experiments, which have contained trajectory lengths up to 1000 frames (10 $\mu$s total at 10 ns stride) and proteins of size up to 724 residues.
>
> Finally, we highlight that within our evaluated baselines, ConfRover (with pair temporal attention) remains the only model that performs competitively across all aspects of evaluation (coverage, stability, dynamics), whereas methods using single-residue temporal attention either struggle with stability, coverage and dynamics, or all three. As such, while we acknowledge the theoretical scaling limitation in the extreme $L \gg N$ regime, we believe that STAR-MD currently offers the best practical trade-off between efficiency and performance.
>
> We hope this clarifies the efficiency considerations of STAR-MD.

---

> ### Author Response · Authors · 2025-11-22
> **Part 3**
>
> > Q3. Please mention how the ATLAS test set was created (even if referring to prior work) since it helps put generalization results into context. Do you find that the model is able to generalize to new proteins with low sequence similarity?
>
> We follow the reviewer's recommendation and have added details on ATLAS in Appendix C.1. As a summary, the ATLAS dataset contains structurally diverse proteins to be representative of different ECOD domains. We follow the prior work to split train/validation/test sets based on protein release time. We verified that 80/82 test proteins are highly dissimilar to any training proteins, with mean sequence identity of 19.6% ± 3.2%, while two proteins show high sequence identity > 80%.
> While the great majority of test cases with low sequence similarity to the training set can be sufficient to support generalization evaluation, we also include a stricter split that excludes the two highly similar proteins (7E2S-A and 7BUY-A), as reported in Appendix L.
> Additionally, to demonstrate robustness across different system sizes, we provide a performance breakdown by protein sequence length in Appendix L, showing consistent performance across all size buckets.
>
> > Q4. Please mention the inference costs of all models to enable a comparison to the ground truth MD.
>
> We thank the reviewer for the suggestion. We have included a comprehensive inference cost analysis in the revised Appendix E (Table 13), which we summarize below. Specifically, we evaluated the wall-clock time required to generate 100ns trajectories for each method, across proteins of varying lengths (51-724 amino acids). The cost of MD simulation is estimated using OpenMM running a 0.1 ns simulation with implicit solvent and extrapolating the cost to 100 ns. MDGen, trained on 100ns trajectories with a 400 ps timestep, can directly generate the target length in a single joint diffusion process, providing the greatest acceleration. AlphaFolding, trained on a fixed time step of 10 ps and window size of 16 frames, requires substantially longer inference time and encounters out-of-memory errors for the two largest proteins. All ML methods were evaluated on a single NVIDIA H100 GPU.
>
> Comparing wall-clock inference times requires nuance due to differing sampling constraints. Fixed-stride methods must generate trajectories at their specific training stride, often necessitating a larger number of frames (e.g., 10,000) to model the same physical duration. In contrast, variable-stride methods like STAR-MD and ConfRover can generate fewer frames (e.g., 80) by matching the benchmark stride directly. Furthermore, the number of denoising steps in diffusion models influences generation speed. Here, we report the total time to generate equivalent physical trajectories for all methods, with all other factors held at their default settings (e.g., we use the models' native stride and denoising steps as per their original implementations).

---

### Official Review · Reviewer_QE3i · 2025-10-28

**Soundness:** 4
**Presentation:** 3
**Contribution:** 3
**Rating:** 8
**Confidence:** 4

**Summary:**

This paper introduces STAR-MD, an autoregressive SE(3) diffusion model designed to simulate long-horizon protein dynamics. The authors identify two key bottlenecks in existing generative models: 1) poor computational/memory scaling (due to $\mathcal{O}(N^3)$ or $\mathcal{O}(N^2)$ pair-based operations) and 2) error accumulation that leads to instability in long rollouts.

STAR-MD's methodology tackles both problems directly. For Scalability, it uses a novel joint Spatio-Temporal (S$\times$T) attention that operates only on single-residue features, avoiding the expensive pair-feature bottleneck of competitors like ConfRover. For Stability, it employs a "contextual noise perturbation" (a form of diffusion forcing) that makes the model robust to its own small errors, preventing them from compounding over time.

The results are very strong. On the 100 ns ATLAS benchmark, STAR-MD achieves the best combination of structural validity and conformational coverage. More importantly, it is the first model to demonstrate stable, physically plausible generation out to the 1 microsecond timescale, a task where all baseline models (MDGen, ConfRover-W, AlphaFolding) fail catastrophically due to either memory limits or error accumulation.

**Strengths:**

1. The primary strength is the joint S$\times$T attention on singles-only features. This is a very clever architectural trade-off. It avoids the $\mathcal{O}(N^3)$ or $\mathcal{O}(N^2 L)$ complexity of competitors (AlphaFolding, ConfRover), which is the main barrier to long-horizon simulation. The KV cache analysis (Fig. 5) proves this is a massive practical win (e.g., 6.6MB vs 1.3GB per layer).

2. The "contextual noise perturbation" is the second key contribution. Autoregressive models are notoriously unstable in long rollouts. This method demonstrably fixes the problem. The stability plots (Fig. 3) and long-horizon results (Table 2) are direct proof that STAR-MD is robust to compounding errors.

3. The results are unambiguous. On the 100ns benchmark (Table 1), STAR-MD has the best balance of coverage and validity. On the 1µs benchmark (Table 2), it's the only model that works, maintaining 80% validity while competitors are unusable (e.g., MDGen 23% validity, AlphaFolding 0.06%).

4. The paper provides strong evidence that this is a trajectory simulator, not just an "ensemble generator." The long-horizon stability (Fig. 3) and kinetic fidelity metrics (tICA and RMSD-vs-lag) show that the model generates a temporally coherent and physically plausible "movie," not just a shuffled bag of good-looking frames.

**Weaknesses:**

1. The paper's greatest strength is also its biggest unproven assumption. The model trades physical explicitness (i.e., operating on pair features) for computational speed. It bets that its S$\times$T attention is powerful enough to implicitly learn all the complex, long-range pairwise interactions (like allostery) just from single-residue features. While this seems to work for the ATLAS proteins, it is a major leap of faith that this will hold for more complex systems defined by subtle, cooperative, long-range motions.

2. The "Dyn." metrics are a bit superficial. The tICA score is just a single number, and the paper's own Table 1 shows that the "AlphaFolding" baseline (which is 99.89% invalid) can get a "perfect" 0.17 tICA score. This "AlphaFolding Paradox" proves that the tICA metric alone is an insufficient measure of a good simulator. The RMSD/autocorrelation plots are good, but the paper would be much stronger if it included a more rigorous kinetic analysis (e.g., implied timescales from an MSM, VAMP-2 scores, or transition rate calculations).

3. For the long-horizon tests (Table 2), the ConfRover baseline had to be modified to a windowed variant (ConfRover-W) to manage its high memory requirements . While this strongly supports STAR-MD's superior scalability, it is not a direct comparison of kinetic fidelity, as the baseline is architecturally constrained by a limited temporal window. The AlphaFolding baseline appears to be a non-competitive comparator. With a reported structural validity of only 0.11% (Table 1), this model is fundamentally unsuited for this task . Its "perfect" tICA score (0.17) serves primarily to demonstrate the insufficiency of the tICA metric in isolation, rather than providing a meaningful performance benchmark.

**Questions:**

1. Can the authors please comment on the "singles-only" trade-off? How confident are they that this architecture can capture complex, cooperative phenomena like allostery, which are fundamentally defined by long-range pairwise correlations? Have they considered testing STAR-MD on a known allosteric benchmark protein to validate this?

2. The validation of conformational coverage (JSD/Recall) is done in the PCA space of the reference trajectory. This metric is a good measure of ensemble overlap but does not validate the kinetic pathways or transition probabilities. A model could, in theory, match this distribution without being a good simulator. Could the authors comment on this choice and why a more kinetically-focused metric (like implied timescales) was not also used?

3. The tICA metric details are sparse. Given the "AlphaFolding Paradox" (where a 0.11% valid model gets a perfect tICA score), this metric needs more justification. What featurization, lag time, and number of components were used?

4. Related to the AlphaFolding baseline: the 0.11% validity is shockingly low. Can the authors confirm their setup for this baseline? It's hard to believe this is a representative result for that model, and it makes the comparison less meaningful.

---

> ### Author Response · Authors · 2025-11-22
> **Part 1**
>
> We thank the reviewer for their enthusiastic support and for stating that our "results are very strong" and "unambiguous." We are happy they recognized our "very clever architectural trade-off" as a "massive practical win" for scalability. We further appreciate the reviewer's recognition that STAR-MD is the "first model to demonstrate stable... generation out to the 1 microsecond timescale," validating it as a "trajectory simulator." We address the reviewer's thoughtful questions regarding the "singles-only" trade-off and kinetic metrics below.
>
> > W1. The paper's greatest strength is also its biggest unproven assumption. The model trades physical explicitness (i.e., operating on pair features) for computational speed. It bets that its S x T attention is powerful enough to implicitly learn all the complex, long-range pairwise interactions (like allostery) just from single-residue features.
> >
> > [...]
> >
> > Q1. Can the authors please comment on the "singles-only" trade-off?
>
> We thank the reviewer for this insightful comment. We agree that capturing long-range cooperative phenomena is critical for modeling complex biomolecular systems.
>
> Regarding the "singles-only" concern, we would like to clarify that STAR-MD explicitly maintains and updates pairwise features throughout the network via 1) the input OpenFold emeddings, which contain both single and pair features, 2) the `FrameEncoder` layer at the input, and 3) the `EdgeTransition` layer (a lightweight MLP layer from AlphaFold2 that updates pair features based on the current single-residue representations) after each spatio-temporal attention block. The critical architectural changes we make are:
>
> 1. We remove the heavy Pairformer "trunk" (standard in AlphaFold-like models). Instead, we use `InvariantPointAttention` and `EdgeTransition` layers to update single and pair features in each spatio-temporal attention block, avoiding the prohibitive $\mathcal{O}(N^3)$ spatial complexity.
> 2. We limit spatio-temporal attention to single-residue features. This avoids the cost of performing temporal attention over pairs (as done in ConfRover), which creates a massive memory bottleneck ($\mathcal{O}(N^2 L)$ for ConfRover's KV cache vs. $\mathcal{O}(N L)$ for STAR-MD).
>
> By relying on the joint spatio-temporal attention on singles to drive dynamics while updating pairs via efficient local operations, we achieve a balance that allows for much longer context windows and larger systems.
>
> We hope this clarifies the design choices made in STAR-MD. We have added a new section in Appendix F to explicitly describe these mechanisms and their motivations. We have also updated section 3.1 to make these points clearer in the main text. We thank the reviewer for prompting us to clarify this important aspect of our architecture.
>
> > W1. [...] While this seems to work for the ATLAS proteins, it is a major leap of faith that this will hold for more complex systems defined by subtle, cooperative, long-range motions.
> >
> > [...]
> >
> > Q1. [...] How confident are they that this architecture can capture complex, cooperative phenomena like allostery, which are fundamentally defined by long-range pairwise correlations? Have they considered testing STAR-MD on a known allosteric benchmark protein to validate this?
>
> To test whether STAR-MD is capable of modeling complex cooperative movements, we include several case studies on function-related dynamics (Appendix K), including complex movements and cryptic-pockets with allosteric binding effects.
> These case studies demonstrate that STAR-MD successfully samples the bound state (holo) from an unbound state for all three cases and demonstrate promising results on sampling the active and inactive states of a kinase, providing empirical evidence on STAR-MD capturing conformational changes beyond local dynamics.
>
> Finally, in addition to these empirical results, we invite the reviewer to consult the new Section 3.4, which provides a theoretical justification for our modeling choices based on the Mori-Zwanzig formalism.
>
> We thank the reviewer for suggesting these experiments.

---

> ### Author Response · Authors · 2025-11-22
> **Part 2**
>
> > W2. The "Dyn." metrics are a bit superficial. The tICA score is just a single number, and the paper's own Table 1 shows that the "AlphaFolding" baseline (which is 99.89% invalid) can get a "perfect" 0.17 tICA score. This "AlphaFolding Paradox" proves that the tICA metric alone is an insufficient measure of a good simulator. The RMSD/autocorrelation plots are good, but the paper would be much stronger if it included a more rigorous kinetic analysis (e.g., implied timescales from an MSM, VAMP-2 scores, or transition rate calculations).
> >
> > [...]
> >
> > Q2. The validation of conformational coverage (JSD/Recall) is done in the PCA space of the reference trajectory. This metric is a good measure of ensemble overlap but does not validate the kinetic pathways or transition probabilities. A model could, in theory, match this distribution without being a good simulator. Could the authors comment on this choice and why a more kinetically-focused metric (like implied timescales) was not also used?
>
> We thank the reviewer for this critical assessment of our kinetic metrics.
>
> The tICA correlation metrics were used to maintain consistency with ConfRover's ATLAS evaluation procedure. We agree with the reviewer that more rigorous kinetic analysis would be beneficial.
>
> Therefore, as suggested, we have computed VAMP-2 scores as a function of lagtime and included them in the Appendix C.4.1 and Figure 10.
> We observe that the results from the VAMP-2 analysis are consistent with our autocorrelation findings, that STAR-MD better approximates the level of dynamics of MD references.
>
> We have updated our main tables to include three new scalar metrics: the Average Difference to MD Reference for RMSD, Autocorrelation, and VAMP-2 curves. These metrics condense the information from the lag-time plots (Figure 6) into single numerical values, providing a direct measure of how well the model preserves the temporal evolution statistics of the ground truth MD. We have also moved the autocorrelation plot to the main text (Figure 2a) to highlight its importance.
>
> Finally, we have also updated the tICA correlation metric to only consider pairs that involve structurally valid frames (see Appendix C.3 for clarification). This should improve the reliability of this metric.
>
> Regarding implied timescales from Markov State Models (MSMs), we did not use them because stable estimation generally requires the simulation to reach equilibrium. Since our ATLAS benchmark consists of 100 ns trajectories of non-equilibrium structures, MSMs are difficult to construct reliably in this setting.

---

> ### Author Response · Authors · 2025-11-22
> **Part 3**
>
> > Q3. The tICA metric details are sparse. [Given the "AlphaFolding Paradox" (where a 0.11% valid model gets a perfect tICA score), this metric needs more justification.] What featurization, lag time, and number of components were used?
>
> Our tICA evaluation follows the protocol of ConfRover (Shen et al., 2025), using flattened $C_\alpha$ coordinates as features, lagtimes $\tau \in [1, 5, 10, 20]$, and the top 2 independent components to compute the Pearson correlation of per-residue contribution scores. To address the validity concern, we refine this procedure to explicitly filter for structurally valid time-lagged pairs, ensuring that the metric reflects the dynamics of physically plausible conformations. We have added these full implementation details in Appendix C.3.
>
> > W3. For the long-horizon tests (Table 2), the ConfRover baseline had to be modified to a windowed variant (ConfRover-W) to manage its high memory requirements . While this strongly supports STAR-MD's superior scalability, it is not a direct comparison of kinetic fidelity, as the baseline is architecturally constrained by a limited temporal window.
>
> The modification to a windowed variant (ConfRover-W) was necessitated by the prohibitive memory cost of the full ConfRover model for long trajectories. For the long-horizon baselines (240 ns and 1 $\mu$s), as explained in Section 4.3, the standard ConfRover model exceeded hardware memory limits on our H100 GPUs. Before settling on the windowed variant, we attempted other strategies such as CPU-offloaded KV caches, but these also hit system memory limits (even with a batch size of 1).
>
> It is important to note that ConfRover-W is a non-trivial baseline. We employed attention sink tokens (Xiao et al., 2023), which we found to be critical; without them, the windowed model suffered from extremely high error accumulation and instability in long rollouts. We have updated Appendix M with more details on ConfRover-W's setup and performance. To quantify the impact of windowing on fidelity, we compared the full ConfRover against ConfRover-W on the 100 ns benchmark, where both models fit in memory. Our results indicate that ConfRover-W performs on-par with the full ConfRover in terms of coverage and dynamics, albeit with a slight drop in stability. This suggests that while windowing does impose some limitations, ConfRover-W remains a strong baseline for long-horizon comparisons. We have added these additional analyses and comparisons to Appendix M.
>
> > Q4. Related to the AlphaFolding baseline: the 0.11% validity is shockingly low. Can the authors confirm their setup for this baseline? It's hard to believe this is a representative result for that model, and it makes the comparison less meaningful.
>
> We thank the reviewer for raising this concern. We re-examined our implementation of AlphaFolding to ensure correctness. While we identified a checkpoint loading mistake for our 100 ns results, fixing this issue did not improve the performance significantly. The model still exhibits extremely low conformation validity at 0.52%. We have updated Table 1 with the corrected 100 ns results. We have confirmed that our original setups for the 240 ns and 1 $\mu$s benchmarks were correct.
>
> To further understand the cause, we conducted a more detailed analysis. We found that AlphaFolding has extremely low all-atom (AA) validity due to side-chain rotamer outliers and its CA validity drops quickly during the block extension process for generating long trajectories:
>
> 1. When examining the AA validity within the first 16 frames (before subsampling, 10 ps interval), we found that although only 0.2% of samples violate the Ramachandran criteria (i.e., good backbone dihedral geometry), 96.7% of samples violate the rotamer criteria, with an average rotamer outlier rate of 12.99% ± 3.82%. This indicates the poorer side-chain rotational predictions, a quality aspect often overlooked in prior analyses.
> 2. For CA validity, we computed the average validity in each extension windows. We observed reasonable quality in the initial generation window, but a rapid degradation starting from the first extension. This indicates that AlphaFolding's extension mechanism struggles to maintain structural integrity over long horizons and leads to significant error accumulation.
>
>     | Extension window (16 frames) | 0            | 1            | 2            | 3            |
>     | :--------------------------- | :----------- | :----------- | :----------- | :----------- |
>     | CA-Val                       | 70.3% ± 7.1% | 30.0% ± 3.9% | 21.3% ± 2.0% | 16.7% ± 2.1% |
>
> We thank the reviewer again for pointing out this issue and helping us better understand the behavior of the baseline model. We have updated Appendix C.2 with additional details on how we set up the AlphaFolding experiments for our benchmarking, including key arguments such as `sample_step`, `n_motion`, `n_frames`, etc.

---

> > ### Comment · Reviewer_QE3i · 2025-11-26
> >
> > I thank the authors for their detailed response and the inclusion of the new experiments.
> >
> > The clarification regarding the EdgeTransition layer effectively addresses my concern about the "singles-only" architecture by explaining how pairwise information is preserved and updated locally. Furthermore, the new case studies in Appendix K (specifically on kinases and cryptic pockets) provide strong empirical evidence that the model can capture complex, long-range allosteric effects despite the architectural trade-offs.
> >
> > These additions significantly strengthen the manuscript and fully address my previous concerns. I maintain my score of 8.

---

### Official Review · Reviewer_jt5g · 2025-10-29

**Soundness:** 3
**Presentation:** 2
**Contribution:** 3
**Rating:** 6
**Confidence:** 4

**Summary:**

This paper introduces STAR-MD, a new approach to modeling temporal protein dynamics, focusing on long-term consistency, using generative diffusion models. To this end, the authors use a scalable residue frame representation, following previous work. However, this corresponds to a coarse-grained representation compared to the fully atomistic molecular dynamics simulations that are used to train the model. Based on this observation, it is possible to show with Mori-Zwanzig theory that an accurate modeling of the temporal dynamics in the coarse representation requires the incorporation of memory (opposed to a fully Markovian approach of some prior works) as well as complex spatial-temporal correlations. Therefore, the authors introduce an architecture that not only leverages a single previous frame but a context window of several frames, thereby capturing the system's history as required by Mori-Zwanzig theory. Moreover, they use explicit joint temporal-spatial attention layers to also capture the necessary complex correlations. That aside, the authors leverage established architecture modules, like invariant point attention and pair representations. Conceptually, the approach can be considered related to methods in the video generation literature. The authors then validate their method and outperform prior baselines. The simulated protein dynamics trajectories are scored in terms of structural quality, conformational coverage and dynamic fidelity. In particular for long roll-outs the method performs better than baselines, avoiding major drift and error accumulation.

**Strengths:**

- **Clarity**: Overall the paper is easy to follow (however, I believe the motivation for the approach could be explained better, see weaknesses below).
- **Originality:** The paper makes a novel and original contribution in the area of protein dynamics by explicitly modeling long context windows and using joint spatio-temporal attention with conditioning noise to achieve long-term temporal generative modeling of molecular dynamics trajectories. However, the different components can be found elsewhere in the literature, e.g. in video generation, and the approach is therefore not overly creative from an architecture perspective. This is not a major concern, though, as the results seem mostly strong. On the other hand, the coarse-graining and Mori-Zwanzig formalism perspective is novel in the context of generative method design and interesting (but this could be presented better, see comments below).
- **Experimental results:** Overall, the paper shows satisfactory experimental results, outperforming baselines. In particular the long-horizon results are strong.
- **Significance**: Accelerating and emulating molecular dynamics simulation via generative methods is an important research topic with impact in biology, physics and chemistry. The paper makes a non-negligible step in this direction, and therefore the contribution can be considered significant.

**Weaknesses:**

- **Presentation of motivation:** In the main text, the authors argue without much explanation that long context and complex spatio-temporal modeling is necessary to accurately model molecular dynamics trajectories. Naively, it will be difficult to understand for most readers why that is, given that the original atomistic molecular dynamics data is essentially Markovian, corresponding to a simple integration of equations of motion (possibly with a thermostat/barostat). The reasoning for this is only explained in more depth in the appendix, when looking at the problem through the lens of coarse graining and Mori Zwanzig theory, which many readers won't know. Looking at the problem of generative modeling of molecular dynamics from this perspective is very interesting and novel, yet this is only "hidden" in the appendix. I would suggest the authors to restructure the paper and explain more of this in the main paper. Meanwhile, the diffusion formalism and other basics are completely standard and could be moved to the appendix. This would improve the paper, I believe.
- **Results in Figure 2:** In Figure 2, right, the MD trajectory of STAR-MD covers more phase space than the baselines, but it still never explores the mode on the right hand side (high PC1, low PC2 values). While the method is evidently better than the baselines, it is still unclear how well STAR-MD explores the configurational state space.
- **Configurational sampling and baselines:** More generally, long-horizon molecular dynamics, the core problem tackled in this paper, is meant to explore the full phase space. I believe the authors should study this in more depth and compare to other methods that sample molecular conformations. For example, how does STAR-MD compare to BioEmu (https://www.science.org/doi/10.1126/science.adv9817) in that regard, one of the most well-known and prominent methods to sample similar molecular systems (proteins)? Can we have full phase space diagrams for different test proteins and compare (e.g. the fast folding test proteins used in BioEmu)? The calculated coverage metrics are nice, but not very interpretable. I would expect the method to show conformational coverage similar to MD to be useful in practice -- this is a critical aspect.
- **Free Energy Calculations:** Also related to the above, does the method enable free energy calculations? For instance of conformational transitions? Free energy calculation is one of the most important tasks of molecular dynamics. Hence, it would be good to know if STAR-MD can be used for such tasks, and it would be great to add results on this.

**Questions:**

In section 4.4, the paper generates trajectories with different strides. How far can this be pushed? The model is trained with a stride length up to 10ns (section 3.3). What happens if we generated data with this maximum stride length and use ~850 frames/steps, as in the other experiments? What simulation time could we reach without error accumulation? Shouldn't this improve coverage and phase space exploration?

That aside, I don't have further questions, but some typos:
- line 036: integrate -> integrates
- line 107: embeddings -> embedding's?
- line 124: blocks -> block

And a suggestion: In Figure 2, the two MD reference curves are almost exactly on top of each other - I initially thought a curve is missing. I would strongly suggest the authors to address this and fix the visualization so it's clear there are two curves on top of each other.

---

> ### Author Response · Authors · 2025-11-22
> **Part 1**
>
> We thank the reviewer for their thoughtful review and for highlighting the "novel and original contribution" of our work in introducing joint spatio-temporal modeling to protein dynamics. We are pleased that the reviewer highlighted our strong experimental results, particularly in the long-horizon setting, recognizing our work as significant in this field. We address the concerns regarding motivation and phase space exploration below.
>
> > W1. **Presentation of motivation:** In the main text, the authors argue without much explanation that long context and complex spatio-temporal modeling is necessary to accurately model molecular dynamics trajectories. Naively, it will be difficult to understand for most readers why that is, given that the original atomistic molecular dynamics data is essentially Markovian, corresponding to a simple integration of equations of motion (possibly with a thermostat/barostat). The reasoning for this is only explained in more depth in the appendix, when looking at the problem through the lens of coarse graining and Mori Zwanzig theory, which many readers won't know. Looking at the problem of generative modeling of molecular dynamics from this perspective is very interesting and novel, yet this is only "hidden" in the appendix. I would suggest the authors to restructure the paper and explain more of this in the main paper. Meanwhile, the diffusion formalism and other basics are completely standard and could be moved to the appendix. This would improve the paper, I believe.
>
> We appreciate the reviewer's insight and agree that the Mori-Zwanzig theoretical perspective provides valuable motivation for our architectural choices. In response to this suggestion, we have added a new section (Section 3.4) in the main paper that presents the coarse-graining perspective and explains how the non-Markovian nature of coarse-grained dynamics necessitates long-context spatio-temporal modeling.
> More specifically, this section aims to clarify 1) why temporal modeling is critical for coarse-grained dynamics and 2) why further coarse-graining from singles and pairs onto singles yields a memory kernel that exhibits non-separable spatio-temporal coupling, thereby motivating our joint spatio-temporal attention mechanism.
> We believe this restructuring better highlights the novel methodological contribution of our work and improves accessibility for readers unfamiliar with Mori-Zwanzig theory. We sincerely thank the reviewer for this constructive suggestion and believe it has significantly strengthened the manuscript.
>
> ---
>
> > W2. **Results in Figure 2:** In Figure 2, right, the MD trajectory of STAR-MD covers more phase space than the baselines, but it still never explores the mode on the right hand side (high PC1, low PC2 values).
>
> We thank the reviewer for this keen observation. We agree that exploring the full phase space is the ultimate goal for conformation sampling, but we found it is important to contextualize STAR-MD's performance against the nature of the ground truth data.
>
> To illustrate this, we have updated Figure 2 to include each individual MD reference run. No single MD trajectory covers all three modes simultaneously; rather, the full reference distribution is an aggregate of three distinct MD simulations (MD 1, MD 2, MD 3), each exploring different regions of the energy landscape. Consequently, the fact that a single STAR-MD trajectory does not cover all modes is consistent with the behavior of individual ground truth MD trajectories.
>
> > While the method is evidently better than the baselines, it is still unclear how well STAR-MD explores the configurational state space.
>
> To provide a more comprehensive assessment of conformational coverage beyond the single example in Figure 2, we have randomly sampled 10 additional proteins from the ATLAS dataset and generated similar phase space coverage plots. These additional results, now included in the Appendix I Fig 14-23, demonstrate that STAR-MD consistently captures significant portions of the phase space, often outperforming baselines, even if single trajectories do not always capture the full multimodal distribution.

---

> ### Author Response · Authors · 2025-11-22
> **Part 2**
>
> > W3. **Configurational sampling and baselines:** More generally, long-horizon molecular dynamics, the core problem tackled in this paper, is meant to explore the full phase space. I believe the authors should study this in more depth and compare to other methods that sample molecular conformations. For example, how does STAR-MD compare to BioEmu ([https://www.science.org/doi/10.1126/science.adv9817](https://www.science.org/doi/10.1126/science.adv9817)) in that regard, one of the most well-known and prominent methods to sample similar molecular systems (proteins)? Can we have full phase space diagrams for different test proteins and compare (e.g. the fast folding test proteins used in BioEmu)? The calculated coverage metrics are nice, but not very interpretable. I would expect the method to show conformational coverage similar to MD to be useful in practice -- this is a critical aspect.
>
> We thank the reviewer for this constructive suggestion. We agree that comparing our model to state-of-the-art conformational samplers provides important context for our work. In response, we conducted additional experiments comparing STAR-MD with BioEmu on two benchmarks: (1) the standard ATLAS ensemble benchmark , and (2) the CATH1 ensemble benchmark introduced in BioEmu. (We did not include the fast-folding benchmark because it requires training separate models exclusively on that dataset, as implemented in BioEmu's paper).
> We provide the full results in Appendix J and summarize the key findings below:
>
> 1. STAR-MD outperforms BioEmu on the ATLAS benchmark, showing better alignment in diversity, more accurate modeling of both global and local flexibility, and higher similarity to the reference ensemble at both distributional and ensemble-observable levels. Please refer to Table 15 in Appendix J, also shown below.
>
>     **Table: ATLAS ensemble benchmark results**
>
>     |                               | STAR-MD | BioEmu |
>     | :---------------------------- | ------: | -----: |
>     | RMSF (MD Ref = 1.63)          |    1.38 |   2.87 |
>     | Pairwise RMSD (MD Ref = 2.76) |    2.34 |   4.51 |
>     | Pairwise RMSD r (↑)           |    0.52 |   0.33 |
>     | Global RMSF r (↑)             |    0.56 |   0.52 |
>     | Per target RMSF r (↑)         |    0.86 |   0.82 |
>     | RMWD (↓)                      |    2.85 |   4.33 |
>     | MD PCA W2 (↓)                 |     1.5 |   1.64 |
>     | Joint PCA W2 (↓)              |    2.33 |   3.43 |
>     | PC sim > 0.5 % (↑)            |    35.4 |   26.8 |
>     | Weak contacts J (↑)           |    0.52 |   0.49 |
>     | Transient contacts J (↑)      |    0.37 |   0.37 |
>     | Exposed residue J (↑)         |    0.57 |   0.54 |
>     | Exposed MI matrix rho (↑)     |    0.24 |   0.26 |
>
> 2. On CATH1, which contains longer microsecond-scale simulations for smaller proteins (<75 residues), BioEmu shows higher diversity and coverage (refer to Figures 24 and 25 in Appendix J), as well as slightly more accurate free-energy estimation, compared to STAR-MD. Below, we have provided the MAE results for free energy estimation on CATH1.
>
>     **MAE on free energy estimation**
>
>     | BioEmu        | STAR-MD       | STAR-MD-iid   |
>     | :------------ | :------------ | :------------ |
>     | 0.838 ± 0.182 | 1.001 ± 0.273 | 0.947 ± 0.275 |
>
> To determine whether the performance gap is due to our autoregressive trajectory generation approach, we also evaluated STAR-MD with unconditional generation (i.e., generating single frames without temporal conditioning) for IID samples, mirroring BioEmu's sampling setup. The IID-sampled results are similar to the trajectory-based results, suggesting that the gap does not arise from the autoregressive mechanism itself, but is likely attributable to differences external to the architecture.
>
> As an example of those differences, BioEmu benefits from large-scale pretraining on the AlphaFold Database (AFDB) and fine-tuning on extensive MD trajectories, including the long CATH-domain simulations. In contrast, our current STAR-MD model is trained on a significantly smaller portion of the ATLAS dataset, as our primary goal in this work is to validate the core architectural contributions (e.g., Joint Spatio-Temporal Attention and Contextual Noise).
>
> Overall, these findings suggest that while STAR-MD's architecture is highly effective within its training domain (e.g., ATLAS), scaling the training data will likely be important for matching the broader generalization and diversity demonstrated by models such as BioEmu. We have incorporated the new comparative results and an expanded discussion of these factors in Appendix J of the revised manuscript.

---

> ### Author Response · Authors · 2025-11-22
> **Part 3**
>
> > W4. **Free Energy Calculations:** Also related to the above, does the method enable free energy calculations? For instance of conformational transitions? Free energy calculation is one of the most important tasks of molecular dynamics. Hence, it would be good to know if STAR-MD can be used for such tasks, and it would be great to add results on this.
>
> STAR-MD supports generating conformational ensembles that can be used for free-energy estimation. We illustrate this capability in our experiments on BioEmu's CATH1 dataset, with detailed results provided in the response to the previous question.
>
> ---
>
> > Q1. In section 4.4, the paper generates trajectories with different strides. How far can this be pushed? The model is trained with a stride length up to 10ns (section 3.3). What happens if we generated data with this maximum stride length and use approximately 850 frames/steps, as in the other experiments? What simulation time could we reach without error accumulation? Shouldn't this improve coverage and phase space exploration?
>
> We thank the reviewer for proposing this stress test. To address this, we pushed the model to generate 10 $\mu$s trajectories (1000 steps at $\sim$10 ns stride) for the ATLAS test set. We report the structural quality metrics for these additional long-horizon rollouts in Appendix G.
>
> **Regarding stability:** As demonstrated in figure 12, the model remains robust. We observe that the generated structures remain valid and physically plausible throughout the full 10 $\mu$s duration, demonstrating that STAR-MD does not suffer from catastrophic error accumulation even at 100x the training horizon.
>
> **Regarding coverage:** While running longer simulations should theoretically explore more phase space, quantitatively verifying this improvement consistently is difficult because we lack ground truth MD simulations at the 10 $\mu$s scale for these proteins. It is also an open question whether the dynamics present in 100 ns data can be extrapolated to reflect those of microsecond scales and whether a data-driven model can learn this extrapolation.
>
> Therefore, while we demonstrate the necessary prerequisite for better exploration (stability over microsecond timescales), we caution against making strong claims about improved phase space coverage without corresponding ground truth validation. Ultimately, we believe scaling up training data---similar to the approach of BioEmu---is the most promising path toward unlocking true long-horizon exploration in future work.
>
> > That aside, I don't have further questions, but some typos:
> >
> > - line 036: integrate -> integrates
> > - line 107: embeddings -> embedding's?
> > - line 124: blocks -> block
>
> We thank the reviewer for catching these typos. We have corrected them in the revised manuscript.
>
> > And a suggestion: In Figure 2, the two MD reference curves are almost exactly on top of each other - I initially thought a curve is missing. I would strongly suggest the authors to address this and fix the visualization so it's clear there are two curves on top of each other.
>
> We have updated Figure 2 (left) and Figure 7 in the revised manuscript to improve the visualization of the two MD reference curves by using distinct transparency levels and line styles. This change makes it clearer that both curves are present and overlapping. We appreciate the reviewer's attention to this detail.

---

> > ### Comment · Reviewer_jt5g · 2025-11-27
> > **Thank you.**
> >
> > I would like to thank the authors for replying to my questions and addressing my concerns. I recommend acceptance and raised my score accordingly.

---

### Official Review · Reviewer_Qv5B · 2025-10-31

**Soundness:** 4
**Presentation:** 3
**Contribution:** 4
**Rating:** 8
**Confidence:** 3

**Summary:**

This paper introduces STAR-MD, a spatio-temporal SE(3)-equivariant diffusion model designed for learning protein molecular dynamics directly from Cartesian coordinate trajectories. The model represents each residue as a rigid frame (translation + rotation) and predicts its time evolution conditioned on physical stride Δt, trained via diffusion-style denoising. It is evaluated on the ATLAS dataset and compared against baselines including MDGen, ConfRover, and AlphaFolding-4D. The authors report that STAR-MD reproduces MD-like short-horizon dynamics (RMSD vs lagtime, tICA correlations), achieves high backbone validity, and maintains stability over extended rollouts (up to 1 μs). Ablation studies test variants without diffusion noise and without joint spatio-temporal attention.

This is a technically solid and promising paper whose model is likely to influence future ML-based coarse-grained MD approaches. However, the evaluation would be much stronger with explicit clarification of (i) representational scale differences, (ii) alignment and Δt normalization procedures, and (iii) statistical variability across rollouts.
The current comparisons convincingly show stability, but not necessarily accuracy, and the role of coarse-graining deserves explicit discussion.

**Strengths:**

1. Clear technical contribution: Combines SE(3)-equivariant frame modeling and continuous-time conditioning in a diffusion framework which conceptually elegant and well-motivated.
2. Strong empirical results: STAR-MD substantially outperforms prior coordinate-based models in conformational coverage, dynamic fidelity, and stability over long horizons.
3. Method clarity (core model): The paper’s architectural description and use of per-residue SE(3) frames are well explained.
4. Long-horizon evaluation: Extending rollouts to hundreds of nanoseconds or microseconds is ambitious and valuable for assessing numerical stability.
5. Well-integrated metrics: Use of RMSD, tICA, and validity metrics ties the work to established MD analysis practices.

**Weaknesses:**

1. Comparative clarity. It’s difficult to realize from the text that the baselines use very different geometric representations: AlphaFolding-4D is all-atom, MDGen and ConfRover are backbone-level, and STAR-MD is Cα-only.
Because this difference likely affects both stability and metric scale, the paper should state it explicitly and discuss how Cα-level evaluation influences comparisons.

2. Alignment and Δt normalization. The paper does not describe how trajectories are aligned prior to RMSD/PCA/tICA computation, nor how variable Δt conditioning was reconciled with baselines’ fixed strides.

3. Single-trajectory visuals. Figures 2–3 show single rollouts; ensemble variability or error bars would make coverage and stability claims more convincing.

4. Unexplained shading in Fig. 3. The caption omits what the shaded bands represent (presumably inter-protein variation).

5. Minimal ablation scope. Only “w/o noise” and “w/ separate attention” are tested. These yield small metric changes, so it remains unclear which components truly drive the reported improvements.

**Questions:**

1. Role of representation. The results suggest that representational scale (Cα vs. backbone vs. all-atom) may contribute more to stability than architectural details. Do the authors have evidence or experiments that could separate representational effects from architectural ones?
2. Alignment. Were trajectories aligned (e.g., via Cα superposition) before computing RMSD and PCA/tICA metrics, and if so how?
3. Δt comparability. How were lag times normalized when comparing models with variable vs. fixed strides?
4. Shading in Fig. 3: What statistic do the shaded bands represent? Standard deviation, percentile range, or something else?
4. Long-horizon ablations. Do the “w/o noise” and “w/ sep attn” variants remain stable in 240 ns – 1 µs rollouts?

---

> ### Author Response · Authors · 2025-11-22
> **Part 1**
>
> We thank the reviewer for their constructive feedback and for recognizing our work as a "technically solid and promising paper" that is "likely to influence future ML-based coarse-grained MD approaches." We are glad the reviewer found our autoregressive diffusion framework "conceptually elegant" and appreciated our "strong empirical results" in "substantially outperform[ing]" prior models on long-horizon tasks. We value the reviewer's detailed suggestions regarding comparative clarity and evaluation, which we address below.
>
> ---
>
> > W1. Comparative clarity. It's difficult to realize from the text that the baselines use very different geometric representations: AlphaFolding-4D is all-atom, MDGen and ConfRover are backbone-level, and STAR-MD is C$\alpha$-only. Because this difference likely affects both stability and metric scale, the paper should state it explicitly and discuss how C$\alpha$-level evaluation influences comparisons.
> >
> > [...]
> >
> > Q1. Role of representation. The results suggest that representational scale (C$\alpha$ vs. backbone vs. all-atom) may contribute more to stability than architectural details. Do the authors have evidence or experiments that could separate representational effects from architectural ones?
>
> To clarify, STAR-MD utilizes SE(3) rigid residue frames (translation + rotation) to represent the backbone coordinates and torsional angles for side-chain atoms, the same as the baseline models MDGen, ConfRover, and AlphaFolding-4D, placing all three models at comparable representational granularity. We have updated Section 4.1 (Implementations) to explicitly state that these baselines share this identical geometric representation, ensuring that comparisons reflect architectural differences rather than representational scale. We agree that this detail is important and thank the reviewer for helping us clarify it.
>
> ---
>
> > W2. Alignment and $\Delta t$ normalization. The paper does not describe how trajectories are aligned prior to RMSD/PCA/tICA computation.
> >
> > [...]
> >
> > Q2. Alignment. Were trajectories aligned (e.g., via C$\alpha$ superposition) before computing RMSD and PCA/tICA metrics, and if so how?
>
> Yes, all trajectories were aligned to the first frame of the reference trajectory using C$\alpha$ superposition before computing RMSD, PCA, and all other metrics. This alignment step removes effects from global translations and rotations. We have updated the experimental setup (Section 4.1) to clarify this point. We have also updated the appendix (Appendix C.3) with more details on the alignment procedure.
>
> ---
>
> > W2. The paper does not describe [...] how variable $\Delta t$ conditioning was reconciled with baselines' fixed strides.
> >
> > [...]
> >
> > Q3. $\Delta t$ comparability. How were lag times normalized when comparing models with variable vs. fixed strides?
>
> For variable-stride models (STAR-MD, ConfRover), we generated trajectories directly at the target intervals using their time-conditioning mechanism. For fixed-stride models (MDGen, AlphaFolding), we generated trajectories at their native training resolution and subsampled them to match the target evaluation intervals. This ensures that all comparisons are made on identical physical timescales. We have updated Section 4.1 to explicitly describe this normalization procedure.
>
> ---
>
> > W3. Single-trajectory visuals. Figures 2-3 show single rollouts; ensemble variability or error bars would make coverage and stability claims more convincing.
>
> To clarify, the original Figures 2(a) and 3 displayed single rollout results averaged over the full test sets. We agree that visualizing ensemble variability strengthens the claims of stability. In the revised manuscript, we have updated Figure 3 to show multiple rollouts (5 seeds) for STAR-MD and the baseline models (with AlphaFolding-4D still pending due to resource limitations, which we hope to finish by the end of the discussion period). The shaded error bands in these plots now represent the standard deviation across five independent runs. We have also added variants of Figure 3 in Appendix H that separately show the per-protein and per-seed variability to further illustrate stability across different systems. Finally, we have updated the results tables to include standard deviations alongside mean values for all reported metrics, providing a clearer picture of performance variability. We believe these additions address the reviewer's concerns and enhance the robustness of our reported results. We sincerely thank the reviewer for this constructive feedback.

---

> > ### Author Response · Authors · 2025-11-25
> > **Correction to multi-seed aggregation**
> >
> > Dear Reviewer Qv5B,
> >
> > In the process of running the final AlphaFolding repeat experiments and finalizing the multi-seed evaluations requested for the ensemble variability analysis (W3), we found a configuration inconsistency for some of the 240ns/1us repeats.
> >
> > We have fixed this and updated Table 2, Figure 3, and Appendix Tables 11 and 12 with the corrected values. Most importantly, **none of our trends or conclusions regarding stability/coverage/dynamics, or comparisons between methods have changed**.
> >
> > We have also included the updated table 2 below:
> >
> > | Model            | Time     | JSD $\downarrow$    | Rec $\uparrow$      | RMSD $\downarrow$   | AutoCor $\downarrow$ | CA% $\uparrow$       | AA% $\uparrow$       | CA+AA% $\uparrow$    |
> > | :--------------- | :------- | :------------------ | :------------------ | :------------------ | :------------------- | :------------------- | :------------------- | :------------------- |
> > | **MD (Oracle)**  | 240 ns   | 0.26                | 0.75                | 0.01                | 0.00                 | 99.53                | 96.83                | 96.36                |
> > |                  | 1 $\mu$s | 0.23                | 0.91                | 0.00                | 0.00                 | 96.25                | 86.50                | 82.75                |
> > | **MDGen**        | 240 ns   | 0.52 $\pm$ 0.01     | 0.38 $\pm$ 0.01     | 0.48 $\pm$ 0.01     | 0.25 $\pm$ 0.01      | 63.25 $\pm$ 2.10     | 87.83 $\pm$ 1.13     | 56.60 $\pm$ 2.10     |
> > |                  | 1 $\mu$s | 0.56 $\pm$ 0.01     | 0.36 $\pm$ 0.03     | 0.37 $\pm$ 0.02     | 0.39 $\pm$ 0.01      | 36.11 $\pm$ 7.34     | 56.99 $\pm$ 4.52     | 24.81 $\pm$ 4.30     |
> > | **Alphafolding** | 240 ns   | 0.61                | 0.16                | 1.74                | 0.14                 | 8.69                 | 0.81                 | 0.45                 |
> > |                  | 1 $\mu$s | 0.65                | 0.20                | 0.77                | **0.05**             | 9.66                 | 0.19                 | 0.06                 |
> > | **ConfRover-W**  | 240 ns   | 0.51 $\pm$ 0.01     | 0.42 $\pm$ 0.02     | 0.35 $\pm$ 0.01     | 0.39 $\pm$ 0.01      | 44.71 $\pm$ 1.55     | 73.13 $\pm$ 0.84     | 36.51 $\pm$ 1.22     |
> > |                  | 1 $\mu$s | 0.55 $\pm$ 0.02     | 0.45 $\pm$ 0.02     | 0.33 $\pm$ 0.02     | 0.38 $\pm$ 0.03      | 54.74 $\pm$ 1.79     | 62.32 $\pm$ 3.43     | 36.91 $\pm$ 1.39     |
> > | **STAR-MD**      | 240 ns   | **0.44 $\pm$ 0.01** | **0.59 $\pm$ 0.01** | **0.20 $\pm$ 0.02** | **0.03 $\pm$ 0.01**  | **85.16 $\pm$ 1.91** | **97.57 $\pm$ 0.13** | **83.15 $\pm$ 1.99** |
> > |                  | 1 $\mu$s | **0.46 $\pm$ 0.01** | **0.61 $\pm$ 0.02** | **0.13 $\pm$ 0.02** | 0.10 $\pm$ 0.02      | **88.47 $\pm$ 1.09** | **89.81 $\pm$ 0.65** | **79.93 $\pm$ 1.04** |

---

> > > ### Comment · Reviewer_Qv5B · 2025-11-26
> > >
> > > Thank you for the detailed responses to my questions and feedback. I appreciate the updates to the paper in terms of clarity and other fixes and maintain my score.

---

> > ### Author Response · Authors · 2025-12-02
> > **AlphaFolding-4D Runs Finished**
> >
> > We have now completed the multiple-rollout experiments for AlphaFolding-4D and have updated Tables 1 and 2 and Figures 3 and 13 accordingly in the revised manuscript. We thank the reviewer for their constructive feedback.

---

> ### Author Response · Authors · 2025-11-22
> **Part 2**
>
> > W4. Unexplained shading in Fig. 3. The caption omits what the shaded bands represent (presumably inter-protein variation).
> >
> > [...]
> >
> > Q4. Shading in Fig. 3: What statistic do the shaded bands represent? Standard deviation, percentile range, or something else?
>
> The shaded bands in the original Figure 3 represented one standard deviation across the different test proteins. In the revised Figure 3, the shaded bands represent the standard deviation across five independent runs. We have updated the caption accordingly to include this information. We appreciate the reviewer's attention to this detail and hope this resolves the concern.
>
> ---
>
> > W5. Minimal ablation scope. Only "w/o noise" and "w/ separate attention" are tested. These yield small metric changes, so it remains unclear which components truly drive the reported improvements.
> >
> > [...]
> >
> > Q5. Long-horizon ablations. Do the "w/o noise" and "w/ sep attn" variants remain stable in 240 ns - 1 microsecond rollouts?
>
> We appreciate the reviewer bringing up the importance of these ablation studies.
> During rebuttal, we have 1) reorganized the presentation of ablation studies to highlight the important findings; 2) expanded ablation experiments to 240ns and 1us benchmarks (see Appendix D); 3) included an additional ablation on the placement of spatio-temporal attentions by putting it before the diffusion module (which mirrors the architecture of ConfRover (Shen et al., NeurIPS 2025)). Below, we list our main findings from our ablations:
>
> - Historical contextual noise is critical to the stability of the generated trajectories, especially for long-horizon generation tasks. We have also added a new figure (Figure 11) illustrating the long-horizon instability of the "w/o noise" variant, further underscoring the importance of this mechanism for long-horizon stability.
> - Using separable attention (i.e. S+T) performs significantly worse than STAR-MD's spatio-temporal attention on coverage and dynamic fidelity metrics. This underscores the expressivity of the spatio-temporal attention mechanism.
> - Placing the spatio-temporal layer as a conditioning layer before the diffusion module underperforms our full model in both coverage and fidelity, and performs similarly to the separable attention variant. This indicates that spatiotemporal attention in the diffusion decoder is important for performance.
> - These findings are generally consistent across all simulation timescales.

---

### Author Response · Authors · 2025-12-02
**Rebuttal Summary for New Area Chair**

Dear Area Chair,

We thank the Area Chair for stepping in to handle our submission in this challenging situation. The AC's time and extra effort are sincerely appreciated, as are the reviewers' valuable contributions in evaluating our work.

We appreciate the unanimous positive recommendations received from reviewers in our initial submission and the constructive suggestions that helped further improve our work during the rebuttal. Below, we provide a concise summary of our paper, the review feedback, and our key rebuttal contributions to assist the rest of the review process.

## Paper Overview

We present STAR-MD, an autoregressive SE(3) diffusion model that leverages efficient joint spatiotemporal attention to generate stable, physically plausible protein trajectories over microsecond timescales. By resolving critical scalability and error-accumulation bottlenecks, STAR-MD establishes new state-of-the-art results on the ATLAS benchmark and enables robust extrapolation to long horizons where existing models fail.

## Review Summary

Reviewers unanimously acknowledged the work's contributions, highlighting four key strengths:

- **Strong Empirical Results:** STAR-MD sets a new state-of-the-art for coverage, kinetic fidelity, and structural validity on ATLAS 100ns, _"substantially outperform[ing] prior coordinate-based models"_ (R-Qv5B).
- **Long-Horizon Extrapolation:** STAR-MD demonstrates robust extrapolation to longer timescales, being _**"the first model to demonstrate stable, physically plausible generation out to the 1 microsecond timescale, a task where all baseline models [...] fail catastrophically due to either memory limits or error accumulation"**_ (R-QE3i).
- **Methodological Innovation:** STAR-MD's scalable and efficient causal diffusion transformer was described as _"likely to influence future [...] approaches"_ (R-Qv5B). Its _"significantly reduced memory requirements"_ (R-yvwX) were recognized as a _"massive practical win"_ (R-QE3i).
- **Theoretical Grounding:** The justification of STAR-MD's architecture via coarse-graining theory and the Mori-Zwanzig formalism was commended as _"very interesting and novel"_ (R-jt5g).

## Key Rebuttal Contributions & Discussion Status

We also received constructive feedback from the reviewers, focusing on: (1) **evaluation of functional dynamics** beyond equilibrium fluctuations (R-yvwX); (2) **comparative assessment** against state-of-the-art conformational ensemble samplers (R-jt5g); (3) **stress-testing stability and generalization** on extreme horizons and unseen proteins (R-jt5g, R-yvwX); and (4) **rigorous kinetic analysis** beyond standard metrics (R-QE3i).

In response, we undertook the following major actions:

1. Validated the model on the **Abl kinase** active-inactive transition (R-yvwX), successfully sampling the transition over 2.5 $\mu$s and capturing complex functional dynamics (**App. K.2**).
2. Compared against **BioEmu (Science 2025)** on ATLAS and CATH1 (R-jt5g), showing STAR-MD outperforms BioEmu on ATLAS (e.g., RMWD 2.85 vs 4.33) (**App. J**).
3. Generated 10 $\mu$s trajectories (100x training horizon) on ATLAS to demonstrate robustness (R-jt5g; **App. G**) and confirmed generalization on proteins significantly larger and dissimilar to the training set (R-yvwX; **App. L**).
4. Added new kinetic metrics (VAMP-2, Autocorrelation, RMSD) to further confirm STAR-MD's fidelity (R-QE3i; **App. C.4.1**).

The positive feedback from reviewers R-jt5g (6 $\to$ 8), R-QE3i (8), and R-Qv5B (8) indicates that our rebuttal has effectively addressed their concerns regarding comparative assessments, long-horizon stability, and kinetic fidelity. Although R-yvwX was unable to provide further comments due to the current situation, we are confident that our response provides strong evidence addressing all their concerns, including functional dynamics and generalization (items #1 and #3 in the list above, respectively).

---

We are happy to answer any further questions regarding our rebuttal or the revised manuscript. Again, we sincerely appreciate the Area Chair's additional effort in this special situation.

---

### Meta-Review · Area_Chair_zJZ6 · 2025-12-18

**Summary:**

The paper presents STAR-MD, an autoregressive SE(3) diffusion model designed to simulate long-horizon protein dynamics. The primary technical innovations include a joint spatiotemporal attention mechanism that operates only on single-residue features (improving memory scaling) and a contextual noise perturbation (diffusion forcing) to ensure stability during long rollouts.

The reviewers were unanimously positive regarding the model’s state-of-the-art performance on the ATLAS benchmark and its unique ability to generate stable, physically plausible trajectories up to the microsecond scale, where previous methods fail. A notable highlight is the theoretical soundness using the Mori-Zwanzig formalism, which justifies the need for non-Markovian modeling in coarse-grained protein systems.

To me, I hold concerns aligning with Reviewer jt5g regarding the model's performance on conformational ensembles compared to models like BioEmu. While STAR-MD excels at trajectory-based fidelity, its performance on free-energy landscapes for small folding proteins (CATH1) showed it was slightly behind BioEmu in coverage. This suggests that while the architecture is a significant leap forward in stability, its ultimate diversity may be limited.

Moreover, ATLAS is a relatively simple dataset for learning protein dynamics, as the energy countours are quite simple and the dynamics behave like local fluctuation. I would recommend authors to try STAR-MD on datasets with more complex energy landscape and longer horizon (e.g., MDCath).

**Reviewer Concerns:**

Addressed by Rebuttal:

- Theoretical Motivation (jt5g): The authors successfully moved the Mori-Zwanzig theoretical justification from the appendix to the main text (Section 3.4), improving the paper’s clarity and impact.
- Comparative Assessment (jt5g, Qv5B): Clarified that all baselines use comparable geometric representations (SE(3) rigid frames), correcting a misunderstanding that STAR-MD was "Ca-only".
- Long-Horizon Stress Testing (jt5g, yvwX): The authors generated 10 $\mu$s trajectories (100x the training horizon), demonstrating that the model maintains structural validity far beyond the training distribution.
- Ensemble Variability (Qv5B): Updated results and figures now include standard deviations and multi-seed analysis, confirming that the stability results are statistically robust.
- Functional Dynamics (yvwX): Added validation on the Abl kinase active-inactive transition ($2.5 \mu s$), showing the model can capture biologically relevant structural changes beyond equilibrium fluctuations.

Outstanding Concerns:

- Large-Scale Pre-training (jt5g): While addressed as a future direction, the model's reliance on a smaller dataset (ATLAS) compared to BioEmu (AFDB pre-trained) remains a point for potential improvement in absolute conformational diversity.

**Reviewer Scores:**

- Reviewer Qv5B: 8 (Maintained score after being satisfied with the clarification on representation and alignment).
- Reviewer jt5g: 8 (Increased from 6 after the authors moved the theoretical section and provided BioEmu comparisons).
- Reviewer QE3i: 8 (Maintained strong support for the clever architectural trade-offs).
- Reviewer yvwX: 8 (Estimated. Although they were absent from the final discussion, the authors fully addressed their requests regarding Abl kinase and generalization).

---

### Decision · Program_Chairs · 2026-01-26

Accept (Poster)